# The hinge-engineered IgG1-IgG3 hybrid subclass IgGh$_{47}$ potently enhances Fc-mediated function of anti-streptococcal and SARS-CoV-2 antibodies

Arman Izadi [1], Yasaman Karami [2,3], Eleni Bratanis [1], Sebastian Wrighton [1], Hamed Khakzad [2], Maria Nyblom [4], Berit Olofsson [1], Lotta Happonen [1], Di Tang [1], Martin Sundwall [1], Magdalena Godzwon [5], Yashuan Chao [1], Alejandro Gomez Toledo [1], Tobias Schmidt [6], Mats Ohlin [5], Michael Nilges [3], Johan Malmström [1], Wael Bahnan [1], Oonagh Shannon [1,7], Lars Malmström [1] & Pontus Nordenfelt [1,8] ✉

*Streptococcus pyogenes* can cause invasive disease with high mortality despite adequate antibiotic treatments. To address this unmet need, we have previously generated an opsonic IgG1 monoclonal antibody, Ab25, targeting the bacterial M protein. Here, we engineer the IgG2-4 subclasses of Ab25. Despite having reduced binding, the IgG3 version promotes stronger phagocytosis of bacteria. Using atomic simulations, we show that IgG3's Fc tail has extensive movement in 3D space due to its extended hinge region, possibly facilitating interactions with immune cells. We replaced the hinge of IgG1 with four different IgG3-hinge segment subclasses, IgGh$_{xx}$. Hinge-engineering does not diminish binding as with IgG3 but enhances opsonic function, where a 47 amino acid hinge is comparable to IgG3 in function. IgGh$_{47}$ shows improved protection against *S. pyogenes* in a systemic infection mouse model, suggesting that IgGh$_{47}$ has promise as a preclinical therapeutic candidate. Importantly, the enhanced opsonic function of IgGh$_{47}$ is generalizable to diverse *S. pyogenes* strains from clinical isolates. We generated IgGh$_{47}$ versions of anti-SARS-CoV-2 mAbs to broaden the biological applicability, and these also exhibit strongly enhanced opsonic function compared to the IgG1 subclass. The improved function of the IgGh$_{47}$ subclass in two distant biological systems provides new insights into antibody function.

The immune system generates antibodies against invading pathogens as an essential part of the adaptive immune response. VDJ gene recombination in B cells defines the basic antigen-binding properties of antibodies[1,2], and B cells selected within the lymphoid tissues undergo class-switching and somatic hypermutation to modify function and enhance binding affinity. The classical model of antibody variable and constant domain independence describes that class-switching only alters antibody effector function and does not alter

affinity[3]. However, some findings suggest that the constant domain could influence the binding to antigens[4-7], providing a potential mechanism where subclass and class-switching add further affinity diversity in addition to VDJ-recombination and somatic hypermutation[4,5].

The human IgG antibody class contains four subclasses with varying abilities to trigger effector functions such as complement activation, phagocytosis, and cellular cytotoxicity. These immune activities are mediated through binding to Fc-receptors on the surface of immune cells or binding to soluble complement proteins through the Fc domain[1]. These subclasses differ in the constant region of the heavy chain. Major differences between the subclasses lie in the hinge region that connects the Fab to the Fc domains (Fig. 1A). Although there are at least 13 allotypes of the IgG3 subclass[1], a common trait is

**Fig. 1 | IgG3 exhibits potent phagocytosis efficacy despite reduced antigen binding.** **A** The four subclasses of Ab25. The constant domains of the heavy chains are depicted in different colors. **B** The Y-axis shows percentage of IgG+ bacteria. The $K_D$ values (nM) with a 95% confidence interval (CI) are shown in each graph with statistical significance up to 99% CI compared to IgG1 due to non-overlapping CIs. **C** $MOP_{50}$ curves of the subclasses, bacteria were opsonized with 15 μg/mL of IgG. The Y-axis shows the percentage of bacteria+ THP-1 cells, while X-axis depicts the ratio of bacteria to phagocytes added (MOP). The $MOP_{50}$ values with a 95% CI are shown in each graph with statistical significance shown up to 99% CI compared to IgG1. The data points from (**B**) to (**C**) are from $N = 3$ independent experiments. **D** Shows the $MOP_{50}$ curves of all the antibody treatments in one graph. Statistical significance compared to IgG1 is ** for IgG3 (99% CI) * for IgG2 and IgG4 (95% CI). **E** Individual $MOP_{50}$ values from the $N = 3$ independent experiments. **F** Percentage of THP-1 cells with internalized bacteria across an MOP range. **G** Amount of bacteria

phagocytosed by the whole THP-1 cell population measured as median MFI of bacterial signal across MOP range. **H** Phagocytosis score for each mAb treatment across MOP 200-12.5. In (**F**)−(**H**) statistically significant fold-change differences between IgG3 and IgG1 are highlighted in each graph. **I, J** Ex vivo phagocytosis with neutrophils and monocytes, respectively, at MOP 25. For (**I**) and (**J**), repeated measures one-way ANOVA was used with Dunnett's post hoc test after multiple comparisons and compared against IgG3. The data points in (**B**)−(**J**) represent the mean value, and the error bars are in SEM. The phagocytosis score seen in (**I**) and (**J**) is normalized to Ab25 IgG1 for each experiment. In (**F**)−(**J**) $N = 6$ independent experiments were performed. $R^2 = 0.99$ for all non-linear regression curves. Statistical analyses for (**F**)−(**H**) were done by comparing IgG1 with one-way ANOVA multiple comparisons corrected by Dunnett's correction test. * denotes $p$-value < 0.05, ** denotes $p$-value < 0.01, and $p$-value > 0.05 is ns. Source data are provided as a Source Data file.

that they have the most extended hinge region. This increased hinge length makes them more flexible, while the shorter hinge region of IgG2 is the most rigid[8]. Furthermore, IgG3 is most efficient at activating the classical complement pathway[9]. IgG3 and IgG1 have similar affinities to Fc-gamma receptors in humans, while IgG2 and IgG4 have lower receptor affinity overall[10].

We have previously generated an IgG1 mAb (Ab25) against the M protein of *Streptococcus pyogenes*[11]. *S. pyogenes* has several virulence factors, of which the M protein is one of the most studied due to its importance for virulence[12]. It is a cell-wall-attached protein with a dimeric coiled-coil structure that extends out from the surface of the bacteria[12]. The M protein has several immune evasive mechanisms, such as binding antibodies via a Fc-binding portion, which can inhibit Fc-mediated phagocytosis[13,14] or reduce phagocytosis through antibody-induced fibronectin binding[15]. In addition, the M protein inhibits complement-mediated phagocytosis by sequestering C4BP and fibrinogen binding[16]. These mechanisms are important for *S. pyogenes* virulence and survival in the host. M protein is a target for vaccine development against *S. pyogenes*. However, no vaccines have been approved for clinical use due to a lack of efficacy or because they promote the emergence of autoreactive antibodies[17]. Generating distinct mAbs against *S. pyogenes* could be a viable therapeutic alternative. Indeed, the IgG1 Ab25 mAb was protective when used as a prophylactic in a mouse infection model where bacteria were inoculated subcutaneously to generate a systemic infection, which we will refer to as diffuse infection model[11]. We have previously observed that enriched binding of polyclonal IgG3 antibodies to M protein was linked to enhanced phagocytosis[18]. However, the importance of subclass for effective function against *S. pyogenes* or other bacterial pathogens is not well understood.

In this work, we generated all four human subclasses of Ab25 and observed that the IgG3 subclass had reduced binding to M1 protein compared to the other subclasses. Still, this antibody subclass had several-fold higher opsonic efficacy compared to the original IgG1 Ab25. We hypothesized that the IgG3 subclass efficacy could be explained by its more flexible hinge region. To analyze hinge flexibility, we modified the original IgG1 to contain the IgG3 hinge instead. Molecular dynamics (MD) simulations showed that the IgG3 mAb displays much higher flexibility than the original IgG1 due to its extended hinge region. In addition, our in silico data suggested that IgG3's weaker binding to M1 protein was due to a loss of Hbond formation between Fab1 (binding to C domain) of IgG3 and the antigen. We generated a panel of mAbs with different lengths of IgG3's hinge with the IgG1 backbone, named IgGh$_{xx}$. The hinge-modified variants displayed an IgG1-like binding phenotype, suggesting that hinge-engineering does not impair the binding to M protein as seen with IgG3. Interestingly, the hinge-modified variants exceeded IgG1 in opsonic function, with the 47-hinge variant having the best outcome. This 47-hinge-engineered mAb, IgGh$_{47}$, was comparable to the potent IgG3 variant in opsonic function but with IgG1's high-affinity binding.

Our in vitro findings were translated to an in vivo infection model in mice where the IgGh$_{47}$ exceeded its parent subclasses. The potent opsonic IgGh$_{47}$ phenotype relative to IgG1 was generalizable to clinical isolate strains of different *emm* types. Finally, we showed that the IgGh$_{47}$ hybrid subclass potently enhances the Fc-mediated function of three anti-SARS-CoV-2 mAbs.

## Results
### IgG3 exhibits potent phagocytosis efficacy despite reduced antigen binding

Previously, we generated a human IgG1 antibody, Ab25, against the M protein of *Streptococcus pyogenes*, which was opsonic in vitro and protective in vivo[11]. This antibody does not bind and react to human cardiac tissue as analyzed by tissue array[11]. This antibody was generated using a human B cell from a convalescent donor. The original subclass is not known. Using PCR and homologous recombination, we exchanged the subclass-specific domains of the heavy chain, generating three new plasmids encoding the remaining subclasses IgG2, IgG3, and IgG4 (Fig. 1A). The hinge length varies, with IgG1 having a 15 aa flexible hinge, IgG2 having a 12 aa rigid hinge (due to more disulfide bridges), and IgG4 having a 12 aa flexible hinge. IgG3 has about four times longer hinge region, distributed across 13 different allotypes with 32 to 62 aa hinge length[1], and here we used allotype IGHG3*11 with a 62 aa hinge.

In previous work, we attempted to measure the affinity of Ab25 IgG1 to both purified and recombinant M protein by using surface plasmon resonance (SPR). These attempts were unsuccessful. This is most likely due to the M protein's conformational stability being highly dependent on temperature, dimerization, and being attached to its native bacterial surface[19]. Instead, we measured the direct functional affinity (likely equivalent to avidity) of antibodies to M protein on live bacteria (Supplementary Fig. 1A)[11]. We calculated the dissociation constant ($K_D$) for all Ab25 subclasses using a non-linear regression analysis. This analysis revealed different affinity profiles across the subclasses (Fig. 1B). IgG1, IgG2, and IgG4 showed similar affinities (IgG1 $K_D = 2.2$; IgG2 $K_D = 1.8$; IgG4 $K_D = 4.7$ nM$^{-1}$, respectively). IgG3 showed a 13-fold decrease in affinity compared to IgG1 (IgG3 $K_D = 28.3$ nM$^{-1}$). Our results indicate that subclass-switching Ab25 IgG1 to IgG3 leads to reduced binding to the M protein, while binding affinity is retained for IgG2 and IgG4.

Having established the binding properties of the Ab25 IgG subclass variants, we investigated their functional output in terms of Fc-mediated phagocytosis. We used a previously established opsonophagocytosis assay for *S. pyogenes* with heat-killed SF370 bacteria and human phagocytes of the monocytic THP-1-cell line (ref. 11 Supplementary Fig. 1B, C). We verified that the THP1-cells expressed the human Fcγ receptors FcγRI, FcγRII, and FcγRIII (Supplementary Fig. 1D). The opsonic abilities of the subclasses were assessed by phagocytosis score as done in other work[20–22]. We also analyzed the interaction between prey and phagocyte by looking at detailed metrics

such as % of cells that are positive for bacteria (association), % of cells with internalized bacteria (internalization), and how much bacteria is involved in the overall interaction (bacterial signal). We use the term MOP[23], which is a relation between the ratio of prey to phagocyte (multiplicity of prey, MOP), which we varied in these experiments to capture different experimental conditions for a comprehensive overview. We calculated the number of bacteria relative to phagocytes needed to elicit a 50% association ($MOP_{50}$)[23]. A lower $MOP_{50}$ value represents a higher efficiency at mediating bacteria-monocyte interaction.

The $MOP_{50}$ analysis showed that IgG3 exhibited a potent opsonic phenotype (Fig. 1C). There was a 6-fold increase in opsonic ability ($MOP_{50}$ = 12, $P < 0.01$) compared to the original Ab25 IgG1 ($MOP_{50}$ = 68) and a 12-16-fold increase compared to IgG2 ($MOP_{50}$ = 145) and IgG4 ($MOP_{50}$ = 192) (Fig. 1D–E). To put these values into context, the negative IgG1 isotype Fc-control exhibited a 30-fold lower efficacy compared to IgG3 ($MOP_{50}$ = 367). All subclass isotype controls were similar (Supplementary Fig. 1E). We analyzed the percentage of phagocytes with internalized bacteria and observed a similar outcome as the $MOP_{50}$ analysis with the following order IgG3 > IgG1 > IgG2 > IgG4. (Fig. 1F). IgG3 significantly exceeded IgG1 between 1.5 to 4-fold for internalization across the various MOPs. Additionally, we also analyzed the amount of bacteria (Oregon Green-stained) associated with the THP-1 population (bacterial signal) (Supplementary Fig. 1B, C). This analysis shows that IgG3 promotes the most efficient phagocyte-bacteria interaction with 3-8-fold larger differences ($P < 0.05$) compared to IgG1 across all MOPs (Fig. 1G). Finally, we calculated a phagocytosis score based on the combined phagocyte association and bacterial signal (MFI) of the THP-1 population. This analysis shows that IgG3 is more potent than the other subclasses, with a 6-fold higher score than IgG1 at MOP 200 (27000 vs. 4600, Fig. 1H, $P < 0.05$) up to 20-fold higher score at MOP 20 ($P < 0.05$). The four assessed metrics ($MOP_{50}$, internalization, bacterial signal, and phagocytosis score) showed that Ab25 IgG3, despite its reduced reactivity to M protein, is a more potent opsonizing antibody than the other subclasses.

Furthermore, we verified our IgG3 findings in an ex vivo setting. Primary neutrophils and monocytes from healthy human donors ($N = 6$) were isolated to compare the opsonic performance of IgG3 with that of IgG1 using flow cytometry as done with the THP-1-system (Supplementary Fig. 2A–C, Supplementary Fig. 3A–C). Due to biological variability between donors, we normalized each experiment to the performance of Ab25 IgG1 when assessing function (except for internalization, where % was used). We did not observe a meaningful difference in association nor internalization for monocytes (Fig. 1I, J). Due to large biological variability, the $N = 6$ was not enough to show statistical significance for the differences observed for bacterial signal and phagocytosis scores for both monocytes and neutrophils (in the instances where IgG3 > IgG1). Due to the large spread attributed to biological variation, we added 6 more repeats ($N = 12$), to increase statistical power, and observed now a statistically significant effect size difference of a 1.3-fold and 1.5-fold enhancement for IgG3 over IgG1 for neutrophils and monocytes, respectively, in terms of phagocytosis score (Supplementary Figs. 2D and 3D). Across the three systems, with more modest effects seen in primary neutrophils and monocytes, our results show that the IgG3 subclass of Ab25, despite its 13-fold lower binding affinity, is a better opsonin than its parent IgG1 subclass in mediating Fc-mediated opsonization of bacteria.

## The IgG3 hinge provides broad Fc spatial mobility spectra and flexibility relative to M protein in silico

Our in vitro findings of increased opsonic ability of IgG3 aligned with other recent findings showing improved Fc-effector functions of this subclass[24–26]. However, we observed a discrepancy with lower binding yet stronger functional output. We hypothesized that this increased

functionality is due to the extended hinge region of IgG3. To address this question we performed in silico simulations with molecular dynamics (MD) simulations to compare the flexibility of IgG3 to that of IgG1. In addition, we designed a new mAb, $IgGh_{62}$, which contained IgG1's backbone with IgG3's 62 aa hinge to see if $IgGh_{62}$ and IgG3 exhibit similar phenotypes (as a control for the amino acid differences between IgG1 and IgG3). To assess if the hinge of IgG3 grants increased Fc flexibility, we performed three replicates of 1 μs molecular dynamics (MD) simulations for IgG1 (Supplementary Movies 1, 2, and 3), IgG3 (Supplementary Movies 4, 5, and 6), and the $IgGh_{62}$ (Supplementary Movies 7, 8, and 9). We determined a crystal structure of the Ab25 Fab to enhance the accuracy of the simulations (Supplementary Table 1, PDB ID: 8C67, Supplementary Fig 14). We have previously found that the Ab25 binds with both Fabs to two different epitopes on the M protein, so-called dual-Fab cis binding[11]. The data obtained by protein cross-linking mass spectrometry was supported by experimental validation showing that the single Fabs of Ab25 IgG1 could barely bind to M protein compared to F(ab')$_2$ fragments[11]. Our current MD analysis of the Fab-M1 interactions thus focused on this dual-Fab cis interaction.

To evaluate the stability of the generated trajectories, we measured the root mean square deviations (RMSD) of different domains (M1 protein, Fc, Fab1 (C domain- binding), and Fab2 (B domain-binding)) with respect to the equilibrated starting structure (Supplementary Fig. 4A–D, Supplementary Table 2). The Fab1 domain of the IgG1 system showed lower RMSD values (average of 3.72 ± 0.64) with respect to the Fab1 of both the IgG3 and $IgGh_{62}$ (average values of 5.76 ± 1.06 and 6.49 ± 1.48 Å, respectively). We also measured the root mean square fluctuations (RMSF) of every residue within different domains with respect to the average conformation (Supplementary Fig. 4E–H). Both IgG3 and $IgGh_{62}$ exhibited larger fluctuations of Fab1 (average RMSF of 2.18 ± 1.16 and 2.32 ± 1.33 Å, respectively) compared to IgG1 (average RMSF of 1.92 ± 0.86 Å), hinting at weaker binding to M1 for this Fab (Supplementary Table 2). More interestingly, $IgGh_{62}$ displayed larger fluctuations and flexibility in its Fc region (average RMSF of 2.88 ± 1.68 Å) compared to both IgG1 and IgG3 (average RMSF of 1.64 ± 1.01 and 1.63 ± 0.96 Å, respectively).

We investigated the dynamics of the Fc of the three mAbs relative to M1. We calculated the displacement between the center of mass of the Fc domain of all mAbs and the M1 protein over the time course of the simulations (Fig. 2A). The results showed larger displacement for both IgG3 and $IgGh_{62}$ with average values of 111.01 ± 24.50 and 167.13 ± 17.60 Å, respectively, compared to IgG1 (94.10 ± 6.77 Å, Supplementary Table 2). The larger mean displacement (movement of Fc in 2D) for IgG3 and $IgGh_{62}$ highlights how the hinge of IgG3 increases movement compared to IgG1. To further analyze the flexibility of the Fc domain in 3D space, we recorded the position of the center of mass of the Fc domain from all three mAbs in every conformation of the MD simulations, with the center of mass of the M1 protein maintained at the origin (Fig. 2B). Individual representations for the three systems are depicted in Fig. 2C–E. This analysis showed that the Fc domain of both the IgG3 and $IgGh_{62}$ systems spans a large 3D volume compared to the Fc domain of IgG1. Moreover, the traces of the Fc domain from IgG3 showed larger degrees of bending that bring the Fc domain closer to the M1 protein. However, in the case of $IgGh_{62}$, while the Fc domain remained flexible, it kept a certain distance from the M1 protein and explored the space differently. Thus, the hinge of IgG3 leads to both larger degrees of freedom and more flexibility, while the IgG1 hinge confers spatial constraints.

Furthermore, we analyzed three sets of angles: *1)* between the Fc domain of the IgGs and the M1 protein, *2)* between the Fc and Fab domains of the IgGs and *3)* between the two Fab domains of the IgGs (Fig. 2F). These angles were recorded from all MD conformations (Supplementary Fig. 5A–C); however, only the values where both Fab domains are in the proximity ( < 30 Å) of the M1 protein are of interest to this study due to the dual-Fab binding phenotype of Ab25. Thus, we

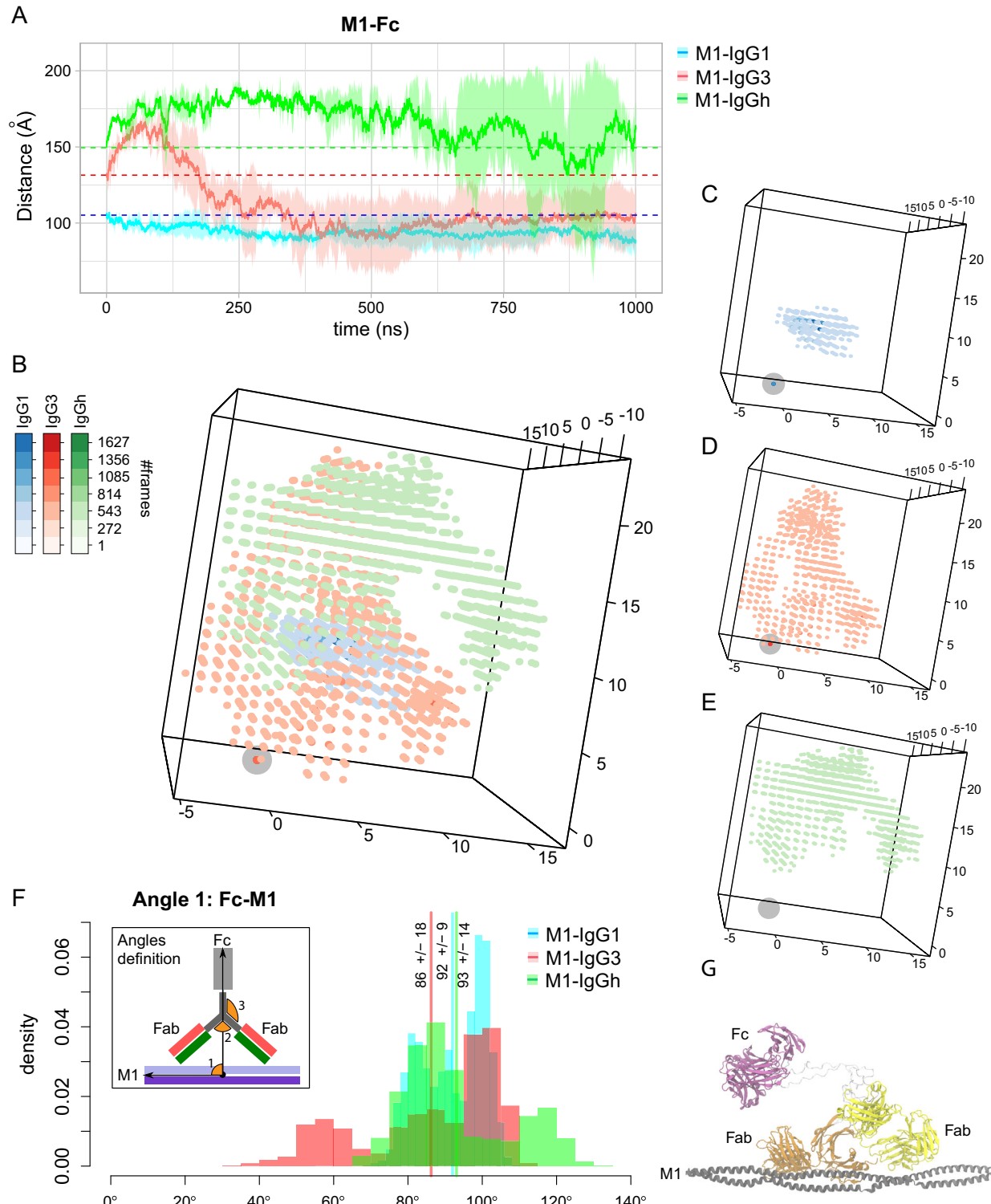

**Fig. 2 | The IgG3 hinge provides broad Fc spatial mobility spectra and flexibility relative to M protein. A** The average distance between the center of mass of the M1 protein and the Fc domain of the IgG, IgG3, and IgGh$_{62}$ was measured over the three replicates of each system (reported in pink and cyan, respectively). The shades correspond to standard deviations, and the initial distances are shown with dashed lines. **B** The center of mass of the Fc domain of IgG1 (in blue), IgG3 (in red), and IgGh$_{62}$ (green) are depicted at every snapshot of the simulations. The intensity of the colors represents the population of each point. The center of mass of M1 is shown with a gray circle at the origin. The individual distribution of points are

reported for **C** IgG1-M1, **D** IgG3-M1, and **E** IgGh$_{62}$-M1 simulations. **F** The angle changes formed between the Fc domain and protein M1 are reported for IgG1 (in pink), IgG3 (in cyan), and IgGh$_{62}$ (green) when both Fabs contact the protein M1. The average values are shown with lines. The vectors forming each angle are depicted in the inset. **G** The most representative conformation of the M1-IgG3 is reported when dual-Fab binding was observed. The IgG-M1 models in the figure are the starting models for the MD simulation. Source data are provided as a Source Data file.

focused on the conformations where dual-Fab binding occurred (Supplementary Fig. 5D–F).

The Fab1-Fab2 angles showed that the antibodies were tilted in different ways but with no substantial variation in the three systems during the simulations (Supplementary Fig. 5F, Supplementary Table 2). For the Fc-Fab angles (Supplementary Fig. 5F, Supplementary Table 2), we observed distributions of angles for IgG1, IgG3, and IgGh$_{62}$ to be within the ranges of 13° - 114°, 1° - 117°, and 0° - 77°, respectively. The average values of the Fc-Fab angles for the three systems were 66° ± 22°, 72° ± 31°, and 31° ± 14°, respectively. The angle span of the Fc-Fab for IgGh$_{62}$ (77°) was roughly two-thirds of the Fc-Fab angle span in the IgG1 (101°) and IgG3 (116°) systems. Considering the Fc-M1 angles (Supplementary Fig. 2D, Fig. 3F), the following spans were observed in the simulations: 69° - 108° for IgG1, 34° - 113° for IgG3 and 66° - 132° for IgGh$_{62}$, with the average values of 92° ± 9°, 86° ± 18° and 93° ± 14°, respectively. Thus, IgG3s Fc exhibited twice as large an angle span (79° vs. 39°) and IgGh$_{62}$ 1.7-fold (66° vs. 39°) compared to IgG1s Fc relative to the M1 protein (Supplementary Table 2) while the hinge of IgG3 did not exhibit a higher Fc-Fab angle as seen with IgGh$_{62}$ compared to IgG1 (77° vs 101°). Also worth noting is that IgG3 and IgGh$_{62}$ show a multimodal distribution between the replicates, which IgG1 does not for the Fc-M1 angle. Thus, the dynamic nature of these IgG-M1 systems leads to larger differences observed against IgG1.

Models for Fc receptor activation suggest that the accessibility of Fc domains is important to increase the avidity of the interactions[27,28]. Therefore, we further analyzed the most prevalent configurations (clusters) regarding the orientation of the Fc relative to the M1 protein for the respective antibodies (Supplementary Fig. 6). Cluster analysis of the MD trajectories revealed that dual-Fab binding occurred in two out of the three most prevalent clusters of the M1-IgG1 system. At the same time, the binding of only Fab2 was observed in the three most significant clusters of the M1-IgG3 system (Supplementary Fig. 6A). The results showed that, on average, the Fc domain of IgG1 is in a perpendicular-like position with respect to the M1 protein in the three most representative conformations (Supplementary Fig. 6B–D). The most prevalent configuration for IgG3 Fc is when it bends from the hinge region and fluctuates in parallel to the M1 protein away from the bacterial surface (Supplementary Fig. 6E, cluster 1 for the IgG3-M1 system, Supplementary Fig. 6A, Fig. 2G). For IgGh$_{62}$, we detected many clusters with small populations of conformations. This suggests large conformational changes and high flexibility of the IgG hybrid system during the simulations (Supplementary Fig. 5A). These results showed that the IgG3 hinge region increased the movement span. Taken together, the IgG3 hinge region provided IgG3 and IgGh$_{62}$ with broad spectra of both Fc mobility and flexibility relative to the M1 in 3D space, where the theoretical IgGh$_{62}$ gained IgG3's increased flexibility through the exchange of IgG1's hinge with that of IgG3's.

## The constant domain influences the interaction network between the IgG Fabs and M1 at an atomic level in silico

For the MD simulations analyzed across all three IgG-M1 systems the binding of Fab2 to the M1 was stable for all three antibodies, while the Fab1 showed a transient behavior and was most stable for IgG1 (Supplementary Figs. 4C, D, 4G, H). One replicate of IgGh$_{62}$ became dislodged from M protein during the simulation and was excluded from further analysis. To better understand the binding stability of both Fabs across all MD trajectories, we extracted the network of hydrogen bonds (Hbond) and salt bridges at the interface between the Fab domain of IgGs and the M1 protein (Fig. 3). This analysis revealed two stable salt bridges (E44-K58; E46-K65) formed between the VH domain in Fab2 of both IgG1 and IgG3 and the M1 protein (Fig. 3A, B). For the IgGh$_{62}$ replicates, E46-K65 salt bridges occurred only in one replicate, while two other salt bridges (E46-K56; E46-K58) were formed in the other replicate. These salt bridges provide insight into the stability of Fab2 seen during the simulations. Interestingly, although the

arrangement of salt bridges are different in IgGh$_{62}$, the analysis also revealed a new salt bridge between Fab1 and M1 (E76-K127). IgG3 and IgG1 did not share this interaction, and neither of these antibodies formed a salt bridge between their Fab1 and M1.

Moreover, we recorded a set of eight Hbonds for the Fab1 domain of IgG1, whereas only two Hbonds were observed for the Fab1 domain of IgG3 and IgGh$_{62}$ (Fig. 3C–E). These hydrogen bonds that IgGh$_{62}$ and IgG3 form are not identical and are formed by different amino acids. In addition, these amino acids in Fab1 of IgG3 and IgGh$_{62}$ bind to different residues on the M1 protein, which do not interact with Fab1 of IgG1. Furthermore, there are notable differences between the three IgG-M1 systems in the Fab2 hydrogen bond interaction network. IgG3 and IgGh$_{62}$ Fab2 form six and seven Hbonds, respectively, with the M1 epitope, while IgG1 Fab2 forms only two (Fig. 3C–E). It is important to note that the Fab2-M1 interactions for all three antibodies are very stable in the simulation (Supplementary Figs. 4C, D, 4G, H), most likely due to the presence of the two salt bridges mentioned before (E44-K58; E46-K65).

To further elucidate the intermolecular interactions between M1 and the IgG Fabs we performed binding free energy calculations using FoldX. Empirical scoring function of FoldX predicted lower binding free energy between the Fab 1 domain of the M1 protein in M1-IgG1 system compared to the other two systems, and a similar binding free energy between the Fab 2 domain and the M1 protein in all the three studied systems. (Supplementary Fig. 7). Moreover, we performed computational alanine scanning using FoldX to identify relevant residues for the interactions between the IgGs and M1 protein (Supplementary Figs. 8 and 9). For that, first we selected the Fab residues that are in close proximity of the M1 protein for each system (i.e. average distance <= 5 Å along the replicates of MD simulations). Then we reported the ΔΔG of those residues for all the MD snapshots (10,000 frames per replicate) of the corresponding replicates for Fab1 and Fab2 in Supplementary Fig. 8 and 9, respectively (the number of such replicates are shown in blue at the top of the figures). Residues forming H-bonds or salt bridges are colored in green. The empirical function predicted that most residues are important for binding between the Fab domains and M1 protein (i.e. ΔΔG > 0 in all or a large set of MD snapshots). In addition, FoldX also predicted other important Fab residues for the binding that are indirectly involved in the interaction network for each of the IgGs. For example, in the case of IgG1 residue I250 of Fab1 is covalently linked to S249, which in turn makes an H-bond with the K256 on M1 protein with a frequency of 46% along the first replicate of M1-IgG1. The analysis of H-bond showed that IgG1's Fab1 has 6 amino acid residues forming H-bonds with M1 (D54, R56, D222, S249, N313 and S314), while only 1 was formed for IgG3 (R87) and 2 for the designed IgGh$_{62}$ (S123 and K127).

Taken together, the analysis of the network of interactions suggests differences in both Fabs across all three IgG-M1 systems. The differences between IgG3 and IgG1 in Fab1 could explain the reduced binding of IgG3 to M1 measured previously (Fig. 1B). The Fab2 interactions with M1 appear more stable due to the salt bridges in all IgG systems. Nonetheless, our analysis suggests that the network of atomic interactions between the antigen and antibody Fabs can be altered when exchanging from IgG1 to IgG3 constant domain.

## Systematic characterization of IgG3 hinge reveals that hinge length influences Fc-mediated function

To fully understand the impact of IgG3's hinge flexibility on the Fc-mediated function of Ab25, we set out to engineer different segments of IgG3's hinge into an IgG1 backbone. All IgG3 hinges consist of a core hinge with 17 amino acids, followed by at least one 15 amino acid exon which can occur up to three times[1]. Thus depending on allotypes, IgG3 can have a 32 aa long hinge (core + one exon repeat), 47 aa long hinge (core hinge + 2 exon repeat) or a 62 aa long hinge (core + 3 exon repeats). To understand how different hinge segments influence

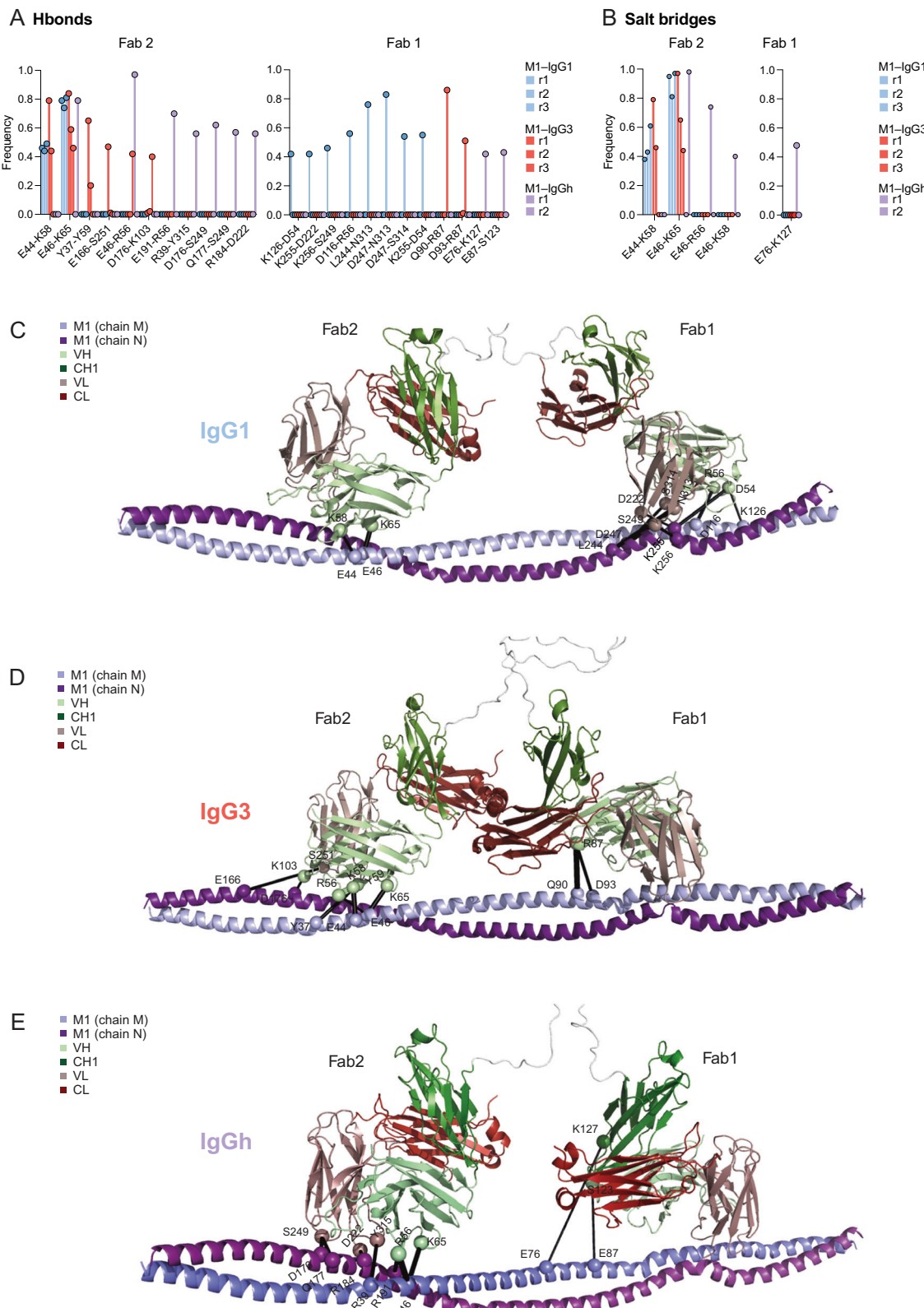

**Fig. 3 | The constant domain influences the interaction network between the IgG Fabs and M1 at an atomic level in silico.** The interaction network at the interface of Fab domain and protein M1. The frequency of **A** Hbonds and **B** salt bridges formed in every replicate of the IgG1-M1 and IgG3-M1 simulations is reported in red and blue shades, respectively. The interactions are also shown on the structure of **C** IgG1-M1, **D** IgG3-M1, and **E** IgGh₆₂-M1 with black lines for both Hbonds and Salt-bridges. The residues involved in the interactions are shown as spheres. The two chains of protein M1 are colored in light and dark purple, and the VH (variable heavy), CH1 (constant heavy domain 1), VL (variable light), and CL (constant domain light chain) domains are shown in light green, dark green, pink, and red, respectively. The IgG-M1 models in the figure are the starting models for the MD simulations. Source data are provided as a Source Data file.

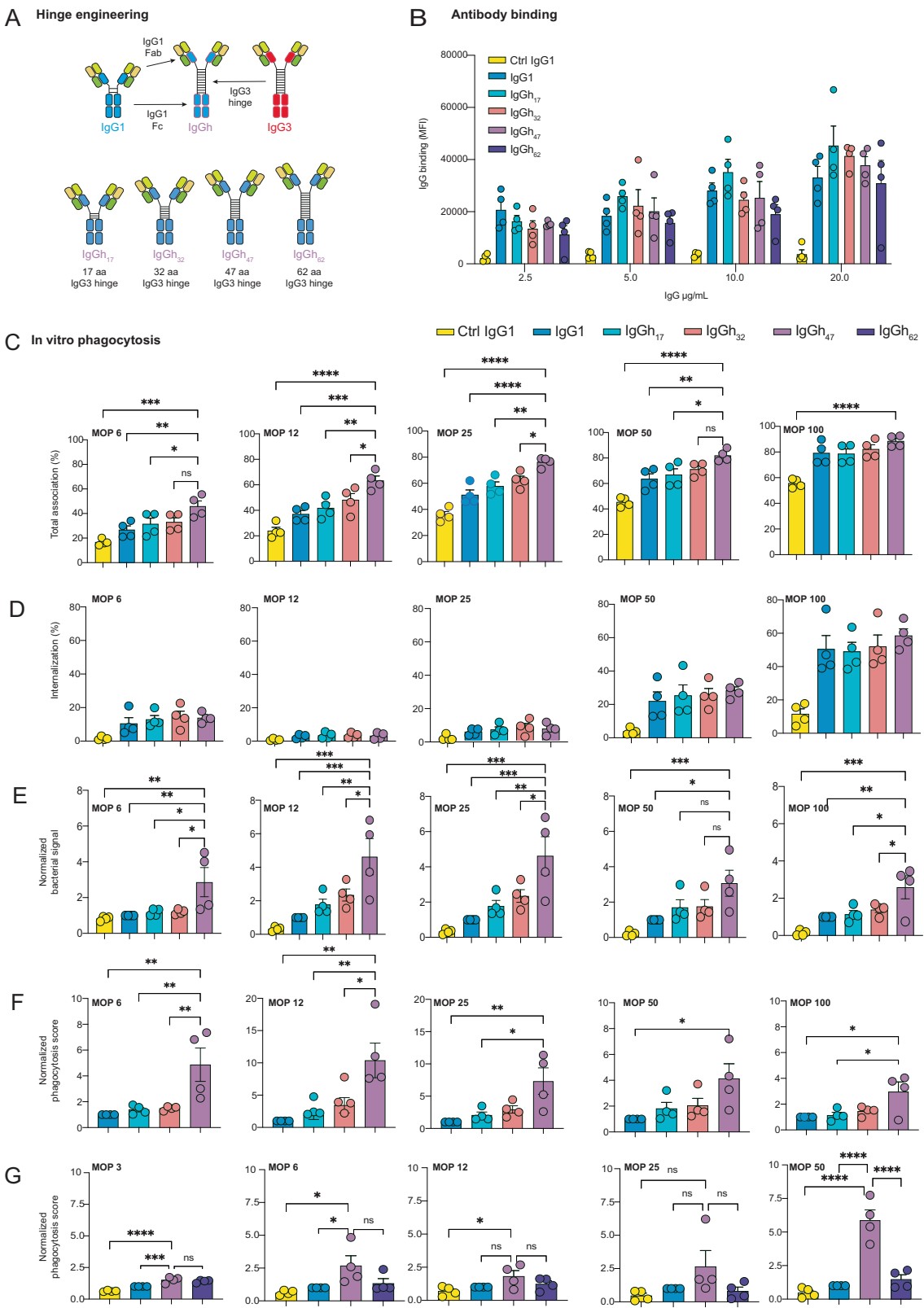

opsonic function, we generated four hybrid IgG1-IgG3's: one containing a 17 amino acid IgG3 hinge (IgGh$_{17}$, core hinge), one with a 32 aa (IgGh$_{32}$, core + one exon repeat) one with 47 aa (IgGh$_{47}$ core + two exon repeats), and one with a 62 amino acid hinge (IgGh$_{62}$, core + three exon repeats Fig. 4A). The IgGh$_{62}$ mAb corresponds to the designed control used in the MD simulations in Figs. 2–3. Comparing IgG1 to these constructs enables us to test whether increasing flexibility could

increase Fc-mediated function as seen in the IgG3 version. First, we characterized the binding reactivity of these hybrid constructs to M1 protein expressed by live SF370 bacteria as done before. These experiments demonstrated intact binding to M1 comparable to IgG1 for all IgGh$_{xx}$ versions (Fig. 4B).

We then proceeded to assess the opsonic function of these constructs. Phagocytosis experiments with THP-1 cells reveal that hinge

**Fig. 4 | Systemic characterization of IgG3 hinge reveals that hinge length influences Fc-mediated function. A** Schematic illustration of hinge-modified variants with Ab25 variable domains, IgG1 backbone but different segments of IgG3 hinge (17, 32, 47 or 62 amino acids). **B** IgG binding to live SF370 bacteria at different concentrations. **C** % of phagocytes associating (FITC +) with bacteria at different MOP's, while **D** shows the same but with % internalized and **E** shows the normalized bacterial signal and **F** shows the phagocytosis score. In **C**–**F** IgG1, IgGh 17, 32, and, 47 hinge variants are assessed. **G** Shows the phagocytosis score at different MOP's for the 62 hinge-variant, 47-variant and IgG1. In B-F 15 ug/mL of mAbs was used. The

mean is shown in all panels, and the error bars are SEM. The phagocytosis score in **E**–**F** is normalized to Ab25 IgG1's phagocytosis score for each individual experiment ($N = 4$ independent experiments). Data in **B**–**G** are from $N = 4$ independent experiments. Experiments in G were done separately from those at B-F. All statistical comparisons were made against the IgGh47 variant. The statistical test was one-way ANOVA with multiple comparisons corrected by Dunnett's post hoc test. **** denotes $P$-value < 0.0001, *** denotes $P$-value < 0.001, ** denotes $P$-value < 0.01, * denotes $P$-value < 0.05 and $P$-value > 0.05 is ns. Source data are provided as a Source Data file.

length directly influences Fc-mediated function. Ab25 IgGh$_{17}$ is comparable to Ab25 IgG1, while the Ab25 IgGh$_{32}$ seems to exceed both the shorter IgGh$_{17}$ variant and Ab25 IgG1. While in turn, the Ab25 IgGh$_{47}$ is superior to all three mAbs (Fig. 4C–E). An increase in hinge length appears to lead to both an increased ability of phagocytes to associate with bacteria (Fig. 4C) and a large increase in the amount of bacteria being engulfed (Fig. 4E), thus resulting in an overall rise in phagocytosis as defined by phagocytosis score (Fig. 4F). No large and statistical differences were seen for the proportion of cells that internalized bacteria across the different variants. We next compared the most extended hinge construct, the IgGh$_{62}$ hybrid with that of both IgG1 and IgGh$_{47}$ since the latter was the most potent construct so far. Interestingly, although the IgGh$_{62}$ hybrid exceeded IgG1 (at specific MOP's such as MOP 3) it was remarkably inferior in phagocytosis score to the IgGh$_{47}$ (Fig. 4F). Thus overall, these results clearly show that increasing hinge flexibility overall correlates with improved function but there is indeed an optimal length for the function. In the case of Ab25 and streptococcal M1 protein, the 47 aa hinge length of IgGh$_{47}$ demonstrated the best balance between flexibility and function.

## IgGh$_{47}$ subclass retains the high affinity of IgG1 and gains the high opsonic function of IgG3

Having established that hinge length influences Fc-mediated function, we set out to investigate how the potent IgGh$_{47}$ compares to IgG3 Ab25 (Fig. 5A). We first measured binding affinity directly through direct fluorophore-labeling of both IgG1 and IgGh$_{47}$. These binding assays show that IgG1 and IgGh$_{47}$ have the same affinity to the M1 protein (Supplementary Fig. 10). Then, we analyzed IgGh$_{47}$'s ability to promote phagocyte-bacteria association by measuring the MOP$_{50}$, showing that IgGh$_{47}$ and IgG3 exceeded IgG1 (Fig. 5B, C). Adding the IgG3 hinge (47 aa version) significantly improved the opsonic efficacy for IgGh$_{47}$ 2.5-fold over IgG1 (MOP$_{50}$ = 25, Fig. 5B, C, $P < 0.01$). Regarding promoting bacterial association, IgGh$_{47}$ was comparable to the potent IgG3 (Fig. 5C). All three mAbs had a much higher association than the negative control (MOP$_{50}$ = 188) (Fig. 5B, C, $P < 0.05$). We further assessed the opsonic function of IgGh$_{47}$ by looking at the proportion of THP-1 cells with internalized bacteria and the level of bacterial fluorescence. Across the various MOPs, IgGh$_{47}$ performed better than both parent subclasses in internalization with more phagocytes internalizing bacteria, with a statistically significant effect at MOP 100 (15 % for IgGh$_{47}$ vs 7% for both parent mAbs, $P < 0.05$) (Fig. 5D). We also analyzed the quantity of phagocytosed bacteria, which was the highest with IgGh$_{47}$ as the opsonin and similar for phagocytosis score, but these differences in these sets of experiments were not statistically significant (Fig. 5E–G).

We performed phagocytosis assays with primary monocytes and neutrophils as additional effector cell lines to see if IgGh$_{47}$ exceeds IgG3 and IgG1 in an ex vivo setting. There was no increase in association with IgG3 and IgGh$_{47}$ for monocytes and neutrophils (Fig. 5H, I). IgGh$_{47}$ exceeded IgG1 and IgG3 in terms of internalization ability with a modest 1.2-fold increase for neutrophils (Fig. 5H, I, $P < 0.05$). IgGh$_{47}$ was comparable to IgG3 in phagocytosis score for monocytes (1.9-fold vs. 1.3-fold) and neutrophils (1.3-fold vs. 1.2-fold) and statistically significantly better than IgG1 for neutrophils in phagocytosis score ($P < 0.05$). Our results show that IgGh$_{47}$ is comparable to IgG3 as an

opsonin and far exceeds IgG1 with THP-1 (ADCP) and exceeds IgG1 for neutrophil phagocytosis (ADNP). Ab25 IgGh$_{47}$ also has the added benefit of IgG1's high affinity – combining qualities from both parent subclasses.

## IgGh$_{47}$ confers protection against systemic infection in mice

To study the biological relevance of increased hinge engineering and to assess the potent IgGh$_{47}$, we proceeded with in vivo experiments. We used an animal model of invasive skin infection and passive immunization where C57BL/6 J female mice were prophylactically pretreated 6 hours before infection with IgG1, IgG3, IgGh$_{47}$, or PBS (Fig. 6A). Bacteria were then administered on the flank in a low volume to generate a localized tissue infection. In this model, the bacteria disseminate systemically from this localized infection foci, which we refer to as a two-stage flank infection model[29]. This model differs from the previous model used to assess Ab25 IgG1, where the mice were injected with higher volumes in the scruff to generate a diffuse tissue infection that instantly becomes systemic (single-stage diffuse infection model, refs. 11,29). The two-stage infection model was chosen over the single-stage infection model because it has recently[29] been shown to better capture the transition from local bacterial infection to systemic infection in a more clinically relevant model that mimics how invasive infection occurs in patients. In addition, the bacterial anti-IgG enzymes IdeS and EndoS are active in this model, and it is speculated that the previously used model does not have these upregulated enzymes since it does not start as a localized infection[29]. We assessed the protective effect of the antibodies by quantifying the percentage of mice with an adverse outcome of bacterial dissemination to distant organs (spleen, liver, and kidneys) and the colony-forming units of those infected tissues (CFU).

Interestingly, in this model, both IgG3 Ab25- and IgG1 Ab25-treated mice showed a tendency of lower bacterial dissemination in the organs compared to PBS by analyzing the median CFU count for each organ (Fig. 6B). However, since the difference with PBS was minor, this is not interpreted to be a protective effect. More importantly, there was no statistically significant difference in terms of preventing bacterial dissemination compared to PBS for either IgG1 nor IgG3 (the effect size was also small, Fig. 6C). Cytokine levels were analyzed in blood plasma samples and showed similar levels of MCP-1, IL-6, and TNF-alpha levels between the IgG1 and IgGh$_{47}$ groups, with elevated levels in the IgG3-treated group (Fig. 6D). Interestingly, we did not observe a notable difference between the different groups concerning the CFU levels in mice with bacterial dissemination. Only in the spleen analysis did IgGh$_{47}$-treated mice show statistically significant efficacy compared to IgG3-treated mice ($P < 0.05$, median CFU count 14 vs $10^3$) and against PBS in the liver. More importantly, IgGh$_{47}$ stood out as the only mAb treatment with a statistically significant protective effect compared to PBS in preventing bacterial dissemination to all three organs (Fig. 6C, E). While the mouse baseline immune system (PBS group) could only clear 10% of the cohort from bacteria, the equivalent comparison for IgGh$_{47}$ was between 60-80% ($P < 0.05$, Fig. 6E) IgGh$_{47}$ showed increased efficacy compared to IgG1 with statistical significance for the liver (80% bacteria free vs 20% bacteria free for IgG1, $P < 0.01$). Compared to the parent IgG3, larger significant differences in efficacy were observed for the liver (80% vs 30%,

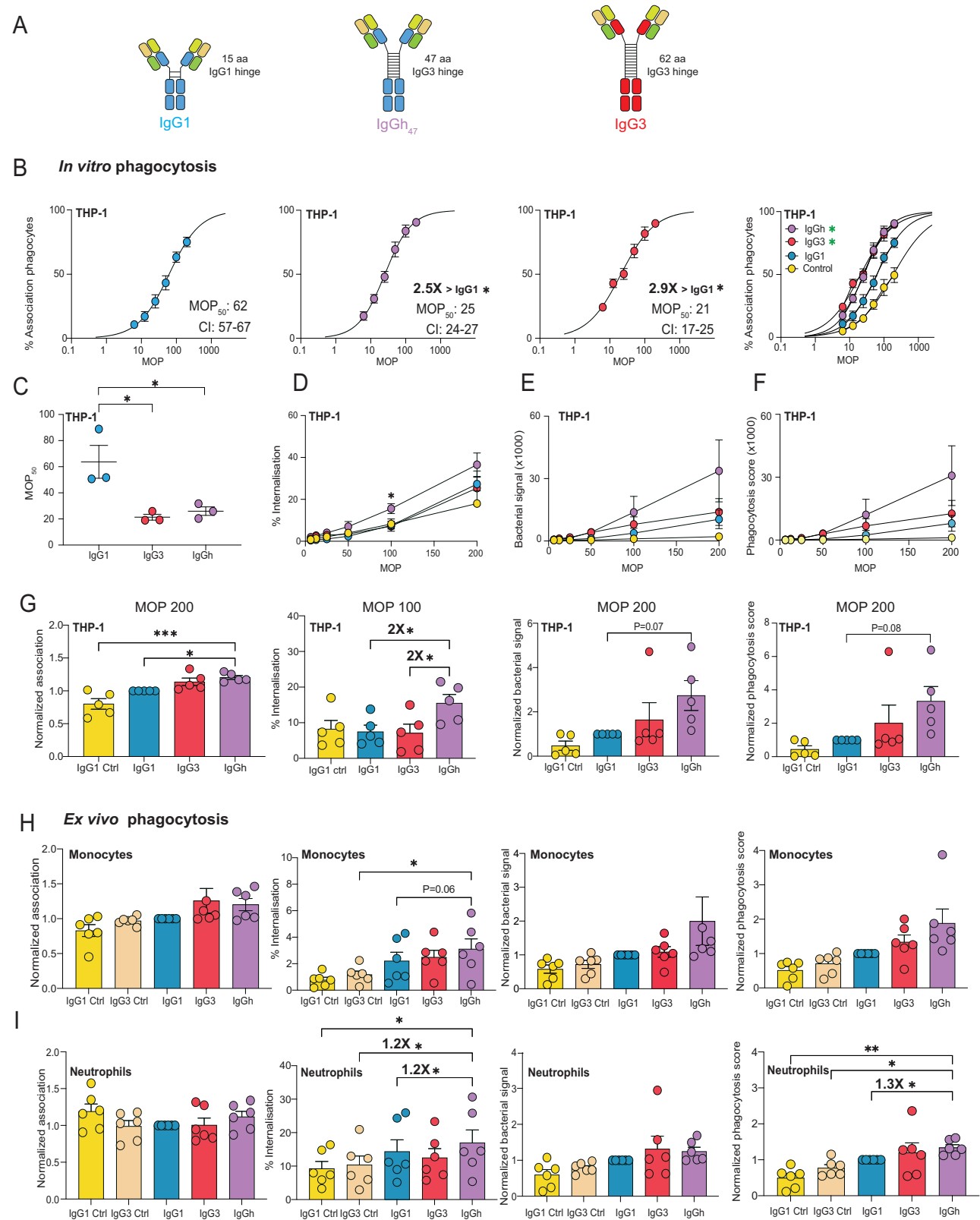

P < 0.05) and spleen (70% vs 20%, P < 0.05). The in vivo data shows that IgGh$_{47}$ exhibits a protective phenotype to prevent bacterial dissemination across all organs compared to PBS, while IgG1 and IgG3 failed to show any efficacy compared to the negative control. Finally, IgGh$_{47}$-treated mice were better protected than mice in the IgG1 group in the liver while better than IgG3-treated mice in the liver and spleen, with no added benefit compared to both IgG treatments in the kidney.

To determine bioavailability, we measured the concentration of human IgG1, IgG3, and IgGh$_{47}$ in plasma from infected mice after 24 hours. Using LC-MS/MS, we quantified the human proteotypic IgG peptides and observed similar levels of IgG1, IgG3, and IgGh$_{47}$ in blood (Fig. 6E, F). Interestingly, we observed more intact IgGh$_{47}$ by quantifying previously established IdeS cleavage signature peptides[30], indicating that IgGh$_{47}$ is less prone to IdeS degradation (Fig. 6G), but

**Fig. 5 | IgGh$_{47}$ subclass retains the high affinity of IgG1 and gains the high opsonic function of IgG3. A** Schematic of IgG1, IgG3, and the IgGh$_{47}$. The constant domains of the heavy chains are depicted in different colors: blue for IgG1 and red for IgG3. **B** MOP$_{50}$ curves of the subclasses, bacteria were opsonized with 15 ug/mL of IgG. The Y-axis shows the percentage of bacteria+ THP-1 cells, while the X-axis depicts the ratio of bacteria to phagocytes added (MOP). The MOP$_{50}$ values with a 95% CI are shown in each graph, and fold differences over IgG1 (non-overlapping 95% CI) to show statistical significance. The data points are from $N = 3$ independent experiments. **C** Shows the MOP$_{50}$ curves of all the antibody treatments in one graph, control is IgG1-isotype. **C** Individual MOP$_{50}$ values from $N = 3$ independent experiments. **D** Percentage of THP-1 cells with internalized bacteria across the MOP range. **E** Amount of bacteria that are phagocytosed by the whole THP-1 cell population measured as median MFI of FITC-A (bacterial signal, Oregon Green). **F, G** Phagocytosis score for each antibody treatment is shown with MOP on the X-axis for **F**, while in **G**, the MOP is 200 except for internalization when the MOP is 100. **H-I** shows Ex vivo phagocytosis with neutrophils and monocytes at MOP 25. The graphs depict %association (normalized to Ab25 IgG1), % internalization, bacterial signal (normalized to Ab25 IgG1) and phagocytosis score (normalized to Ab25 IgG1). For **C–G**, statistical analysis was done by comparing the treatments to IgG1 with one-way ANOVA multiple comparisons corrected by Dunnett's post hoc correction test. For **H** and **I**, repeated measures one-way ANOVA was used and compared against IgGh$_{47}$. *** denotes $P$-value < 0.001, ** denotes $P$-value < 0.01, * denotes $P$-value < 0.05 and $P$-value > 0.05 is ns.* denotes $P$-value < 0.05 and $P$-value > 0.05 is ns. The data points in (**B**)–(**I**) represent the mean value, and the error bars are in SEM. In (**D**)–(**G**), $N = 5$ independent experiments were performed, while in (**H**)–(**I**), data was acquired from $N = 6$ independent experiments and unique donors ($N = 6$). Source data are provided as a Source Data file.

overall, all mouse plasma had significant levels of intact IgG which is to be expected given the time frame of the experiment and the large dose given.

Furthermore, since the half-life of IgG in circulation is influenced by the affinity to the neonatal Fc-receptor[25], we performed SPR experiments to assess IgG1, IgG3, and IgGh$_{47}$ affinity to human and mouse Fc neonatal receptors in addition to mouse Fc-gamma receptors (CD16, CD16-2, CD64). While the binding characteristics of IgG1, IgG3, and IgGh$_{47}$ to human CD64 (FcgRI) were distinctly similar, the affinities of one IgGh$_{47}$ mAb (Ab77) differed compared to the other two (Ab11 and Ab36) for some mouse receptors, suggesting that the variable domains or the light chain constant domain might impact the interaction with various FcR. Clone-specific differences have been observed in other work[31]. Some antibody-FcR interactions (Supplementary Fig. 11) for Ab77 IgGh$_{47}$, did not follow a standard 1:1 interaction model, suggesting that receptor binding was heterogeneous in the population of molecules. Therefore, Ab11 and Ab36 IgGh$_{47}$ provides a more reliable understanding on this construct's interaction with FcRs while Ab77 IgGh$_{47}$'s FcR-affinity profile needs to be interpreted more cautiously. Furthermore, IgG3, as expected, did not bind efficiently to mouse FcRn at pH 6.2, both human IgG1 and IgGh$_{47}$ did bind, suggesting that the hinge-modified version of IgG1 carrying an IgG3-like hinge did bind this receptor and, thus is likely, just as IgG1, to have a prolonged in vivo half-life in comparison to IgG3.

## IgGh$_{47}$ potent opsonic phenotype is generalizable to other *emm* types

To expand on our potential translational findings with IgGh$_{47}$, we tested if its in vitro phenotype can be observed against other *emm* types. These experiments would also help us understand if this mAb remains cross-reactive as its parent IgG1 version. We chose to focus on *emm* types in different clusters: *emm*4 (E1), *emm*12 (A-C4), *emm*28 (E4), *emm*79 (E4), *emm*81 (E6), and *emm*179 (no specific designated cluster)[32]. We choose clinical strains for these selected *emm* types which had previously been isolated from patients with previous group A streptococcal infection – increasing the translational relevance of these experiments. These clinical isolates have been analyzed previously by NGS to confirm the correct *emm* type[11]. The differences between the *emm* sequences covering Ab25's dual epitope span can be seen in Fig. 7A and B. We tested the opsonic performance of IgGh$_{47}$ against IgG1 Ab25 to test if our phenotype with *emm*1 (A-C3 clade) is generalizable to other diverse *emm*-types. Interestingly, we observed that for *emm*4, *emm*12, *emm*28, *emm*79, and *emm*81 IgGh$_{47}$ exceeded IgG1 in opsonic performance when looking at internalization (statistically significant for *emm*4, *emm*28 and *emm*79), and bacterial uptake (statistically significant for *emm*4, *emm*28 and *emm*79). For phagocytosis scores, the trend followed those metrics. Depending on the *emm* type, the differences varied. For *emm*4 it was 3-fold higher performance IgGh$_{47}$ (11000 vs. 33000 $P < 0.05$); for *emm*12 it was 2-fold (30000 vs. 62000, P = 0.076); *emm*28 was 2.6-fold higher (57000 vs.

147000, $P < 0.001$); *emm*79 2.4-fold (10000 vs. 24000, $P < 0.05$); emm81 1.5-fold (60000 vs 92000, P > 0.05) and for *emm*179 it was 1.2-fold (230000 vs. 270000, P > 0.05) (Fig. 7C–F). Both mAbs widely exceeded the negative control, highlighting the potency of this specific clone (Fig. 7C–F). These results show that the opsonic function of IgGh$_{47}$ is generalizable to other *emm* types, including clinical isolates.

## SARS-CoV-2 mAbs engineered from IgG1 to IgGh$_{47}$ subclass exhibit a potent opsonic function

Finally, we wanted to test if hinge-engineering and increasing antibody flexibility could enhance mAb function in other biological contexts. We focused on SARS-CoV-2 because Fc-mediated effector functions of antibodies, non-neutralizing and neutralizing, are highly relevant in immune protection against SARS-CoV-2 and its mutated variants[33–38]. We have previously generated anti-SARS-CoV-2 IgG1 mAbs against the spike antigen[38] and shown that these mAbs are opsonic[37,38]. We aimed to elucidate if increasing Fc flexibility of some of these opsonic anti-spike IgG1 mAbs would also improve Fc-mediated function and potentially be useful in a SARS-CoV-2 setting.

We generated three IgGh$_{47}$ constructs of the mAbs Ab11, Ab36, and Ab77, which, as IgG1's, all bind to the spike protein with nanomolar affinity[37] (Fig. 8A). As done before[38], we utilized biotinylated spike protein conjugated to streptavidin microsphere beads as models for virions (Supplementary Fig 13). These mAbs all elicit Fc-mediated effector functions[37,38]. We performed MOP curves and calculated MOP$_{50}$ values for each respective mAb. We used Xolair (anti-IgE) as a negative control and a two-antibody cocktail of IgG3 mAbs with potent opsonic function[37] as a positive control. Interestingly, we observed a significant enhancement of function for all three clones. For Ab11, the IgGh$_{47}$ version enhanced bead-phagocyte interaction by 1.4-fold (MOP$_{50}$ 27 vs. 37, $P < 0.05$), and with Ab77, a 1.7-fold enhanced function was observed (MOP$_{50}$ 26 vs. 43, P > 0.05) (Fig. 8B). Ab36 IgGh$_{47}$ had a remarkable enhanced function with a 4.3-fold improvement in MOP50 value (9 vs 39, $P < 0.05$) (Fig. 8B). As done with Ab25 we calculated phagocytosis score (PS) to see if the overall bead-uptake differs between the engineered IgGh$_{47}$ and IgG1 mAbs. As the MOP was increased, the differences between the IgG versions became apparent: for Ab11, there was an improvement with a 4-fold increase at MOP 30 (3000 vs. 800) (Fig. 8C). For Ab77, we observed a 6-fold increase (3300 vs. 550), and for Ab36 there was the most substantial improvement over IgG1 with a 13-fold (8400 vs. 600) increase in PS (Fig. 8C). We performed additional experiments at MOP 30 to assess if these changes were statistically significant with the respective mAbs and looked at association and PS. Here, we observed similar findings with remarkable IgGh$_{47}$ enhanced function over IgG1 (Fig. 8D, E) for all three clones, which was statistically significant for both clones 36 and 77. More interestingly, Ab36 IgGh$_{47}$ is shown to be comparable to our most potent opsonin[37], DuomAb IgG3. Taken together, increasing antibody flexibility through hinge engineering shows significantly enhanced efficacy in two different and biologically relevant systems. In

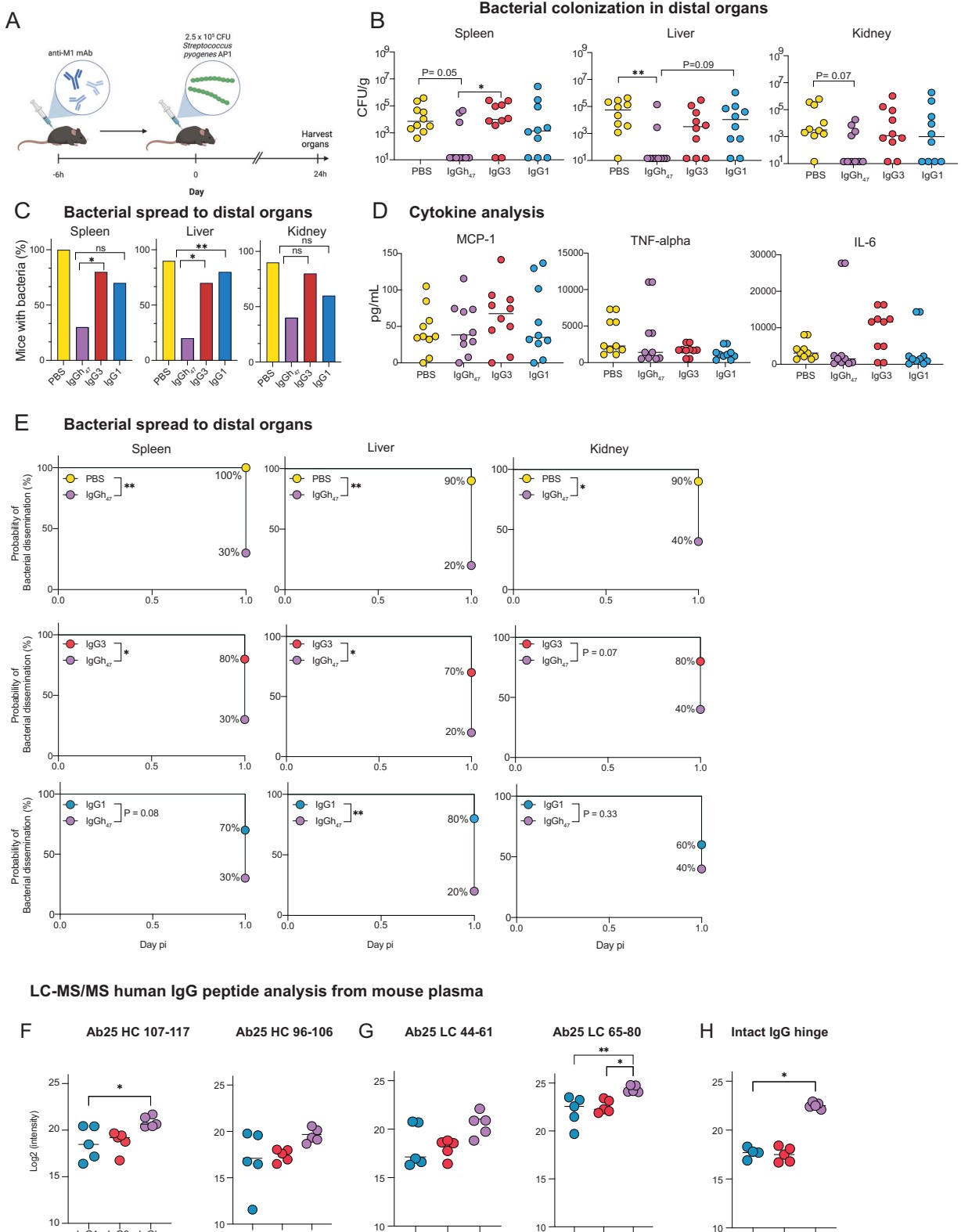

**B Bacterial colonization in distal organs**

**C Bacterial spread to distal organs**

**D Cytokine analysis**

**E Bacterial spread to distal organs**

**LC-MS/MS human IgG peptide analysis from mouse plasma**

**F** Ab25 HC 107-117 · Ab25 HC 96-106 · **G** Ab25 LC 44-61 · Ab25 LC 65-80 · **H** Intact IgG hinge

that regard, Ab36 IgGh$_{47}$ is a promising candidate to further test in vivo in a SARS-CoV-2 context due to its potent Fc-mediated effector functions.

## Discussion

Antibody-based therapeutics are emerging as a rapidly growing class of drugs with a great span in disease targets, ranging from oncology to rheumatology[39]. In the field of infectious diseases, clinical success has been chiefly made with antibodies targeting viral pathogens such as RSV[40], HIV[41], and SARS-CoV-2[42]. Today, most antibodies considered for clinical use are Fc-engineered to exhibit beneficial pharmacokinetics, such as longer half-life[39–41]. However, few clinically approved mAbs have been modified to include beneficial properties of the different subclasses, such as inserting the IgG3 hinge in the backbone of IgG1 to

**Fig. 6 | IgGh$_{47}$ confers protection against systemic infection in mice.**
**A** Schematic of the animal experiment: pre-treatment 0.4 mg of antibody treatment or PBS 6 hours before infection with AP1, made using Biorender. Animals were sacrificed 24 hours post-infection, and organs and blood were harvested for analysis. **B** Shows pooled data from two independent experiments of 5 mice in each cohort (total $n = 10$ mice). Bacterial load (CFU/g) in each organ was determined by serial dilution and viable count determination after overnight incubation (undetectable bacteria was labeled as 14 CFU/g). **C** shows the percentage of mice positive for bacteria in each respective organ for all treatments with statistical significant differences (or non-significance) compared with IgGh$^{47}$ seen above each treatment from test in (**E**). **D** Cytokine levels (pg/mL) in plasma were measured using a cytometric bead array at two different occasions analyzing 5 mice/condition at time ($N = 10$). **E** Kaplan-Meier curves are depicted in E for PBS vs. IgGh$_{47}$, IgGh$^{47}$ vs. IgG3, and IgGh$_{47}$ vs. IgG1 comparisons. The Y-axis shows the probability of bacterial

dissemination (adverse outcome) with log-rank (Mantel-Cox) tests used to determine statistical significance, which are also shown for the PBS comparison in the table in (**C**). **F**−**H** Ab25-derived peptides identified in the plasma of antibody-treated mice challenged with GAS. Plasma was collected at 24 h postinfection (n = 5 animals/condition; treatments corresponded to Ab25 with an IgG1, IgG3, or IgGh$_{47}$ and the peptides were identified by Protein G pulldowns, trypsin digestion and mass spectrometric analysis. Log2 intensity of peptides derived from **F** the heavy chain, **G** light chain and **H** hinge regions of the monoclonal antibodies. In **B** and **D**, the median is shown. Statistical analysis was performed, comparing the treatments to the IgGh$_{47}$ in **B**, by Kruskal-Wallis with multiple comparisons and Dunn's correction test. ** denotes $P$-value < 0.01, * denotes $P < 0.05$ and $P$-value > 0.05 is ns across the figure. In **E**, log-rank (Mantel-Cox) test was performed, in **F**−**H** one-way ANOVA was done with Dunnett's post hoc test to correct for multiple comparisons. Source data are provided as a Source Data file.

increase favorable pharmacodynamic traits. Pre-clinical studies have shown that subclass-switching to IgG3 enhances functional output in vitro for various pathogens[24–26,37,43–45]. Despite the promise of the IgG3 subclass, its use has been hindered due to a lower half-life in serum[1] and a tendency to aggregate when produced on a large scale[46]. Fc-engineering has remedied these issues[25,46], making this subclass more appealing for clinical application.

In this work, we generated and characterized all four human subclasses of Ab25, in which we observed an unexpected phenotype in the IgG3 subclass. It exhibited reduced binding but improved function. We further investigated this reduced reactivity of IgG3 by looking at the network of H-bonds and salt bridges of the M1-IgG interaction in silico. We postulate that an exchange of constant domains from IgG1 to that of IgG3 results in an induced rearrangement of the hydrogen bond network between the Fab domains and the M1 protein. Our results extend previous studies[4–7] by providing a more detailed analysis of atomic interactions that could explain the loss of observed binding. Inserting mutations in the M1 protein and Ab25's variable domain could elucidate the molecular interaction further. A future study is warranted, primarily focusing on this issue but with more mAbs against various antigens where altering the constant domain has been seen to influence the antigen binding capability[4–7,37].

The enhanced performance of the IgGh$_{47}$ (and the IgGh$_{32}$ and IgGh$_{62}$ versions in some regard) compared to IgG1 shows the functional potency of the IgG3 hinge region. Similar work has been done recently with antibodies against adenovirus and HIV[26,47]. Both studies hypothesized that the functional effect associated with IgG3's hinge is due to increased flexibility but did not pursue this hypothesis. We build on those studies and, using in silico analysis, reveal that our IgG3 might possess an increased flexibility of Fc relative to the M1 protein due to its extended hinge region. In a previous study[8], hinge flexibility, measured as SD of Fab-Fc angle (angle 3 in our data), was reported to be ±36° for IgG3 and ±30° for IgG1, with minor differences between the two and no difference in the total Fab-Fc angle span. Notably, these experiments[8] were performed without the antigen and only entailed the inherent Fc flexibility relative to the antibody itself (Fc contra Fabs). Our investigations replicate the findings that IgG3 has a slightly larger Fab-Fc angle span than IgG1. However, the Fab-Fc angle span for IgG3 also exceeds that of IgGh$_{62}$, which instead speaks against the fact that the IgG3 hinge would lead to a higher Fab-Fc angle. More importantly, our MD simulations suggest much greater flexibility regarding absolute Fc movement, spatial movement in 3D space, and Fc-antigen angle span for the IgG3 and IgGh$_{62}$ antibodies, translating more into functional flexibility.

We provide in vivo pharmacodynamics of hinge engineering, complementing previous pharmacokinetics (biostability in serum) studies[26]. Our work shows that utilizing IgG3's flexible hinge has direct in vivo implications. IgGh$_{47}$ protects against systemic bacterial dissemination, while the more rigid IgG1 does not. It is worth noting that IgG1 Ab25 showed a more prominent in vivo effect in a diffuse infection

model[11]. In contrast, in the two-stage model used here, IgG1 did not show the same level of protection. The main difference between the models lies in the route of administration that generates either a diffuse (scruff) or restricted local (flank) infection that progresses to an invasive infection. The route of administration can affect the host-pathogen interactions underlying invasion and disease progression. This intricate interaction has been recently elucidated for the two-stage flank model, showing that bacterial infection leads to enzymatic modifications of human IgGs in the form of IVIG when given as therapy[29]. Thus, we hypothesize that IgG1 was more affected by this effect in the current model than in the previous one[11]. Analysis of the monoclonal IgG from the animal plasma, with higher levels of degraded IgG1 and IgG3 mAbs detected, supports this hypothesis. Regarding the efficacy of IgGh$_{47}$ compared to IgG3, half-life might be an important factor – where IgG3 is known to have a lower half-life in humans and mice due to lower affinity to neonatal FcR[48], which was confirmed by our SPR analysis. However, the LC-MS/MS data from plasma showed that this was not an issue during the current experiments' short timeframe, where similar IgG levels were found in mouse plasma after 24 hours. This is most likely due to the short time frame for the experiment. Furthermore, other factors such as Fab-interactions with the FcRn in vivo[49], clone-specific differences[31] and changes to antibody FcR-affinity upon antigen-binding have all been shown to influence FcR affinity of mAbs. These factors further complicate the interpretation between the relationship of affinity determination and in vitro and in vivo outcome.

For a better understanding of how Ab25 IgGh$_{47}$ has a higher protective ability with respect to both its parents, one would require mouse FcR-affinity data for Ab25, which we were unable to obtain due to Ab25 non-specifically interacting with the chip. The absence of such data is a limitation of the study. Since clone-specific differences can exist[31], it is difficult to estimate what role different affinities to mouse FcR played on the in vivo outcome. Human IgG1 and IgG3 Fc bind with similar relative affinity to murine FcR, reflecting their human counterparts[50], but mouse phagocytes have different expressions of FcR[51]. It is, therefore, difficult to fully compare the in vitro and ex vivo data with the murine model experiment. Furthermore, despite human IgG3 having high-affinity to murine FcRs shown by previous work[50], human IgG3 antibodies have been shown to induce weaker effector responses by mouse phagocytes[52,53]. These instances are contrary to work with human phagocytes, where IgG3 is more potent in inducing effector functions[24–26,37]. The in vivo data so far serves as a proof of concept for hinge-engineering, and future work will have to build on these findings to enable translation to a more human-like setting, with humanized mouse models with other mAb clones against other pathogens such as SARS-CoV-2 for instance.

In this work, we aimed to increase the generalizability of our IgGh$_{47}$ subclass construct. Firstly, we observed that the potent effect against the lab strain SF370, which expresses *emm*1, was generalizable to five other *emm* types tested. Notably, these *emm* types belong to

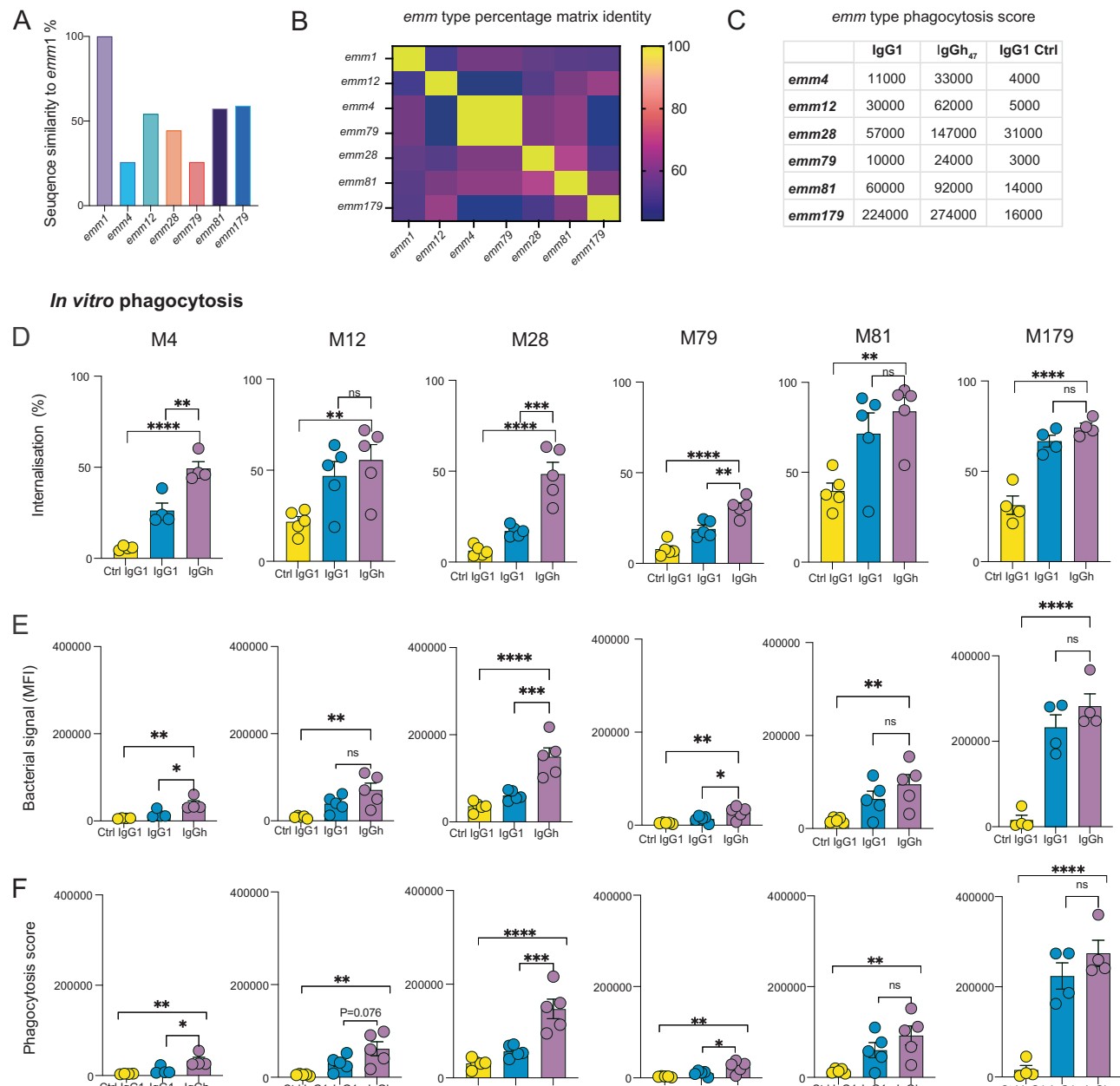

**Fig. 7 | IgGh₄₇ potent opsonic phenotype is generalizable to other *emm* types.**
**A** Percentage of similarity of different *emm* types to *emm*1 covering Ab25's epitope.
**B** Percentage identity matrix of all the *emm* types used for the sequence covering
Ab25's dual-span epitope. **C** Table summarizing the mean value of phagocytosis
score for each mAb for the different *emm* types. **D** Percentage of THP1- cells with
internalized bacteria across the different treatments for different *emm* types.
**E** Bacterial signal detected for the whole THP1 cell population for the different *emm*
types. The F Phagocytosis score (mean value sumarized in C) for the treatments
across the *emm* types. For all *emm* types, MOP 50 was used in the experiments,
except for *emm*179 (MOP 25) and *emm*4 (MOP 100). Statistical comparison was
done by comparing all treatments to Ab25 IgGh₄₇ with one-way ANOVA with mul-
tiple comparisons and post hoc Dunnett's test to correct for multiple comparisons.
**** denotes $P$-value < 0.0001, *** denotes $P$-value < 0.001, ** denotes $P$-value < 0.01 *
denotes $P$ < 0.05 and $P$-value > 0.05 is ns. The mean is shown and the error
bars are SEM throughout the figure. In **D**–**F**, for *emm*4 and *emm*179, $N$ = 4 inde-
pendent experiments were performed, while for *emm*12, 28, 79, and 81, $N$ = 5
independent experiments were performed. Source data are provided as a
Source Data file.

distant *emm* clades, and the strains used were clinical isolates from
infected patients. Thus, our findings, combined with the in vivo and
in vitro data against *emm*1, suggest that the IgGh₄₇ construct in Ab25 is
a promising pre-clinical candidate to consider for further develop-
ment. In addition, Ab25's variable domain does not react with human
tissues as previously determined by a tissue microarray[11]. Finally, we
further increased the generalizability of our construct by engineering
three anti-SARS-CoV-2 mAbs into IgGh₄₇ versions from the parent IgG1.
All three IgGh₄₇ mAbs exhibited enhanced function compared to the
original IgG1's. Interestingly, the difference in efficacy suggests that

although it is a generalizable effect, depending on clones, the effect
can vary. In that regard, Ab36 IgGh₄₇ is an exciting candidate to test
in vivo since it was comparable to our double-IgG3 cocktail, which is
based on the two most potent opsonins we have generated by our lab
against SARS-CoV-2[37]. Given that Fc-mediated functions are increas-
ingly more important against mutated SARS-CoV-2 variants[33–38],
increasing mAb efficacy through hinge engineering shows promise in
generating a protective mAb against this pathogen that is not depen-
dent on neutralization. In conclusion, our work suggests that future
mAb development would benefit from using the engineered IgGh₄₇

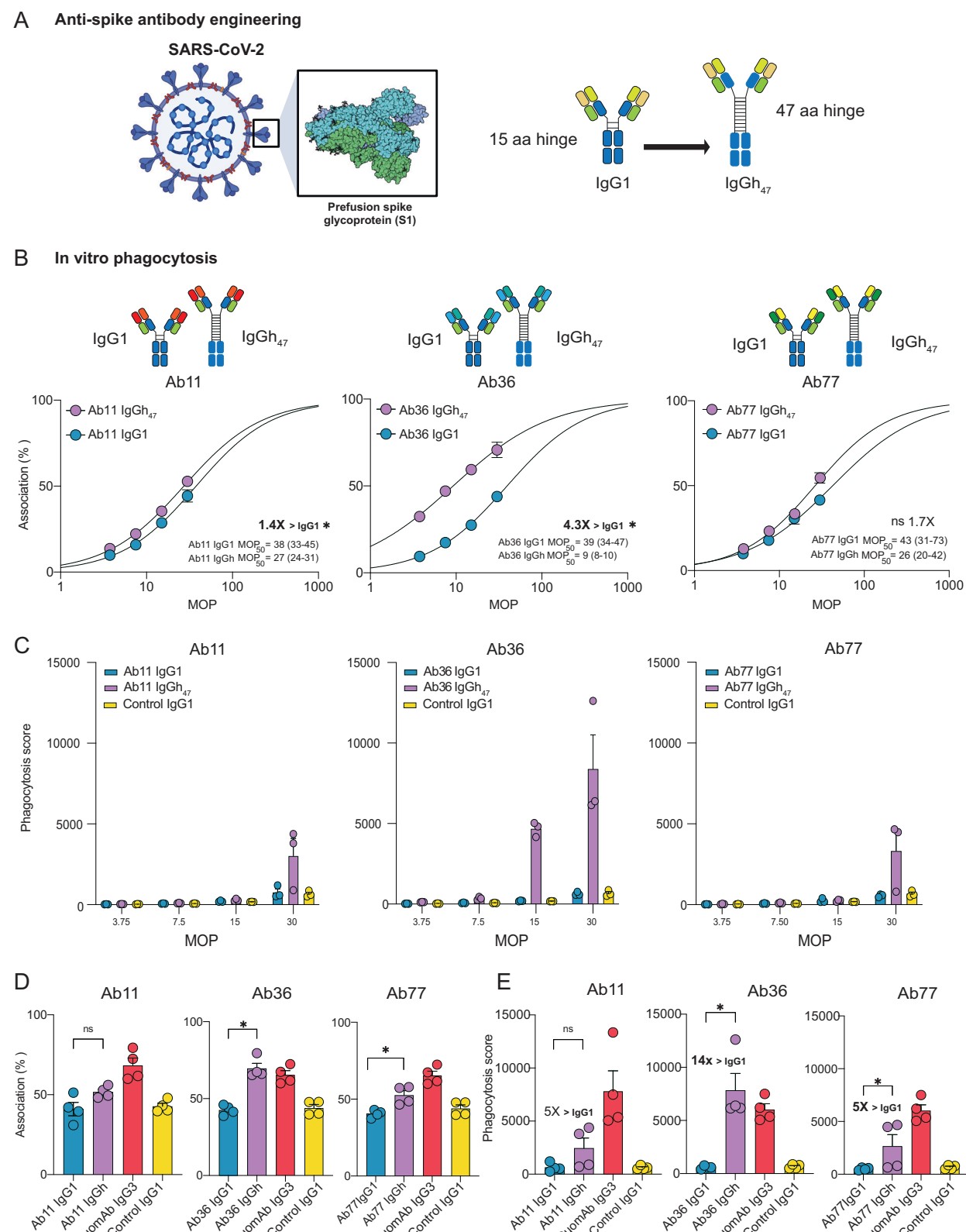

**Fig. 8 | SARS-CoV-2 mAbs engineered from IgG1 to IgGh47 subclass exhibit a potent opsonic function. A** Illustration depicting a SARS-CoV-2 virion with the spike protein trimer antigen and the three different clones Ab11, Ab36 and Ab77 which were made into IgGh47 from an original IgG1. Schematic made using Biorender. **B** MOP curves with MOP50 and 95% CI are in brackets for the different mAbs, with statistically significant differences (non-overlapping 95%CI) between the two mAbs are highlighted (*). **C** Phagocytosis score across the different MOPs. **D, E** Depicts association and phagocytosis score at MOP 30. Statistical comparison

was made by comparing the IgG1 with that of IgGh47 version with Mann-Whitney U two-tailed test. Across the figure mean is shown and error bars are SEM.*** denotes *P*-value < 0.001, ** denotes *P*-value < 0.01 * denotes *P* < 0.05 and *P*-value > 0.05 is ns. Positive control is DuomAb IgG3, and negative control is Xolair IgG1. In B-C *N* = 3 independent experiments were performed while at D and E *N* = 4 independent experiments were performed. Source data is provided as a Source Data file.

subclass, where this study has shown its effectiveness in two biologically distant systems with streptococci and SARS-CoV-2.

# Methods

## Ethics statements

All animal use and procedures were approved by the local Malmö/Lund Institutional Animal Care and Use Committee, ethical permit number 03681-2019. For ex vivo experiments where human neutrophils and monocytes were isolated, ethical approval was granted by the regional Swedish ethical review authority in Lund (Etikprövningsnämnden Lund, 2015/01801) and donors gave written and oral consent.

## Plasmid generation, transformation, and plasmid purification

To generate heavy chain plasmids of the different subclasses, pVITRO1-Trastuzumab-IgG2 (Addgene plasmid #61884; RRID: Addgene_61884), IgG3 (Addgene plasmid #61885; RRID:Addgene_61885), and IgG4 (Addgene plasmid #61887; RRID:Addgene_61887) was a gift from Andrew Beavils lab[54]. The heavy chain and light chain of Ab25 were generated in Bahnan et al.[11]. The cloning was done by amplifying the constant region of the heavy chains for IgG2, 3, and 4 by using the high-fidelity Phusion (NEB) 3-step PCR protocol. The original heavy chain of Ab25 was linearized and so that the constant region of IgG1 was removed. The primer design was planned so the amplified vector and insert contain overlapping ends, which are compatible with HIFI DNA assembly (NEB Biolabs). For the generation of the IgGh$_{47}$, a 2-step PCR was done to amplify the insert and the vector, and the fragments were combined by HIFI DNA assembly. Competent 5-alpha *E. coli* (NEB) were transformed according to instructions. The generation of IgGh$_{62}$ was done in a similar manner, but with a 3-step phusion protocol. The truncated hinge versions (IgGh$_{17}$ and IgGh$_{32}$) were generated in a completely different method. Two gene fragments were generated (Eurofins Genomics), both having the IgG1´s CH1 region but different lengths of the IgG3 hinge. Overhangs complementary to the linearized vector were added to the fragments by Phusion 3-step PCR. The IgGh$_{47}$ vector was linearized as described above. After successful amplification, linearized vector and insert were joined using HiFi DNA Assembly. Resulting plasmids were sequenced (Eurofins Genomics) to verify a correct insertion. The anti-Spike IgGh$^{47}$ variants were generated by amplifying the VH domain from previously discovered antibodies[38] creating inserts with overhangs. The IgGh$_{47}$ plasmid was linearized by Phusion 3-step PCR and merging was done using HiFi DNA Assembly thereby replacing the VH of Ab25 with anti-spike VH.

Transformed bacteria were grown as described by the manufacturer. Plasmids were generated using QIAGEN Spin Miniprep or QIAGEN plus Midi kit according to the manufacturer's protocol. Plasmid concentration was measured using spectrophotometry (DeNovix). Plasmid integrity was controlled by sequencing.

## Bacteria culturing

Bacteria culturing for live binding assays and heat-killing was done as in Bahnan et al.[11]. In short, SF370 bacteria with GFP were streaked on Todd-Hewitt Yeast (THY, fisher scientific 11708962) agar plates with 1:1000 erythromycin added, and for WT SF370 no antibiotics were added. SF370 bacteria were grown to log phase in 10 mL THY media with or without antibiotics (1:1000 Erythromycin, Sigma E5389-5G) overnight depending on which strain was used. After the overnight bacteria was diluted 1:20 in 10 mL volume for binding assays or 50 mL for preparation for heat-killing. Bacteria were grown for 2 hours and 30 minutes to reach mid-log phase. For heat-killing, bacteria were concentrated from 50 mL into 1 mL PBS and were put on ice for 15 minutes and subsequently put on a shaking heat-block at 80 degrees for 5 minutes followed by ice for 15 minutes. Heat-killed bacteria were stained with Oregon green 488 x succinimidyl ester (2 μL from stock in

1 mL PBS, 5 μM, Fisher Scientific 11549186). CypHer5E (Fisher Scientific, 11505834) was used to stain heat-killed bacteria as an internalization marker in a 1:40 dilution from a 1 mg/mL working stock in a volume of 200 microL containing sodium carbonate buffer (0.1 M, pH 8.3).

## Cell culture, antibody production, and antibody purification

THP-1 monocytic cells (Sigma Aldrich, 88081201-1VL) were cultured in RPMI with 10% FBS and L-glutamine and were kept at $2.5 \cdot 10 \times 10^5$ cells per mL. HEK293 cells (Sigma Aldrich, 12022001-1VL) and Expi293F (Thermofisher, A14527) cells were used to produce antibodies similarly as done before[37,38]. Supernatant containing produced antibodies was purified through incubation with protein G sepharose 4 fast flow (Cytiva, 17-0618-05) according to the manufacturer's instruction.

## Binding assays with secondary conjugated antibodies

For binding assays, live *S. pyogenes* strain SF370 transformed with GFP was used. 10 mL of mid-log culture was concentrated and resuspended in a final volume of 1 mL in PBS. The bacteria were sonicated for $2 \times 2$ minutes to break clumps. 10 μL of concentrated bacteria were added to each well of a 96-well low-binding plate containing 90 μL of anti-M antibodies (concentrations of antibodies varying between 20-0.65 μg/mL) in PBS. Serial dilutions of antibodies were prepared with concentrations ranging from 20 μg/mL to 0.65 μg/mL. The bacteria were opsonized for 30 min at 37 °C with gentle shaking. After opsonization, the bacteria were washed twice with PBS before adding 50 μL (1:500 dilution from stock) secondary antibody, Alexa Fluor® 647 AffiniPure F(ab')$_2$ Fragment Goat Anti-Human IgG, F(ab')$_2$ fragment specific, (109-606-097, Jackson Immunoresearch). The secondary antibody was incubated for 30 minutes at 37 °C with gentle shaking. Following this, PBS was added so that the final volume reached 150 μL. The plates were analyzed by using a CytoFLEX flow cytometer (Beckman Coulter). The data acquired from the flow cytometer experiments described above were processed on Flowjo software for gating strategy. The positive gate for the live GFP-expressing bacteria was set by gating for size (SSC) and GFP fluorescence (FITC) (Supplementary Fig. 1A). The gate for antibody reactivity to M was set by using a low concentration of Xolair (0.5 μg/mL) for Fc-background binding (Supplementary Fig. 1A). Antibody reactivity was determined by gating on percentage bound bacteria positive for the secondary antibody (APC) (Supplementary Fig. 1A). The data were analyzed in GraphPad Prism. The percentage of parents was plotted against antibody concentration followed by a non-linear regression analysis with B$_{max}$ constrained to 100% and K$_D$ unconstrained.

## Directly-conjugated antibody binding assay

The antibodies were conjugated with Alexa Fluor 647 using the GlyClick kit (Genovis) according to the manufacturer's protocol. The binding assay was performed with live SF370 expressing GFP as described above with some small modifications. The concentrations of antibodies varied from 10-0.6 μg/mL for IgG1 and IgGh$_{47}$. The positive gate for the live GFP-expressing bacteria was set by gating for granularity (SSC) and GFP fluorescence (FITC) (Supplementary Fig. 10A). Antibody reactivity was determined by gating on percentage bound bacteria positive for APC-A (Supplementary Fig. 10A). The data were analyzed in GraphPad Prism, and the percentage of bacteria positive for antibody was plotted on the Y-axis with antibody concentration on the X-axis. A non-linear regression analysis was performed, with B$_{max}$ constraint set to 100% while Hill-slope and K$_D$ were unconstrained.

## M protein sequence alignment

M protein sequences from the clinical isolates were sequenced in Bahnan et al.[11]. The partial M protein sequences, covering Ab25's epitope, were aligned and analyzed by Clustal Omega (1.2.4). A heat-map of M protein percentage similarity was performed by using the analysis from Clustal Omega by plotting the results in GraphPad Prism.

## Phagocytosis assay

Phagocytosis assays with THP-1 cells were done as in bahnan et al.[11] but with some modifications. Serial dilutions of bacteria were done with MOP (bacteria per phagocyte) as the variable, and each well was opsonized with 15 µg/mL of anti-M antibodies or Fc-control for 45 minutes at 37 °C gently shaking. 100 µL was the end volume before the addition of the cells. While the plate was opsonizing, the THP-1 cells were collected and washed once to remove RPMI and exchanged to sodium media. THP-1 cells, numbering $1 \times 10^5$ cells, were added at a volume of 50 µL (concentration of $2 \times 10^6$ per mL) after being on ice for 5 minutes. The cells were left phagocytosing for 30 minutes at 37 °C while gently shaking. The plate was put on ice for 20 minutes to stop further phagocytosis. The plate was analyzed in the CytoFLEX flow cytometer (Beckman Coulter). The acquisition was set to capture 5000 THP-1 cells with medium velocity (30 µL/min). While running, the 96-well plate was kept on an ice-cold insert. THP-1 cells were gated first by size (FSC) and granularity (SSC) and then for single cells by SSC-A and SSC-H. By using untreated THP-1 cells, we set a gate for non-associated THP-1 cells, associated with bacteria and associated with bacteria and internalized by using the FITC and APC signal (Supplementary Fig. 1B). The data acquired from the flow cytometer experiments described above were processed on Flowjo software for gating strategy and later analyzed on Graphpad Prism. The $MOP_{50}$ analysis was calculated by using the PAN-assay model as described in Prism[23], which in short, is a non-linear regression model where $EC_{50}$ of MOP is calculated based on the persistent association (Bacteria+ cells, stained with Oregon Green marker) with a top value set to 100% and bottom value constrained to 0%. Finally, the phagocytosis score was calculated by multiplying the percentage of bacteria-positive THP-1 cells (associating) with the bacterial signal (MFI, Oregon Green) for the whole population of THP-1 cells. Phagocytosis score was used as the primary metric to determine opsonic function for phagocytosis.

For assessment of the SARS-CoV-2 mAbs similar experiments were performed with the THP-1 cells but with some differences. The MOP used was 30-15-7.5-3.75 and the prey was spike-coated microsphere beads. Spike-conjugated beads were produced by biotinylating 25 ug spike protein trimer (produced by transfecting Expi293 cells with 20 ug of Spike CS/PP plasmid donated by Florian Krammer's lab) using EZ-Link™ Micro Sulfo-NHS-LC-Biotinylation Kit (Thermofischer; Cat: 21935) and conjugating with 70 µL from stock of streptavidin-coated microsphere beads (1 µm, Bangs Laboratories, Cat: CFR004) according to the bead manufacturers instruction[38]. Spike-beads were opsonized at a concentration of 2.5 µg/mL at a volume of 100 µL for 30 minutes at 37 degrees on a shaking heat-block. THP-1 cells were gated for APC fluorescence (bead signal) to assess association (Supplementary Fig. 12). The remaining analysis for $MOP_{50}$ and phagocytosis score was done similarly as for with the bacteria.

THP-1 cells FcR expression was determined by staining $1 \times 10^6$ live cells for 20 minutes on ice suspended in sodium media at a volume of 100 µL. 5 µL of Anti-CD16 mAb conjugated with eFluor™ 450 was used from stock to stain the cells (#48-4714-82, Thermo fisher) for CD16 expression. While for CD32 20 µL of Anti-CD32 mAb conjugated with APC (#559769, BD biosciences) from stock was used to stain the cells for CD32. This specific anti-CD32 mAb recognizes all CD32 variants (FcγRIIA-C). Finally, for CD64 staining, 10 µL of anti-CD64 mAb conjugated with FITC (# MA5-16434, Thermo fisher) from stock was used to stain the cells for CD64. These stock had been prediluted by the manufacturer and we used the recommended volumes.

Monocytes were isolated from the blood of healthy human donors first by acquiring a PBMC layer by using PolymorphPrep (Abbot). Monocytes were then further purified from the PBMC layer through positive selection using CD14 Microbeads (Cat#130-050-201, Miltenyi Biotec) according to the manufacturer's instructions. Following isolation, monocytes were then counted using an XN-350 hematology analyser (Sysmex). Neutrophils were, in turn, isolated with PolymorphPrep gradient (Abbot) according to the manufacturer's instructions and were counted by using an XN-350 hematology analyser (Sysmex). Isolated cells were kept on ice for 30 minutes after being resuspended in sodium media and adjusted to $2 \times 10^6$ cells/mL. Donors had given written and oral consent to participate in this study. They were provided oral information on the purpose of the donation, which was used only to isolate and use the monocytes and neutrophils in flow and microscopy experiments.

For the monocytes experiments, 100,000 cells (50 µL volume) were added to 2.5 million (MOP 25) of opsonized heat killed SF370 stained and left to phagocytose for 30 minutes in a 96-well plate. After this, the cells were left on ice and stained with an anti-CD14 marker (Brilliant Violet 421™ anti-human CD14 Antibody, Biolegend) in a volume of 20 µL diluted from the stock 1:50, 3 µg/mL final concentration, for 20 minutes. The 96-well plate was then analyzed on a Beckman Coulter CytoFLEX flow cytometer. Monocytes were gated for size and granularity (Supplementary Fig. 3), then a single cell gate was drawn by SSC-H and SSC-A. From this population, using unstained cells, a CD14+ gate was drawn (Supplementary Fig. 3). Using a well with cells only stained for CD14 (without bacteria) a gate for association and internalization was made (Supplementary Fig. 2). Phagocytosis score was calculated by multiplying the MFI of the FITC+ population with the percentage of FITC+ cells divided by 100.

For the neutrophil experiments, 100,000 cells (in a volume of 50 µL), were added similarly to 2.5 million (MOP 25) opsonized heat killed SF370. After 30 minutes, the cells were kept on ice for 20 minutes. Before being put on ice, the cells were stained with 20 µL of an anti-CD18 marker (BV421 Mouse anti-Human CD18, BD Biosciences) diluted 1:100 to a final concentration of 2 µg/mL. After staining, the cells were analyzed on a Beckman Coulter CytoFLEX flow cytometer. The neutrophils were gated for size and granularity (Supplementary Fig. 3), then a single cell population was selected for by SSC-H and SSC-A. Using unstained cells, a CD18+ gate was drawn. Using stained cells without bacteria, an association and internalization gate was made. Phagocytosis score was calculated by multiplying the MFI of the FITC+ population (associating cells) with the percentage of FITC+ cells (associating, Oregon Green stained bacteria) divided by 100.

## Animal model

The background strain and the gender of the mouse profoundly affects susceptibility to *S. pyogenes* infection[55]. In order to use both female and male mice two different infectious doses would be required, therefore we focused on comparing the same infectious dose in female mice only. 40 nine-week-old female C57BL/6 J mice (Scanbur/Charles River Laboratories) were used. Monoclonal antibodies (0.4 mg/mouse) were administered intraperitoneally 6 h pre-infection. The pretreatment groups were coded, and the experimenters were blinded to which group had which intervention. Unblinding was done after the data had been analyzed and compiled. *S. pyogenes* AP1 was grown to an exponential phase in Todd–Hewitt broth (37 °C, 5% $CO_2$). Bacteria were washed and resuspended in sterile PBS. $2.5 \times 10^5$ CFU of bacteria were injected subcutaneously on the right flank, leading to a local infection that progressed to systemic infection within 24 h. Mice were sacrificed 24 h post-infection, and organs (blood, livers, spleens, and kidneys) were harvested to determine the degree of bacterial dissemination. Cytokines in plasma were quantified using a cytometric bead assay (CBA mouse inflammation kit, BD Biosciences) according to manufacturer instructions. The blood from the mice was analyzed by Western blots to control for human IgG integrity at the end of the experiment. Kaplan-Meier curves analyzing adverse outcomes in the form of bacterial dissemination to organs were done with Prism Graphpad, and a Log-rank (Mantel-Cox) test for statistical analysis was performed to determine statistical significance between the two groups.

## Structural models

To provide structural models, the system was divided into three parts: Fc domain including the hinge region, the M1 fragment-Fab1 interactions, and the M1 fragment-Fab2 interactions. As depicted in Supplementary Fig. 13, the Fc and Fab domains of IgG1 and IgG3 were de novo modeled separately, using AlphaFold2[56,57], considering the homo-oligomer state of 1:1. The multiple sequence alignments were generated by MMseqs2[58]. The high pLDDT values showed AF2 models are highly confident. For IgG1, the ranked 1 model, and for IgG3, the model with elongated hinge region was selected. These output models were then relaxed using Rosetta relax protocol[59], where the side chains and disulfide bridges were adjusted. The M1-IgG models were generated using a targeted cross-linking mass spectrometry (TX-MS) approach[60], using X-ray structures for Fab domains (pdb id: 8C67) shown in Supplementary Fig. 14 and details in Supplementary Table 1. The cross-link constraints (XLs) were derived from a set of experiments generated in our previous study[11], and the threshold of 32 Å was applied to map XLs on the structure. Accordingly, two epitopes of the M1 protein in the B and C1 domains interact with the Fab2, and Fab1 domains of IgGs, respectively. These two M1 fragment-Fab interactions for both IgGs were separately characterized (TX-MS) and re-adjusted using Rosetta Relax protocol. The loops connecting the hinge region of both IgGs to Fc and Fab domains were then re-modeled and characterized using the DaReUS-Loop web server[61,62]. Finally, the full-length structure of both IgGs was relaxed, and all disulfide bridges (specifically in the hinge region) were adjusted using Rosetta relax protocol. Concerning the M protein, the M protein is a long dimeric coiled coil embedded in the membrane from one side. Also, it has been shown that the surface of bacteria is covered with a dense layer of M proteins forming PPIs with other proteins. However, the limits of computational resources would not allow us to perform MD simulations of the full-length M protein embedded in the membrane. Therefore, we had to consider only the portion of M protein that interacts with the antibodies (taking into account the experimental information and data from cross-linking mass spectrometry). Since this is not the ideal condition, the M1 protein, without any restraints, could be highly flexible. To avoid such artifacts, we decided to impose restraints as to provide a condition closer to the native one.

## SPR assay for FcR affinity measurements

Surface plasmon resonance (SPR) experiments were performed using a Biacore 1 K+ instrument (Cytiva, Uppsala, Sweden) to determine the binding profiles of antibodies with different constant domains to human and mouse FcR, with a particular focus on antibodies carrying the IgGh$_{47}$ constant domains. The running buffer for studies involving the FcRn receptor was PBS (Dulbecco's PBS, GIBCO, Fisher Scientific, Waltham, Massachusetts) adjusted to pH 6.2 and supplemented, with 0.05% Tween 20. For the remaining receptors, the running buffer consisted of PBS (Thermo Fisher Scientific, Waltham, Massachusetts, USA) supplemented with 0.05% Tween.

Analysis was performed utilizing a Series S Sensor Chip CAP (Cytiva, Uppsala, Sweden). Oligonucleotide-labelled streptavidin (Biotin CAPture reagent; Cytiva) was caught onto the chip surface (contact time: 120 s; flow rate: 2 μl/min). Biotinylated Fc receptors were then injected over the surface (contact time: 120 s; flow rate: 10 μl/min) to achieve 70-500 RU immobilization. Subsequently, IgG was injected over the flow cells at five concentrations (1.6-400 nM; contact time: 120 s; flow rate: 30 μl/min) followed by dissociation during 320 s. Complete regeneration of the surface was achieved between each cycle by injection of the regeneration solution (6 M guanidine-HCl, 0.25 M NaOH; Cytiva). All steps were performed at a temperature of 25 °C. Running buffer injections were used for background subtraction. Subsequently, all data were fitted using a 1:1 binding model in the Biacore 1 K+ Evaluation Software. These assays were performed with a set of human antibodies specific for the receptor binding domain of

SARS-CoV-2, as variants of Ab25 showed very substantial interaction with the chip surface at the conditions used to measure antibody binding to FcRn.

## Molecular dynamics simulations

We performed three replicates of 1 μs MD simulations starting from the models obtained for M1-IgG1, M1-IgG3, and M1-IgGh$_{62}$ systems. This resulted in a total of 9 μs MD simulations. In one replicate of the M1-IgGh$_{62}$ system, we observed large conformational changes in the IgGh with respect to the M1. In this replicate, the Fab domains were detached, resulting in a global shift of the antibody and, finally, attachment of the Fc domain to the M1. Therefore, this replicate was discarded from the analysis. The 3D coordinates of the two models generated for M1-IgG1 and M1-IgG3, described in the previous section, were retrieved. Both IgG1 and IgG3 have 6 chains (A-F); four disulfide bonds are present on the Fc domain, and four on each Fab domain. However, the IgG1 has a shorter hinge region of 14 residues (E222-P236) with 2 disulfide bonds, while the hinge region of the IgG3 is composed of 61 residues (E222-P283) and 11 disulfide bonds. The M1 protein has 2 chains (M, N), and for the MD simulations, we considered its central region (residues A149-L291 on chains M and N). MD simulations were carried out with the GROMACS 2020.6[63] using the CHARMM36m force field parameter set[59]: (i) Na$^+$ and Cl$^-$ counter-ions were added to reproduce physiological salt concentration (150 mM solution of sodium chloride), (ii) the solute was hydrated with a tri-clinic box of explicit TIP3P water molecules with a buffering distance of up to 12 Å, and (iii) the environment of the histidine was checked using MolProbity[64]. First, we performed 5000 steps of minimization using the steepest descent method keeping only protein backbone atoms fixed to allow protein side chains to relax. After that, the system was equilibrated for 300 ps at constant volume (NVT) and for a further 1 ns using a Langevin piston (NPT)[65] to maintain the pressure while the restraints were gradually released. For every system, three replicates of 1 μs, with different initial velocities, were performed in the NPT ensemble. We applied positional restraints with the force constant of 10 kcal/mol/Å$^2$ on the heavy atoms of the M1 protein during the production runs to avoid drastic rearrangements. The temperature was kept at 310 K and pressure at 1 bar using the Parrinello-Rahman barostat with an integration time step of 2.0 fs. The Particle Mesh Ewald method (PME)[66] was employed to treat long-range electrostatics, and the coordinates of the system were written every 100 ps. Typically, the M1-IgG1 and M1-IgG3 systems are composed of ~481,152 and ~1,230,941 atoms in a triclinic water box with a volume of ~4925 and ~12,567 nm3, respectively. The root mean square deviations (RMSD) of the studied complexes from the equilibrated structure were measured on the Cα atoms along simulation time for all the replicates (Fig. 2). All systems were fully relaxed after 100 ns. Then, we calculated the residue root mean square fluctuations (RMSF) over the last 900 ns of every simulation (Fig. 2) with respect to the average conformation and over the Cα atoms. For the center-mass calculations: The "gmx trj" command of GROMACS was used to extract coordinates of the Fc domain and M1 protein using the -com option for center of mass.

**Hydrogen bonds and salt bridges.** The hydrogen bonds (H-bonds) were detected using the HBPLUS algorithm[67]. H-bonds are detected between donor (D) and acceptor (A) atoms that satisfy the following geometric criteria: (i) maximum distances of 3.9 Å for D-A and 2.5 Å for H-A, (ii) minimum value of 90° for D-H-A, H-AAA and D-A-AA angles, where AA is the acceptor antecedent. For a given H-bond between residues i and j, interaction strength is computed as the percentage of conformations in which the H-bond is formed between any atoms of the same pair of residues (i and j). Moreover, the salt bridge plugin of the VMD was used to detect all salt bridges at the interface between the Fab domains and the M1 protein[68]. The stability of salt bridges was recorded as the percentage of conformations in which the distance

between the center of mass of the oxygens in the acidic side chain and the center of mass of the nitrogens in the basic side chain is within the cut-off distance of 3.5 Å.

**Angles.** We measured the angle formed between 1) the Fc domain and the M1 protein, 2) Fc and Fab domains, and 3) Fab1 and Fab2 domains. The vectors used to calculate each angle are defined here:

- *between the Fc domain and the M1 protein*: i) the center of mass of the Fc domain and center of mass of the M1 protein, ii) the center of mass and the N-terminal region of the M1 protein.
- *between the Fc domain and Fab domains*: i) the center of mass of the Fc domain and center of mass of the hinge region, ii) the center of mass of the Fab domain and center of mass of the hinge region.
- *between the Fab1 and Fab2 domains*: the center of mass of the hinge region and center of mass of each Fab domain.
- *Fab2 domain*: the center of mass of the hinge region and center of mass of the Fab2.

The angles were then defined based on the lines that best fit these segments. The results are reported in two scenarios: for all trajectories (Supplementary Fig. 5A–C) and in dual-Fab binding conformations where the distance between residues at the interface of Fab1 and the M1 protein is less than 30 Å (Supplementary Fig. 5D–F). The second scenario resulted in 20,074 angles for M1-IgG1, 10,871 angles for M1-IgG3, and 11,306 angles for M1-IgGh$_{62}$ simulations.

## Clustering analysis

To characterize representative conformations, we performed a cluster analysis of the MD trajectories using backbone RMSD as the similarity metrics by the GROMOS clustering approach[68] with a cutoff equal to 0.7 nm. The trajectories of replicates were first concatenated, clustering was then performed, and finally, the population of each cluster was calculated. The same analysis was performed only considering the replicates where the dual-Fab binding was observed (replicates 1 and 3 for M1-IgG1, replicate 3 for M1-IgG3, and replicate 1 for IgGh$_{62}$ simulations). Clusters with a population of at least 3000 conformations were colored in shades of green on the plot (Supplementary Fig. 6A). The results are reported for all MD trajectories of M1-IgG1, M1-IgG3 and IgGh$_{62}$ systems (30000, 30000 and 20000 conformations, respectively) and only for dual-Fab binding conformations (20000, 10000 and 10000 conformations, respectively). The three highly populated clusters of IgG1 consist of 9071, 4852, and 3239 conformations. And the top three highly populated clusters of IgG3 consist of 5363, 4195, and 4031 conformations. The representative conformations of the highly populated clusters are shown when dual-Fab binding was observed (Supplementary Fig. 6B–E).

## Binding free energy and alanine scanning

The AnalyseComplex in FoldX version 4[69] was used to estimate the binding free energy between the Fab1 and M1 protein, and between the Fab2 and M1 protein (Supplementary Fig. 7). The AlaScan in FoldX version 4 was used to identify relevant residues involved in the interactions between the IgGs and M1 protein. For each system we identified Fab residues that are at the interface with M1 protein, as those representing an average distance of at most 5 Å with the closest M1 residue over the course of the MD simulations. Then we reported the ΔΔG values for interface Fab residues only for the replicates when the average distance is less than or equal to 5 Å. The results of alanine scanning are reported for the interface residues of Fab1 (Supplementary Fig. 8) and Fab2 (Supplementary Fig. 9). The values in blue at the top of each box show number of replicates from which the ΔΔG values were reported (i.e. number of replicates in which the average distance between the corresponding Fab residue and the M1 protein is at most 5 Å).

## Crystallization data analysis

The Ab25 Fab sample was concentrated to 6.75 mg/mL in the buffer of 20 mM HEPES(2-[4-(2-hydroxyethyl)piperazin-1-yl]ethane-1-sulfonic acid), 150 mM NaCl, pH 7.0. A crystallization experiment set up against commercially available screens was performed with a Mosquito crystallization robot (SPT Labtech) using sitting drops with a total volume of 300 nL and different ratios between the protein sample and reservoir solution. Optimization of initial hits gave crystals appearing within a week in 27.50% (w/v) PEG8000, 0.1 M CHES pH 8.5. After growing to full size (50 x 100 x 250 μm), the crystals were harvested, cryoprotected in reservoir solution supplemented with 20 % (v/v) glycerol, and flash frozen in liquid nitrogen. Testing for diffraction and data collection was performed at the BioMAX beamline at the MAX IV Laboratory (Lund, Sweden). Diffraction data were collected by fine-slicing with an oscillation range of 0.1°, 360° total. The data were auto-processed with the software EDNA[70] to a resolution of 1.88 Å, and the structure was solved by molecular replacement with PHASER of the PHENIX software suite[71], using PDB entry 6JC2.pdb as a search model for both the heavy and light chain. Electron and difference density maps were manually inspected, and the model was improved using Coot[72] and several rounds of refinement with PHENIX[71]. The calculation of Rfree used 5.04 % of the data. Statistics over data and refinement can be seen in Supplementary Table 1. The crystallographic data have been deposited to the Protein Data Bank (PDB) with the accession ID: 8C67.

## IgG pulldowns

IgGs were purified in a 96-well plate setup using the Protein G AssayMAP Bravo (Agilent) technology, according to the manufacturer's instructions. Briefly, 10 μl plasma was diluted with PBS to a final volume of 100 μL and applied to pre-equilibrated Protein G columns. Columns were washed with PBS and eluted in 0.1 M glycine (pH 2). The final pH was neutralized with 1 M Tris and saved until further use.

## Trypsin digestion and peptide desalting

Protein samples were resuspended in 8 M urea and reduced with 5 mM Tris(2-carboxyethyl)phosphine hydrochloride, pH 7.0 for 45 min at 37 °C, and alkylated with 25 mM iodoacetamide (Sigma) for 30 min at RT, followed by dilution with 100 mM ammonium bicarbonate to a final urea concentration below 1.5 M. Proteins were digested by incubation with trypsin (1/100, w/w, Sequencing Grade Modified Trypsin, Porcine; Promega) overnight. Digestion was stopped using 10% trifluoracetic acid (Sigma) to pH 2 to 3. Peptide clean-up was performed by C-18 AssayMAP Bravo (Agilent), according to the manufacturer's guideline. Samples were lyophilized and resuspended in 50 μl HPLC-water (Fisher Chemical) with 2% acetonitrile and 0.2% formic acid (Sigma).

## LC-MS/MS and bioinformatic analysis

Mass spectrometry analysis was performed on a Q Exactive HF-X mass spectrometer (Thermo Fisher Scientific) connected to an EASY-nLC 1200 ultra-HPLC system (Thermo Fisher Scientific). Peptides were trapped on pre-column (PepMap100 C18 3 μm; 75 μm × 2 cm; Thermo Fisher Scientific) and separated on an EASY-Spray column (ES903, column temperature 45 °C; Thermo Fisher Scientific). Equilibrations of columns and sample loading were performed per the manufacturer's guidelines. Mobile phases of solvent A (0.1% formic acid) and solvent B (0.1% formic acid, 80% acetonitrile) were used to run a linear gradient from 5% to 38% over 90 min at a flow rate of 350 nl/min. Complete MS analysis was conducted at a resolution of 120000, the automatic gain control (AGC) was 3e6, and the injection time (IT) was set to 50 ms. The MS2 acquisition was performed in a data-dependent mode, where the 15 most intense ions were fragmented and analyzed at a resolution of 60000, AGC 1e5, 120 ms of IT, and a normalized collision energy (NCE) of 25. Raw files were searched using Proteome Discoverer 2.5 and

against a mouse UniProt database, including the Fasta sequences of Ab25 antibodies. The precursor and fragment mass tolerances were set to 10 ppm and 0.02 Da, respectively. Methionine oxidation was set as a dynamic modification. Carbamidomethylation of cysteines was included as a static modification.

## Statistical analysis

Statistical analysis was performed with a one-way ANOVA multiple comparisons test followed by a post-hoc correction with Dunnette's correction test for in vitro data or RM one-way ANOVA with multiple comparisons test with Dunnette's for ex vivo data. Kruskall-Wallis test was performed with multiple comparisons corrected with Dunn's post-hoc test for statistical comparisons for the in vivo experiments. Mann-Whitney U two-tailed test was performed when comparing two treatments only. All tests were done with GraphPad Prism.

## Reporting summary

Further information on research design is available in the Nature Portfolio Reporting Summary linked to this article.

## Data availability

The in vitro, in silico and in vivo data shown in the graphs in Figs. 1–8 are provided in the source data file provided with the manuscript. The structural data generated in this manuscript has been deposited in the Worldwide protein data bank (wwPDB) PDB ID: 8C67. The mass spectrometry data generated in this study have been deposited to MassIVE, under the accession: MSV000094287 and is publicly available. Source data are provided with this paper.

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

## Acknowledgements

This work was funded by the Swedish Research Council (Vetenskaps-rådet, JM, PN (VR-2020-01511), the Knut and Alice Wallenberg Founda-tion (JM, OS, LM, PN), the Institut Pasteur (Pasteur-Cantarini-Roux fellowship to Y.K.), and the Alfred Österlunds Foundation (JM, LM, PN). L.M. is funded by the Swedish Research Council (Vetenskapsrådet, VR) (VR-2020-02419). PN and OS are funded by the Mats Paulsson Foun-dation. This work was granted access to the HPC resources of IDRIS under the allocation 2021- A0110711500 made by GENCI (granted to Y.K. and M.N.). HK was supported by the French Agence Nationale de la Recherche (ANR) under grant ANR-22-CPJ2-0075-01. We thank Anki Mossberg for technical assistance. AI was funded by the Royal Physio-graphic Society (43972) and Lars Hiertas Memory (FO2023-0534).

## Author contributions

A.I., Y.K., H.K., J.M., W.B., O.S., L.M., M.O., M.N. and P.N. designed research; A.I., Y.K., E.B., S.W., H.K., M.N.y., B.O., L.H., M.S., M.G., T.S., Y.C.,

A.G.T. and D.T., performed research; A.I., Y.K., E.B., H.K., M.N.y., M.G., A.G.T., L.H. and P.N. analyzed data; A.I., Y.K. and P.N. wrote the paper. All authors contributed to reading and editing the final manuscript.

## Funding

## Competing interests
AI, HK, WB, LH, OS, JM, LM, and PN have filed patents related to the mAbs described in this manuscript. The remaining authors declare no competing interests.

## Additional information

[1]Department of Clinical Sciences Lund, Infection Medicine, Faculty of Medicine, Lund University, Lund, Sweden. [2]Université de Lorraine, CNRS, Inria, LORIA, F-54000 Nancy, France. [3]Institut Pasteur, Université Paris cite, CNRS UMR3528, Structural Bioinformatics Unit, Department of Structural Biology and Chemistry, F-75015 Paris, France. [4]Department of Biology & Lund Protein Production Platform (LP3), Lund University, Lund, Sweden. [5]Department of Immunotechnology and SciLifeLab Drug Discovery and Development Platform, Lund University, Lund, Sweden. [6]Department of Clinical Sciences Lund, Division of Pediatrics, Faculty of Medicine, Lund University, Lund, Sweden. [7]Section for Oral Biology and Pathology, Faculty of Odontology, Malmö University, Malmö, Sweden. [8]Department of Laboratory Medicine, Clinical Microbiology, Skåne University Hospital Lund, Lund University, Lund, Sweden. ✉e-mail: pontus.nordenfelt@med.lu.se

