## [Peer Review File · Nature Communications]

REVIEWER COMMENTS

Reviewer #1 (Remarks to the Author):

The study by Izadi et al describes the development of a hinge-engineered antibody against the M protein of *S. pyogenes*, which displays increased phagocytic activity and in vivo protection in a prophylactic model of *S. pyogenes* infection. The study focuses on a monoclonal that the authors have previously isolated and characterized. When this antibody was expressed as the four different IgG subclasses (human IgG1-4), it exhibited paradoxically lower affinity as IgG3, compared to the other IgG subclasses. By contrast, when the same antibody subclass variants were evaluated in vitro for their phagocytic activity, IgG3 exhibited superior phagocytic activity. The authors then generated an engineered version of IgG (termed IgGh) in which the hinge domain of IgG1 was swapped for that of IgG3. This variant demonstrated potent phagocytic activity in vitro and improved protective activity in vivo. Overall, the manuscript is well-written, and the results are presented in a clear manner; however, there are several limitations (outlined below) and the interpretation of the presented data is inadequate. Further, the translational relevance of the presented approach is rather limited, as results are shown only for one specific monoclonal antibody and the effect in in vivo protection seems to be minor.

1. As the authors mention, assessment of the affinity of the Ab25 mAb for protein M cannot be assessed due to the M protein's conformational stability being dependent on temperature, dimerization, and being attached to its native bacterial surface. Would these limitations also impact the modeling of binding? In other words, given all these variables in terms of Protein M stability, how confident could the authors be about the results of the molecular modeling?
2. To assess affinity (actually it's avidity), the authors measured direct binding of the Ab25 mAb to bacteria by flow cytometry. Surprisingly, they observed lower affinity for the IgG3 subclass variant, but no explanation is given for this result. Have the authors assessed purity, stability, aggregation etc for the IgG subclass variants?
3. Throughout the study, all the assays are based on a single system – phagocytosis using THP-1 cells, measurement of affinity is done entirely by flow cytometry etc. It is necessary (especially for phagocytosis assays) that alternative, complementary readout and assays be used (e.g. using other effector lines). Also, the FcR expression profile of THP cells has not been characterized.
4. It is unclear what the affinity of IgGh for protein M is. Is it comparable to IgG1 or to IgG3?
5. Several controls are missing for the in vitro and in vivo assays (mainly isotype controls). Also, since all the experiments relate to the comparison of IgG3 (or the hinge variant IgGh) to IgG1, the statistical comparison should always be against IgG1 (instead of PBS).
6. The affinity of mouse and human FcRs needs to be validated for the constructs that the authors generated (mainly IgGh) to aid the interpretation of the in vitro and in vivo data.
7. Although not necessary, the authors need to include data from an additional mAb against protein M to determine whether the effects are clone specific or extend to other antibodies that target the same protein.

Reviewer #2 (Remarks to the Author):

In this manuscript, Izadi et al. present data from in vitro and in vivo assays, as well as molecular dynamics simulations, aimed at addressing the hypothesis that the opsonic activity of the anti-M protein monoclonal antibody Ab25 depends on both the antibody's affinity to the antigen and flexibility of the hinge that connects its Fc region to its Fab domains. Working from the observation that although an IgG3 version of Ab25 has reduced antigen binding relative to an IgG1 version, it exhibits high opsonic activity, the authors engineered a hybrid IgG antibody that combined an IgG1 backbone with an IgG3 hinge, found that it had high (IgG1-like) antigen binding affinity and high (IgG3-like) opsonic activity, and concluded that increased Fc flexibility improves opsonic function. Accordingly, this work attempts to address dependence and cooperation between constant and variable regions in IgG antibodies but falls short of doing so for the following reasons.

Major concerns include:

1. A lack of mechanistic understanding/data. The authors make a single hybrid antibody, IgGh (IgG1 backbone/IgG3 hinge), from which they base all of their mechanistic conclusions. If opsonic function really depends on two variables – antigen affinity and hinge flexibility – it requires more than one molecule that varies in only one of these two variables in only one way to define “how altering Fc flexibility can improve Fc-mediated opsonic function and how modifications in the constant domain can regulate Fab-antigen interactions.” To make such claims, a more comprehensive approach is needed. One could imagine engineering a series of chimeric antibodies that have varying antigen affinities across a range of several orders of magnitude, as well as those that have increased and decreased hinge flexibility (there are probably many ways to do this, but something as simple as shortening the IgG3 hinge disulfide bond by disulfide bond and systematically replacing cysteines in the IgG3 hinge with serines could work), and combinations thereof. With opsonic activity data from this suite of engineered antibodies that sufficiently interrogate the possible antigen affinity and hinge flexibility space, one could make meaningful correlations that speak to mechanism.
2. The molecular dynamics simulations, to which half of the figures, tables and supplementary information are devoted, make no substantial contribution to the conclusions of the paper. The one thing that is clear from the MD studies is that a longer hinge results in more relative flexibility of the Fc region to the Fab domains. Arguably, one does not need MD simulations to know this. The remaining measurements reported in Supplementary Table 2 appear, by and large, to not be statistically significantly different between the various antibodies – IgG1, IgG3 and IgGh. Finally, counting H-bonds and other noncovalent intermolecular contacts is meaningless in the absence of mutagenesis and determination of the resulting change in binding energy contribution (e.g., by alanine scanning mutagenesis).
3. In the animal model experiments, shown in Figure 6, the authors find some statistically significant differences between IgGh and PBS. The correct comparisons, though, to suggest that “Increased hinge flexibility of IgG1 confers antibody protection against infection in mice” (the title of the figure), are between IgGh, IgG3 and IgG1. By eye, these do not appear statistically significantly different (although they may be). While the authors address the limitations of the mouse models nicely in the Discussion, they appear to overinterpret potentially nonsignificant data in the Results section.

Minor concerns include:

4. In Figure 1b, the affinity (KD) units are mislabeled. They should be nM not inverse nM (nM^{-1}).

Reviewer #3 (Remarks to the Author):

In the present paper, the authors generated human IgG antibody subclasses based on IgG1, the latter of which they previously demonstrated to be effective targeting the M protein. Based on their findings they also generated a hybrid antibody by combining parts of IgG1 and IgG3.

They find reduced binding of IgG3 towards the M protein, which they attribute to increased flexibility of IgG3 (compared to other subclasses).

They use extensive molecular dynamics simulations to relate the experimental findings to structural and dynamical features at the atomistic level. The topic is of high relevance for the basic structural understanding of antibody interactions and treatments based on antibodies.

I find the topic interesting and relevant. However, in my opinion the conclusions concerning structural dynamics and flexibility are not well supported by the findings. The authors have analyzed the MD trajectories in many ways, however the analysis remains superficial and the physical context of some parts of the analysis is not clear. My following comments will only be concerned with the computational/theoretical part of the work.

Minor (no particular order)

- In the RMSD analysis it is not clear (A) which atoms have been used for RMSD calculation and (B) whether there was a structural superposition prior to RMSD calculation.
- The differences of the RMSF estimate for IgGh and IgG1 and IgG3 are not large, given the large uncertainty in the estimate.
- Large portions of the starting structures seem to originate from in silico modelling. An assessment of their quality is missing.
- Method section: What were the cutoffs for non-bonded interactions?
- Is M domain the same as M protein?
- What are "larger degrees of freedom"?
- Figure 3: Explanation of panel F seems to be missing

Major (no particular order)

- It is not clear how exactly the center-of-mass displacement is calculated (page 5). Is this just the displacement between the endpoints of the simulations or an MSD over the trajectory? If it is the former (which I am assuming), I wonder how this measure can be insightful with respect to the dynamics of the system at all. Furthermore, it is not clear how the uncertainty (assuming 1SD?) of this estimate can suggest a higher flexibility of the Fc domain? ("The larger standard deviations for both IgG3 and IgGh systems suggest the higher flexibility of the Fc domain in those two systems with respect to the M1 protein.")

- In the methods section, the authors state that restraints had to be imposed in order to “avoid drastic rearrangements”. Those drastic rearrangements should be discussed and should not only be mentioned in the methods section. The fact that “drastic rearrangements” have to be avoided, points out that there could be a more fundamental problem with the starting structures which largely seem to originate from in silico modelling.
- Many of the angle distributions seem to be clearly multi-modal. This is not really discussed. The distribution of M1-IgG3 seems to have three distinct modi, whereas the others have only one or two (not clearly visible from the overshadowing plots). Also, discussion of mean and std value is not very meaningful given the multi-modality of these distributions.

Reviewer #4 (Remarks to the Author):

This paper by Izadi and colleagues is an interesting study examining the role of the hinge region in a mAb against the central region of the M-protein of group A streptococci. The investigators started with a monoclonal, Ab25 (an IgG1), which they previously generated. It has an interesting and unusual specificity, which they defined as ‘dual-Fab cis’ binding, meaning that each of the two Fabs binds a different epitope on the M-protein. They then interchanged the subclass-specific domains of the heavy chains of IgG2, 3 and 4. They found that the affinity of the IgG3 derivative to the M-protein was significantly less than the affinity of the parent isotype, IgG1. However, functionally, through a variety of assays showed that it was significantly more potent than the other subclasses. They hypothesized that this was due to a longer and more flexible hinge region.

To test this, they took the hinge from the IgG3 and placed that in the IgG1 antibody to create a hybrid molecule, referred to as IgGh and undertook a number of molecular dynamic simulation tests. Previous studies had suggested that the accessibility of the Fc region is important to increase the avidity of interactions of antigen with the Fab. They demonstrated that the Fc of the IgG1 is perpendicular to the M-protein whereas the IgG3 Fc is most likely parallel to the target M-protein. They recorded different H-bond and salt bridge interactions between the target antigen and the different subclass antibodies. They also observed that the IgG3 hinge present in IgGh was associated with a significant increase in opsonic activity of IgGh over IgG1.

To study the biological significance of the different IgGs, they tested the antibodies’ abilities to reduce systemic bacterial load following skin infection. While IgG3 did not reduce bacterial load, the IgGh significantly reduced the bacterial dissemination to the spleen, liver and kidneys. The authors claim that both IgG1 and IgG3 reduced bacterial dissemination (page 8) but the data (Figure 6) show that the differences with the PBS control are not significant. The authors hypothesize that the different biological functions of IgG3 and IgGh is likely due to the different half-lives of the different antibody subclasses, with IgG3 having a shorter half-life.

Critique

My overall impressions of the work are that the data are interesting, but do not prove that the hinge is responsible for the increased effects. The authors need to control for the half-life differences by

administering more IgG3 antibody and measuring its presence to explain the discrepancy between the biological functions of the IgG3 and IgGh.

The authors state in the Abstract that their data provide new therapeutic opportunities for engineering the hinge of IgG3 into other antibody subclasses. To show generic applicability they need to show the same effect with at least one other (if not more) antibodies. This is perhaps more important because this is an unusual antibody ('dual-Fab cis binding'). Their findings might not be generic if a more conventional antibody was studied.

In the original description of the IgG1 antibody (Ab25) they showed biological effect against 3 different emm types. That original data does not appear to be reducible, at least for the emm I type studied here, in that the IgG1 has limited biological effect (Figure 6). The authors say in the Discussion that this is due to the different methods of infection employed. This is surprising given that the effect that is monitored is not in the skin but in distant systemic organs (liver, spleen, kidneys). Perhaps, the IgGh would be no better than the IgG1 as demonstrated in the previous paper if the methods of infection were the same as in the previous paper? The authors need to replicate the original finding and show that the IgGh is better than the IgG1 in that model. I would also like to see more than one challenge strain used, as in their previous paper.

The last sentence of the Discussion hints that this antibody (IgGh) could be useful in treating severe streptococcal infections. Before that could be undertaken, they would need to demonstrate that the antibody does not cross-react with human heart or brain proteins, as the M-protein is thought to be central to the pathogenesis of rheumatic fever. This should at least be acknowledged.

Reviewer #5 (Remarks to the Author):

Dear Reviewers,

We, on the authors' behalf, are grateful for the constructive comments given by the 5 reviewers and the chance to properly address them with major new experiments and text clarifications:

1. Based on comments from reviewer #1: We have performed experiments with primary neutrophils and monocytes from healthy donors to confirm our phenotype in an ex vivo setting (see updated **Figure 1 and Figure 5**). We have added SPR experiments with FcγR affinity for our mAbs (**Supp. Figure 11**). We have also added necessary subclass controls for the phagocytosis assays (**Supp. Figure 1E**) and validated the Fc-gamma receptor expression of the THP-1 cells (**Supp. Figure 1D**).
2. Based on comments from reviewer #2: We have systematically characterized the hinge of IgG3 in 4 hybrids, where we observed a fascinating phenotype that the 47-hinge version (the original IgGh) exceeds the shorter versions (17 and 32 aa) and even the slightly longer 62 aa version (see **new Figure 4**).
3. Based on comments from reviewer #4: To increase the translational relevance of the IgGh₄₇ construct based on Ab25, we have shown that against different *emm* types from clinical isolates, it shows the same potent phenotype over IgG1 in 5 out of 6 strains tested (see **new Figure 7**).
4. To show general applicability, as reviewers #1 and #4 suggested, we have generated 3 anti-SARS-COV-2 mAbs from IgG1 to IgGh₄₇ subclass and demonstrated that they exhibit the same phenotype of enhanced Fc-mediated function (see **new Figure 8**).
5. We have used LC-MS/MS on plasma from the animal experiments to analyze the levels of human IgG (from the previous animal experiment) and observed similar levels of IgG3 as the other mAbs (see **Figure 6E-F**). This helps us to interpret the animal experiment outcome better as requested by reviewer #4.
6. We have now changed the statistical comparison from the negative control to IgG1 or IgGh₄₇ when more suitable for **all figures**, as requested by reviewers #1-2.
7. We have added several new supplementary figures for gating strategies for in vitro experiments (**Supp. Figures 2-3**), in silico binding free energy calculations (**Supp. Figure 7**), and alanine scanning by FoldX (**Supp. Figures 8-9**).
8. We also added a more in-depth description of the modeling for IgG3-M1, as requested by reviewers #3 and 1 (**Supp. Figure 13**), to show the measures taken to have a high-quality in silico model, including using crosslinking experimental data and X-ray crystal structures of the Ab25 Fab's and M1 protein crystal structure.
9. We have clarified that Ab25 does not react with human tissue as determined by tissue microarray and published previously, as requested by reviewer #4.
10. The difference between the previous infection model, where Ab25 IgG1 was assessed, and the one used in this work is the route of administration of bacteria. The model used herein is a two-stage infection model where a localized skin infection is generated, which may progress to an invasive systemic infection. The importance of the route of administration for the pathogenesis of mouse disease models with these bacteria was recently elucidated in a publication in Nature Communications (2). This work highlights how anti-IgG bacterial enzymes are not active in a direct systemic (IP) model of systemic infection. These enzymes are active in the two-stage infection model used in the current manuscript and more accurately capture the clinical picture of how these pathogens cause disease (local infection > systemic) and defend against host immune mediators. For this reason, we did not use Ab25 in the older model and have now highlighted the discrepancy between the models, as requested by reviewer #4.
11. Finally, we have clarified several points regarding the MD simulations raised by reviewers #1, #2, and #3. In addition, we clarified the novel findings shown in the MD compared to older studies (Roux et al, Journal of Immunology, ref. 3) concerning IgG3 flexibility.
12. **List of major figure changes:**
 - Figure 1: new panels L-M
 - Figure 4: new figure
 - Figure 5: new panels H-I

Figure 6: new panel E
Figure 7: new figure
Figure 8: new figure

Supp. Figure 1: new panels D-E
Supp. Figure 2: new figure
Supp. Figure 3: new figure
Supp. Figure 4: previous main figure
Supp. Figure 7: new figure
Supp. Figure 8: new figure
Supp. Figure 9: new figure
Supp. Figure 11: new figure
Supp. Figure 13: new figure

Due to these extensive additions, we have restructured the manuscript and added or modified text throughout the manuscript. We agree with the reviewers that the *in vivo* data has weaknesses, which we find difficult to address with the current animal models available for group A streptococci. We, therefore, decided to remove the word protective from the title to tone this aspect down. The hinge length characterization and the results that specifically the 47 aa hinge version is the best variant, and due to the extensive new findings on the generalizability of our construct with SARS-CoV-2 mAbs and more *emm* types, we have now altered the title to reflect these new facts: “The hinge-engineered IgG1-IgG3 hybrid subclass IgG₄₇ potently enhances Fc-mediated function of anti-streptococcal and SARS-CoV-2 monoclonal antibodies”.

We are grateful for these constructive suggestions as they have substantially improved the manuscript. See our detailed point-by-point response below.

Best regards,
Arman Izadi and Pontus Nordenfelt on behalf of the authors

REVIEWER COMMENTS

Reviewer #1 (Remarks to the Author):

The study by Izadi et al describes the development of a hinge-engineered antibody against the M protein of S. pyogenes, which displays increased phagocytic activity and in vivo protection in a prophylactic model of S. pyogenes infection. The study focuses on a monoclonal that the authors have previously isolated and characterized. When this antibody was expressed as the four different IgG subclasses (human IgG1-4), it exhibited paradoxically lower affinity as IgG3, compared to the other IgG subclasses. By contrast, when the same antibody subclass variants were evaluated in vitro for their phagocytic activity, IgG3 exhibited superior phagocytic activity. The authors then generated an engineered version of IgG (termed IgGh) in which the hinge domain of IgG1 was swapped for that of IgG3. This variant demonstrated potent phagocytic activity in vitro and improved protective activity in vivo. Overall, the manuscript is well-written, and the results are presented in a clear manner; however, there are several limitations (outlined below) and the interpretation of the presented data is inadequate. Further, the translational relevance of the presented approach is rather limited, as results are shown only for one specific monoclonal antibody and the effect in in vivo protection seems to be minor.

Reviewer: 1. *As the authors mention, assessment of the affinity of the Ab25 mAb for protein M cannot be assessed due to the M protein's conformational stability being dependent on temperature, dimerization, and being attached to its native bacterial surface. Would these limitations also impact the modeling of binding? In other words, given all these variables in terms of Protein M stability, how confident could the authors be about the results of the molecular modeling?*

Author response:

We agree with the reviewer that these limitations, in general, can impact the accuracy of the modeling; however, we took careful steps to avoid such impacts. First, what we reported here is not purely computational models but is supported by experimental data. The integrative structural models are based on cross-linking mass spectrometry data. The MS experiment was carried out in our previous study, and computational models here are carefully selected to fulfill the cross-link constraints (via TX-MS protocol). Second, the Fab domain and partial structure of M1 are defined via X-ray crystallography. Third, we put constraints on the M1 protein for MD simulations to avoid disabilities imposed by detachment from the bacterial surface (please see detailed answer to reviewer 3). Considering all these points together, we believe the integrative models in this study are highly confident. We have added a new supplementary figure (**Supp. Fig. 13**) in the manuscript to show the model's integrity.

See **lines 211-217** in the Results section.

See **lines 867-897** in the Methods section.

See new **Supp. Fig. 13**.

Reviewer: 2. *To assess affinity (actually it's avidity), the authors measured direct binding of the Ab25 mAb to bacteria by flow cytometry. Surprisingly, they observed lower affinity for the IgG3 subclass variant, but no explanation is given for this result. Have the authors assessed purity, stability, aggregation etc for the IgG subclass variants?*

Author response:

This is a good point brought up by the reviewer. Regarding quality control, we always perform SDS-PAGE analysis on our antibodies to show purity and correct assembly of heavy and light chains. We have worked with these antibodies for several years and have noticed no change in their properties when stored at 4°C for months. We have also tested sterile filtering of the antibody preparations (and always do this before in vivo studies) with no detectable loss in protein concentration or change in the SDS-PAGE results, suggesting no or minimal aggregation.

Regarding the avidity of the antibody-M protein interaction, we left the explanation for the weaker binding by IgG3 in the result section and discussion section regarding the hydrogen-network interaction between the IgG's

and M1. Although only a simulation based on the crystal structure of the IgG Fab, we believe that our alteration of the constant domain has directly influenced the folding of the Fabs so that the interaction between Ab25 IgG3 and the M1 protein is altered, resulting in weaker binding. We discuss the impact of the constant domain on the epitope binding domain in the result and the discussion sections. The reviewer can look at **Figure 3** for Hbond and salt bridge interactions and **Supp. Fig. 8-9** for computational alanine scanning mutagenesis to determine which residues are critical for the Hbond formation. We discuss the limitations of this approach in the discussion section and how we believe a more extensive study is warranted to focus primarily on this issue of elucidating constant-domain and variable-domain interaction using more techniques such as cloning chimeric construct from a broad panel of mAbs, using structural biology techniques such as cryo-EM combined with MD simulations to find important residues which can be mutated to elucidate the molecular mechanisms further.

See **lines 300-356** in the Results section.

See **lines 569-580** in the Discussion section.

See **Figure 3** and new **Supp. Fig. 8-9**.

Reviewer: 3. *Throughout the study, all the assays are based on a single system – phagocytosis using THP-1 cells, measurement of affinity is done entirely by flow cytometry etc. It is necessary (especially for phagocytosis assays) that alternative, complementary readout and assays be used (e.g. using other effector lines). Also, the FcR expression profile of THP cells has not been characterized.*

Author response:

We believe this is a very constructive and valid issue brought up by the reviewer. We generally use the THP-1 cell line as it is well-established for phagocytosis assays and has lower variability due to donor variation. However, to address this, we isolated primary neutrophils and monocytes from 6 volunteering donors and tested the opsonic function of our respective mAbs. As seen in the data added to **Figure 1L-M**, IgG3 is a more potent opsonin across the three effector cell line systems where it induces stronger uptake of bacteria by phagocytes measured in phagocytosis score.

Similarly, for the IgGh₄₇ experiments, we used monocytes and neutrophils as additional effector cell lines. In these experiments, IgGh₄₇ was more potent than both the parent IgG1 and IgG3, respectively, in terms of phagocytosis score and internalization. This new data can be found in **Figure 5H-I**.

Finally, we stained THP1 cells with anti-CD16, anti-CD32, and anti-CD64, showing expressions of all these Fc-gamma receptors. This can be found in **Supplementary Figure 1D**.

See **lines 177-190** and **415-426** in the Results section.

See **lines 621-624** in the Discussion section.

See **lines 785-830** in the Methods section.

See updated **Figures 1, 5, and Supp. Fig 1**.

Reviewer: 4. *It is unclear what the affinity of IgGh for protein M is. Is it comparable to IgG1 or to IgG3?*

Author response: This data is present in the **Supplementary Figure 5** (which is now **Supp. Figure 10B**). The IgGh₄₇ binds with almost exactly the same affinity as IgG1.

See **lines 395-398** in the Results section.

See **Supp. Fig 10**.

Reviewer: 5. *Several controls are missing for the in vitro and in vivo assays (mainly isotype controls). Also, since all the experiments relate to the comparison of IgG3 (or the hinge variant IgGh) to IgG1, the statistical comparison should always be against IgG1 (instead of PBS).*

Author response:

We agree with the reviewer on this point. We have added the IgG1-4 controls for the THP1-phagocytosis experiments corresponding to **Figure 1**. The new figure is shown in **Supp. Fig. 1E**. In addition, all new experiments now contain the IgG1 or IgG3 control when IgG3 was used (**Figures 1, 4-5**).

In addition, concerning the reviewer's point regarding statistical comparison, we have altered our analysis to compare against IgG1 or IgGh₄₇ as the reviewer suggested for all statistical comparisons (see **Figures 1, 4-8**).

See updated **Figures 1, 4-8, and Supp. Fig 1.**

Reviewer: 6. *The affinity of mouse and human FcRs needs to be validated for the constructs that the authors generated (mainly IgGh) to aid the interpretation of the in vitro and in vivo data.*

Author response:

We agree with the reviewer that having this additional information could significantly improve the conclusions of our manuscript, in addition to broadening our understanding of how hinge-engineering influences interactions with the immune system and the pathogen. We performed SPR experiments to determine the molecular affinity of antibodies with different constant domains to human FcRn (important for antibody half-life in vivo) and FcγRI. Similarly, we investigated the interaction with mouse Fc-Receptors to help us understand how the constructs bind to receptors important for functional activity and antibody half-life in relation to the studies conducted in vivo. Unfortunately, the Ab25 variable domain was a bit difficult to work with in these assays, as it, in particular at the more acidic pH used to measure antibody-FcRn interaction, nonspecifically bound to the streptavidin-coated sensor chip surface to such a degree that a meaningful analysis on kinetics could not be performed. Since we now had access to the IgGh₄₇ constructs (with the same constant domain) of SARS-CoV-2 spike-specific antibodies (discussed under point 7 by the reviewer) that did the technical issues of Ab25, we performed the analysis with them instead. See **lines 473-487** in the Results section.

See **lines 881-902** in the Methods section.

See new **Supp. Fig 11.**

Reviewer: 7. *Although not necessary, the authors need to include data from an additional mAb against protein M to determine whether the effects are clone specific or extend to other antibodies that target the same protein.*

Author response: The reviewer has a valid point on the generalizability of our findings. Reviewer #4 had a similar concern; however, it was focused on non-M protein antibodies. So, to address both reviewer #1's and reviewer #4's concerns, which we agree on, we engineered 3 IgG1 mAbs into the IgGh₄₇ subclass. These antibodies bind to the spike protein, thus increasing the generalizability of our findings. We observed remarkably enhanced function (up to 23-fold improvement, depending on the metric) when making the mAbs as IgGh₄₇ compared to the IgG1 versions for the three unique clones. These experiments requested by the reviewers have greatly improved the generalizability of our findings and the manuscript, so we thank the reviewer for pointing this out. In addition, the Ab36 IgGh₄₇ version is highly potent and a promising candidate to further develop in a future project concerning SARS-COV-2 due to the importance of Fc-mediated effector functions against this viral pathogen.

See **lines 516-551** in the Results section.

See **lines 646-659** in the Discussion section.

See **lines 684-688, 777-784** in the Methods section.

See new **Figure 8.**

Reviewer #2 (Remarks to the Author):

In this manuscript, Izadi et al. present data from in vitro and in vivo assays, as well as molecular dynamics simulations, aimed at addressing the hypothesis that the opsonic activity of the anti-M protein monoclonal antibody Ab25 depends on both the antibody's affinity to the antigen and flexibility of the hinge that connects its Fc region to its Fab domains. Working from the observation that although an IgG3 version of Ab25 has reduced antigen binding relative to an IgG1 version, it exhibits high opsonic activity, the authors engineered a hybrid IgG antibody that combined an IgG1 backbone with an IgG3 hinge, found that it had high (IgG1-like) antigen binding affinity and high (IgG3-like) opsonic activity, and concluded that increased Fc flexibility improves opsonic function. Accordingly, this work attempts to address dependence and cooperation between constant and variable regions in IgG antibodies but falls short of doing so for the following reasons.

Major concerns include:

Reviewer: 1. *A lack of mechanistic understanding/data. The authors make a single hybrid antibody, IgGh (IgG1 backbone/IgG3 hinge), from which they base all of their mechanistic conclusions. If opsonic function really*

depends on two variables – antigen affinity and hinge flexibility – it requires more than one molecule that varies in only one of these two variables in only one way to define “how altering Fc flexibility can improve Fc-mediated opsonic function and how modifications in the constant domain can regulate Fab-antigen interactions.” To make such claims, a more comprehensive approach is needed. One could imagine engineering a series of chimeric antibodies that have varying antigen affinities across a range of several orders of magnitude, as well as those that have increased and decreased hinge flexibility (there are probably many ways to do this, but something as simple as shortening the IgG3 hinge disulfide bond by disulfide bond and systematically replacing cysteines in the IgG3 hinge with serines could work), and combinations thereof. With opsonic activity data from this suite of engineered antibodies that sufficiently interrogate the possible antigen affinity and hinge flexibility space, one could make meaningful correlations that speak to the mechanism.

Author response: We thank the reviewer for the constructive feedback on our manuscript. The reviewer's point on more constant-domain modified mAbs is an excellent point to test the strength of our conclusions. Thus, we set out to engineer further modified hinge-variants: one mAb with IgG1 backbone and a 17 aa hinge (the core IgG3 hinge), one mAb with a 32 aa hinge (the core IgG3 hinge with the exon repeat 1), the original IgGh (47 aa hinge, core hinge + 2 exon repeats) and the most extended IgG3 hinge variant (62 aa, core hinge + 3 exon repeats). We first determined their reactivity to M1 and showed they are similar in binding to IgG1, showing that merely extending the hinge does not necessarily affect binding and that other factors must explain the reduced binding we see with IgG3 (see discussion from MD data).

Regarding opsonization, we observed a fascinating phenomenon where each subsequent hinge-length increase from IgG1's 15aa up to 47aa (15=17, 17<32<47) hinge enhances Fc-mediated function. Interestingly, there is a clear cap at 47 aa since the 62 aa version (albeit better than IgG1) was inferior to the 47 aa hinge version. This result is interesting because:

- 1) it shows how different hinge lengths influence function, a direct correlation which 2) is not linear and perhaps more biphasic
- 3) there is a clear cap to how much we can increase mAb flexibility before the effect is diminished.

In line with this much broader understanding (which we thank the reviewer for) we have changed the title from “The increased hinge flexibility of an IgG1-IgG3 hybrid monoclonal enhances Fc-mediated protection against group A streptococci” to “The hinge-engineered IgG1-IgG3 hybrid subclass IgGh₄₇ potently enhances Fc-mediated function of anti-streptococcal and SARS-CoV-2 monoclonals” as to be more nuanced in the interpretation of hinge-modification (as more hinge and more flexibility is not linearly better endlessly, only up to 47 aa). Throughout the manuscript, we have nuanced our interpretation of hinge-engineering and modifying antibody flexibility. We believe this has improved the manuscript and better balanced the conclusions. The abstract has been changed as well, and the sentence “how altering Fc flexibility can improve Fc-mediated opsonic function and how modifications in the constant domain can regulate Fab-antigen interactions” has been removed and replaced with:

“We explored the impact of hinge-engineering by generating a panel of IgG antibodies, IgGh_{xx}, containing the CH1-3 domains of IgG1 and different segments of IgG3's hinge. Hinge-engineering enhanced opsonic function, with the most potent hinge having 47 amino acids. IgGh₄₇ far exceeded the parent IgG1 and even the IgG3 version. The IgGh₄₇ was protective against *Streptococcus pyogenes* in a systemic infection mouse model, contrary to parent IgG3 and IgG1. The *in vitro* phenotype of IgGh₄₇ was generalizable to clinical isolates with different *emm* types. Finally, we generated IgGh₄₇ versions of anti-SARS-CoV-2 mAbs, which exhibited strongly enhanced *in vitro* opsonic function compared to the original IgG1's. The improved function of the IgGh₄₇ subclass in two distant biological systems provides new insights into antibody function and how to enhance it for opsonic function”.

See lines 360-391 in the Results section.

See lines 582-585 in the Discussion section.

See lines 664-692 in the Methods section.

See new Figure 4.

Reviewer: 2. *The molecular dynamics simulations, to which half of the figures, tables and supplementary information are devoted, make no substantial contribution to the conclusions of the paper. The one thing that is clear from the MD studies is that a longer hinge results in more relative flexibility of the Fc region to the Fab domains. Arguably, one does not need MD simulations to know this. The remaining measurements reported in Supplementary Table 2 appear, by and large, to not be statistically significantly different between the various antibodies – IgG1, IgG3 and IgGh. Finally, counting H-bonds and other non-covalent intermolecular contacts is meaningless in the absence of mutagenesis and determination of the resulting change in binding energy contribution (e.g., by alanine scanning mutagenesis).*

Author response:

We believe the MD data does help elucidate some important new observations. To clarify the role of MD better and address the points raised by the reviewer, we added more supplemental data and adjusted the text regarding MD. Still, we also slightly reduced its role in the paper by moving one figure into the supplement, adding more supplement data for clarification, and shortening and adjusting the text regarding MD.

To motivate why the MD data adds value, it is important to highlight previous work regarding IgG subclass flexibility (Roux et al. Flexibility of human IgG subclasses. J Immun. 1997, ref. 3). The authors performed electron microscopy to determine the molecular flexibility of IgG Fc relative to its Fab in the absence of any antigen. IgG3 flexibility seemed to exceed that of IgG1 ($\pm 36^\circ$ for IgG3 and $\pm 30^\circ$ for IgG1). However, our study does not show this. The control IgGh₆₂ in the simulation has less Fab-Fc flexibility than both IgG1 and IgG3 in total angle span and SD. Rather, the angle between Fc and the antigen and movement in 3D space is much larger for IgG3 than IgG1. We believe these interesting observations are of value since many studies extrapolate from the earlier study by Roux et al. Thus, even if one can imagine that a longer hinge equals higher flexibility, the term "higher flexibility" is unspecific and can mean different things. In this work, "higher flexibility" is clearly not Fab-Fc flexibility, nor Fab1-Fab2 flexibility, but rather increased movement and flexibility of the Fc relative to the center of mass of the antigen (M1 protein). Without the MD simulations, we would also have assumed a larger Fab-Fc angle. So, although we understand the reviewer's point that the MD is not adding definite evidence to the conclusions, its descriptive nature reveals important interactions that we felt were crucial to highlight since this phenotype has not been observed previously. Below is an image from Roux et al, which highlights the "increased flexibility of IgG3" which differs from our findings.

FIGURE 7. Histogram showing the angular measurements between the tagged IgG Fab arms and between Fc and the Fab arm showing the smallest angle.

We agree with the reviewer that the differences from Supplementary Table 2 are not statistically significant. One main reason is that those values are averaged over the replicates of each system. Also, in the case of the RMSD, the analyses were performed on individual domains and out of the context of the M1-IgG. Therefore, the subtle differences somehow confirm the convergence of simulations yet highlight differences between them. Also, the analysis of distances and angles fails to fully demonstrate the differences since these are only at the 2D level. For this reason, we also included **Figure 3 B-E**, which shows the displacement of the Fc domain in 3D space and highlights significant differences between the antibodies.

The MD simulations also revealed atomic details of the interactions between the antibodies and M1. I) From these simulations, we could highlight the existence of important salt bridges between the M1 and the Fab2 domains of all the antibodies. II) We also observed an important shift in the H-bond network when comparing the M1-IgG1 to M1-IgG3. While many H-bonds were observed between the Fab1 and the M1 in M1-IgG1 system, the simulations revealed that both for IgG3 and IgGh₆₂, the network shifted toward more H-bonds between the Fab2 and M1. This finding could explain the loss of binding affinity for IgG3. III) The cluster analysis suggested how the hinge flexibility could result in the bending of the Fc domain. This representative conformation (reported in **Figure 3F**) shows that the Fc domain could possibly become more accessible, which, in turn, could facilitate the opsonization.

To address the concern regarding the intermolecular contacts, we performed binding free energy calculations and alanine scanning using FoldX (**Supp. Fig. 7-9**). We thank the reviewer for this comment because it allowed us to identify a set of Fab residues indirectly involved in the binding between the Fab domain and the M1 protein. The results are reported in the Results and Methods sections.

Finally, to address the reviewers' concern regarding inserting mutations in the M1 protein or paratope of Ab25 to elucidate the contribution of each residue, we discussed the limitations of our study in the discussion section. We believe that a larger study is warranted to properly elucidate the influence of the constant domain on antigen binding capability with a broad range of antibodies targeting different antigens and then generating chimeric mAbs to understand which segment of the constant domain influences the binding property for each mAb. By performing MD or CryoEm, it would be possible to elucidate which residues are altered concerning Hbond formation, salt bridges and subsequently perform mutagenesis to confirm that data. However, given the extensive revisions we have already included in the revised manuscript and the primary focus on opsonization, we believe that work is better fitted in a follow-up study primarily focusing on this fundamental issue regarding how antibodies function.

See **lines 315-356** in the Results section.

See **lines 569-580** in the Discussion section.

See **lines 998-1010** in the Methods section.

See **Figure 3 and new Supp. Fig. 7-9**.

Reviewer: 3. *In the animal model experiments, shown in Figure 6, the authors find some statistically significant differences between IgGh and PBS. The correct comparisons, though, to suggest that "Increased hinge flexibility of IgG1 confers antibody protection against infection in mice" (the title of the figure), are between IgGh, IgG3 and IgG1. By eye, these do not appear statistically significantly different (although they may be). While the authors address the limitations of the mouse models nicely in the Discussion, they appear to overinterpret potentially nonsignificant data in the Results section.*

Author response: We previously compared to PBS since it was evident that the IgG1 and IgG3 were not as protective when compared to IgGh₄₇ with reference to the negative control PBS, and thus, we wanted to highlight this by that statistical comparison. However, given that the effects of IgG1 and IgG3 are comparable to PBS, we understand that comparing them to PBS and not IgGh₄₇ is less informative. In line with this, we have now altered the statistics and compared PBS, IgG1, and IgG3 with that of IgGh, which shows some important statistically significant differences between IgG3 vs. IgGh₄₇, IgGh₄₇, and PBS and almost for IgG1 (P=0.07 for liver). Since it is difficult to determine the mechanism of protection in an in vivo experiment, we have rephrased the figure title

to: "IgGh₄₇ confers protection against systemic infection in mice" since we interpret it as being protective while IgG1 and IgG3 are not. Given that most mice in the cohort do not have bacteria in the different organs (70-90% bacteria-free) while IgG1 and IgG3 only have 10-30% bacteria-free mice in their cohort, this is a clear improvement in protective effect. In addition, the bacterial count is between 100-1000-fold lower.

See **lines 108-109** in the Introduction section.

See **lines 429-465** in the Results section.

See **lines 601-636** in the Discussion section.

See updated **Figure 6**.

Reviewer: *Minor concerns include:*

4. In *Figure 1b*, the affinity (KD) units are mislabeled. They should be nM not inverse nM (nM^{-1}).

Author response: We thank the reviewer for noticing this error. It has now been corrected.

Reviewer #3 (Remarks to the Author):

In the present paper, the authors generated human IgG antibody subclasses based on IgG1, the latter of which they previously demonstrated to be effective targeting the M protein. Based on their findings they also generated a hybrid antibody by combining parts of IgG1 and IgG3.

They find reduced binding of IgG3 towards the M protein, which they attribute to increased flexibility of IgG3 (compared to other subclasses).

They use extensive molecular dynamics simulations to relate the experimental findings to structural and dynamical features at the atomistic level. The topic is of high relevance for the basic structural understanding of antibody interactions and treatments based on antibodies.

I find the topic interesting and relevant. However, in my opinion the conclusions concerning structural dynamics and flexibility are not well supported by the findings. The authors have analyzed the MD trajectories in many ways, however the analysis remains superficial and the physical context of some parts of the analysis is not clear. My following comments will only be concerned with the computational/theoretical part of the work.

Minor (no particular order)

Reviewer: *In the RMSD analysis it is not clear (A) which atoms have been used for RMSD calculation and (B) whether there was a structural superposition prior to RMSD calculation.*

Author response: The RMSD analysis was performed using Ca atoms, and the trajectories were superposed on the equilibrated structure. The details of the RMSD and RMSF calculations could be found in the "Materials and Methods" section, under the "Molecular dynamics simulations" paragraph. Here is an excerpt:

*"The root mean square deviations (RMSD) of the studied complexes from the equilibrated structure were measured on the C α atoms along simulation time for all the replicates (**Fig. 2**). All systems were fully relaxed after 100 ns. Then, we calculated the residue root mean square fluctuations (RMSF) over the last 900 ns of every simulation (**Fig. 2**) with respect to the average conformation and over the C α atoms."*

See **lines 957-962** in the Methods section.

Reviewer: *The differences of the RMSF estimate for IgGh and IgG1 and IgG3 are not large, given the large uncertainty in the estimate.*

Author response: For simplicity, the results reported in **Figure 2** and **Supplementary Table 2** are averages over all the replicates of each simulation and could not clearly highlight the differences. The figure below represents the RMSF values in each replicate, in which the differences between the dynamic behavior of the IgGs are better distinguishable.

Reviewer: Large portions of the starting structures seem to originate from *in silico* modelling. An assessment of their quality is missing.

Author response: We agree with the reviewer that the starting structure as a whole originated *in silico* modeling; however, there is extensive experimental support for each part. While the Fc domain of the IgGs are predicted with AlphaFold with high confidence as a conserved part of IgG, we have determined the structure of Fab domains via X-ray crystallography. Also, the M1 model is a mixture of X-ray crystallography (pdb id: 2nxn) and homology modeling, as used in several previous studies (Hauri et al, Nature Communications; Khakzad et al, PLoS Computational Biology; Bahnan et al, Embo Molecular Medicine, ref. 4-5 and 1) validated by experimental data. We used a similar strategy to model M1 as shown in a publication where M28 was modeled in a similar manner (Chowdhury et al, mSystems, ref 6). Finally, to determine the interactions, we used cross-linking mass spectrometry data as constraints, increasing the accuracy of the final complex. To show how the *in silico* models were made, we added a new supp figure (**Supp. Fig 13**) to highlight the quality of this model. We thank the reviewer for this constructive suggestion.
 See **lines 850-879** in the Methods section.
 See **lines 216-2229** in the Results section.
 See new **Supp. Fig. 13**.

Reviewer: Method section: What were the cutoffs for non-bonded interactions?

Author response:

In this study, we only reported the analysis of H-bonds and salt bridges. The details for all the cut-offs used to measure those interactions are explained in the “Materials and Methods” section, in the “Hydrogen bonds and salt bridges” paragraph.
 See **lines 946-957** in the Methods section.

Reviewer: Is M domain the same as M protein?

Author response: We thank the reviewer for pointing out this mistake. The correct term is M protein. We modified that in the manuscript.

Reviewer: What are “larger degrees of freedom”?

Author response: We referred to large and flexible movements of the Fc domain with this phrase. A clear representation of the movements has been shown in Figure 3 B-E.
See Fig. 3.

Reviewer: Figure 3: Explanation of panel F seems to be missing

Author response: We should have added the F label to the figure legend. The explanation starts with this sentence: “The angle changes formed between the Fc domain and protein M1 are reported for IgG1 (in pink), IgG3 (in cyan), and IgGh₆₂ (green) when both Fabs contact the protein M1.” We added the label on the figure legend.
See Fig. 3.

Reviewer: It is not clear how exactly the center-of-mass displacement is calculated (page 5). Is this just the displacement between the endpoints of the simulations or an MSD over the trajectory? If it is the former (which I am assuming), I wonder how this measure can be insightful with respect to the dynamics of the system at all. Furthermore, it is not clear how the uncertainty (assuming 1SD?) of this estimate can suggest a higher flexibility of the Fc domain? (“The larger standard deviations for both IgG3 and IgGh systems suggest the higher flexibility of the Fc domain in those two systems with respect to the M1 protein.”)

Author response: The center-of-mass calculations were performed on all the generated trajectories (snapshots were taken every 100 ps, leading to 10000 conformations for each replicate of every system); thus, they were not taken at endpoints and used the whole dataset, taking into accounts 10000 observations of each replicate. This analysis provides a 3D-level assessment of the flexibility of the Fc domain (Figure 2B-E). In contrast, the analysis of distances between the Fc domain and M protein (Figure 2A) corresponds to 1D-level behavior and the analysis of angles (Figure 2F) to the 2D-level. Thus, figure 2B-E clearly shows how the Fc is much more mobile in 3D space for IgG3 and IgGh₆₂ while for IgG1 it is highly restricted. In addition, the standard deviation was highlighted because, across the replicates, the mean value of displacement was larger for IgG3 and IgGh₆₂ “111.01±24.50 and 167.13±17.60 Å, respectively, compared to IgG1 (94.10±6.77 Å, Supp. Table 2).” This again highlights the more dynamic behaviors of IgG3 and IgGh₆₂, and that is what we referred to with the statement: “The larger standard deviations for both IgG3 and IgGh₆₂ systems suggest the higher flexibility of the Fc domain in those two systems with respect to the M1 protein.”. Since perhaps it's better to highlight the absolute displacement as the reviewer is possibly suggesting, we now altered the statement to reflect that fact.
See lines 959-977 in the Methods section.
See lines 231-248 in the Results section.

Reviewer: In the methods section, the authors state that restraints had to be imposed in order to “avoid drastic rearrangements”. Those drastic rearrangements should be discussed and should not only be mentioned in the methods section. The fact that “drastic rearrangements” have to be avoided, points out that there could be a more fundamental problem with the starting structures which largely seem to originate from in silico modelling.

Author response: The M protein is a long dimeric coiled coil embedded in the membrane from one side. Also, it has been shown that the surface of bacteria is covered with a dense layer of M proteins forming PPIs with other proteins. However, the limits of computational resources would not allow us to perform MD simulations of the full-length M protein embedded in the membrane. Therefore, we had to consider only the portion of M protein that interacts with the antibodies (taking into account the experimental information and data from cross-linking mass spectrometry). Since this is not the ideal condition, the M1 protein, without any restraints, could be highly flexible. To avoid such artifacts, we decided to impose restraints to provide a condition closer to the native one. We thank the reviewer for pointing this out and have clarified the methods section concerning the structural models.
See lines 850-879 in the Methods section.

Reviewer: Many of the angle distributions seem to be clearly multi-modal. This is not really discussed. The distribution of M1-IgG3 seems to have three distinct modi, whereas the others have only one or two (not clearly

visible from the overshadowing plots). Also, discussion of mean and std value is not very meaningful given the multi-modality of these distributions.

Author response: The apparent multi-modality reflects the existence of different replicates. Also, the Fc-Fab angles reported in the manuscript are an accumulation of both Fc-Fab1 and Fc-Fab2 values. The fact that IgG3 and IgGh₆₂ have a multimodal phenotype while IgG1 does not (Fc-M1 angle) highlights the more dynamic nature of these mAbs, which is also reflected in the absolute movement and mobility in 3D space. In previous work, which has been highly cited by the antibody field (Roux et al. Flexibility of human IgG subclasses. J Immun. 1997, 3) where IgG3 flexibility was elucidated, flexibility was determined by taking electron microscopy images of isolated antibodies and determining angle-flexibility (Fab-Fc flexibility) and then pooling the overall samples to get a sampling range, mean value and SD (see also figure from their work in response above). In that work, hinge flexibility, measured as SD of Fab-Fc angle (angle 3 in our data), was reported to be $\pm 36^\circ$ for IgG3 and $\pm 30^\circ$ for IgG1, with minor differences between the two and no difference in the total Fab-Fc angle span (they did not use antigen-antibody complexes to determine angles as in our work). The differences of 6 SD have been highly cited and established the notion that IgG3 is more flexible than the other subclasses. The data set from that previous publication is also clearly multi-modal since different images were taken and pooled to calculate the overall span. In that sense, our dataset only highlights that IgG1's Fc-M1 angle is much less dynamic (not being multimodal), while IgG3 and IgGh₆₂ are (Supp. Fig. 5F). More interestingly, perhaps, our work shows that the Fab-Fc angle is not the most critical metric in this biological context, but rather the angle compared to the antigen (M1 protein) since IgG3 has a larger Fab-Fc angle span and SD than IgG1 which in turn also exceeds that of IgGh₆₂, speaking against the fact that a flexible hinge would increase this metric. Thus, we acknowledge the reviewer's point on multimodality but think that mean value and SD are important given how the field's understanding of IgG subclass flexibility has been cited from the aforementioned work, which used mean and SD to highlight those differences in angles. In that regard, our findings with angles of the antigen-antibody complex highlight how IgG3 hinges provide larger flexibility relative to the antigen (measured as SD and total span) due to this multimodal nature. Below is a figure from our work showing the individual replicates:

The changes of angles along the replicates of MD simulations. The reported angles are formed between A-C) the Fc and M1 domain, D-F) Fc and Fab domain, and G-I) Fab1 and Fab2 domains. The plots from left to right correspond to M1-IgG1, M1-IgG3 and M1-IgGh₆₂, respectively. The shades of blue, red and green represent the results for every replicate of the corresponding MD simulations.

We thank the reviewer for highlighting the multimodality of IgG3 and IgGh₆₂, and we have now added this in the results section to acknowledge it.

See **lines 258-274** in the Results section.

Reviewer #4 (Remarks to the Author):

This paper by Izadi and colleagues is an interesting study examining the role of the hinge region in a mAb against the central region of the M-protein of group A streptococci. The investigators started with a monoclonal, Ab25 (an IgG1), which they previously generated. It has an interesting and unusual specificity, which they defined as 'dual-Fab cis' binding, meaning that each of the two Fabs binds a different epitope on the M-protein. They then interchanged the subclass-specific domains of the heavy chains of IgG2, 3 and 4. They found that the affinity of the IgG3 derivative to the M-protein was significantly less than the affinity of the parent isotype, IgG1. However, functionally, through a variety of assays showed that it was significantly more potent than the other subclasses. They hypothesized that this was due to a longer and more flexible hinge region.

To test this, they took the hinge from the IgG3 and placed that in the IgG1 antibody to create a hybrid molecule, referred to as IgGh and undertook a number of molecular dynamic simulation tests. Previous studies had suggested that the accessibility of the Fc region is important to increase the avidity of interactions of antigen with the Fab. They demonstrated that the Fc of the IgG1 is perpendicular to the M-protein whereas the IgG3 Fc is most likely parallel to the target M-protein. They recorded different H-bond and salt bridge interactions between the target antigen and the different subclass antibodies. They also observed that the IgG3 hinge present in IgGh was associated with a significant increase in opsonic activity of IgGh over IgG1.

To study the biological significance of the different IgGs, they tested the antibodies' abilities to reduce systemic bacterial load following skin infection. While IgG3 did not reduce bacterial load, the IgGh significantly reduced the bacterial dissemination to the spleen, liver and kidneys. The authors claim that both IgG1 and IgG3 reduced bacterial dissemination (page 8) but the data (Figure 6) show that the differences with the PBS control are not significant. The authors hypothesize that the different biological functions of IgG3 and IgGh is likely due to the different half-lives of the different antibody subclasses, with IgG3 having a shorter half-life.

Critique

Reviewer: *My overall impressions of the work are that the data are interesting, but do not prove that the hinge is responsible for the increased effects. The authors need to control for the half-life differences by administering more IgG3 antibody and measuring its presence to explain the discrepancy between the biological functions of the IgG3 and IgGh.*

Author response: This is a good point brought up by the reviewer. Instead of speculating on the amount of IgG3 present compared to IgGh₄₇, we performed LC-MS/MS analysis on mouse plasma from the previous animal experiments taken 24 hours post-infection to measure levels of human IgG. This analysis showed that comparable levels of IgG1, IgG3, and IgGh₄₇ light and heavy chains were present in the mouse plasma. These results indicate that similar levels of IgG were present across all three treatments. Thus, why IgG3 performed worse than IgGh₄₇ could be explained by the lower affinity to mouse FcR (see comment above regarding new FcR-data) or its several-fold worse affinity to M1. In addition, in our *in vitro* assays, IgGh₄₇ is more potent than IgG3, which likely also adds to its better performance. In addition, it has been shown in some studies that human IgG3 performs worse in murine models compared to IgG1 despite potent *in vitro* function with human immune cells (7-9). This discrepancy has been speculated to be possibly explained by its ability to weaker induce effector functions by murine phagocytes than human IgG1 (7-8). Other work suggests that human IgG3 has a worse half-life in murine models, and that improving half-life would lead to enhanced function in murine models (10). But at the reviewers' behest, we showed that *in vivo* half-life was not an issue in our 24-hour experiment. How human subclasses perform in murine models (7-10) is not clear, and we take this into account in our interpretation of IgG3's usefulness against *S. pyogenes*. In the discussion section, we discuss the limitations of comparing IgG3 vs

IgG1 and IgGh47 of this model. However, the comparison with IgG1 and IgGh₄₇ does not suffer from these confounders, and the data thus serves as a valid proof of concept of using this construct.

See **lines 1052-1068** in the Methods section.

See **lines 467-487** in the Results section.

See **lines 601-636** in the Discussion section.

See updated Figure 6 and new **Supp. Fig. 11**.

Reviewer: *The authors state in the Abstract that their data provide new therapeutic opportunities for engineering the hinge of IgG3 into other antibody subclasses. To show generic applicability they need to show the same effect with at least one other (if not more) antibodies. This is perhaps more important because this is an unusual antibody ('dual-Fab cis binding'). Their findings might not be generic if a more conventional antibody was studied.*

Author response: We agree with the reviewer on this point. Thus, we set out to re-engineer 3 mAbs targeting the spike protein to show generic applicability. We observed remarkably enhanced function (up to 23-fold, depending on the metric) when making the mAbs as IgGh₄₇ compared to the IgG1 versions for the three unique clones. These experiments requested by the reviewers have greatly improved the generalizability of our findings and the manuscript. In addition, the Ab36 IgGh₄₇ version is highly potent and a promising candidate to further develop in a future project concerning SARS-COV-2 due to the extensive importance of Fc-mediated effector functions against this viral pathogen. We thank the reviewer for this constructive suggestion, which has strengthened the argument for using this hybrid construct.

See **lines 516-551** in the Results section.

See **lines 638-659** in the Discussion section.

See **lines 684-688, 777-784** in the Methods section.

See new **Figure 8**.

Reviewer: *In the original description of the IgG1 antibody (Ab25) they showed biological effect against 3 different emm types. That original data does not appear to be reducible, at least for the emm 1 type studied here, in that the IgG1 has limited biological effect (Figure 6). The authors say in the Discussion that this is due to the different methods of infection employed. This is surprising given that the effect that is monitored is not in the skin but in distant systemic organs (liver, spleen, kidneys). Perhaps, the IgGh would be no better than the IgG1 as demonstrated in the previous paper if the methods of infection were the same as in the previous paper? The authors need to replicate the original finding and show that the IgGh is better than the IgG1 in that model. I would also like to see more than one challenge strain used, as in their previous paper.*

Author response: The reviewer is referring to the original paper by Bahnan et al, which used only one *emm* type (*emm1*) in a different scruff model of infection, which was the AP1 strain. During this revision, our colleagues (who are co-authors on this manuscript) published a paper in Nature Communications (2), which showed that depending on the route of infection, the bacterial response to host (endogenous) and exogenous IgG is different. More specifically, human IgG is degraded by bacterial enzymes IdeS and EndoS in the two-stage infection model (the one used herein). The integrity of IgG1 was not investigated in the scruff model used in Bahnan et al. However, the administration route differs from the flank infection model used herein. Thus, the explanation for why IgG1 performed worse in this model may be due to altered-host-pathogen interactions in these distinct models. This notion is supported by our data with more intact IgGh₄₇ hinge than IgG1 (**Figure 6E**). Although we agree with the reviewer that the older model might give different results, the translational relevance of such a result might be limited. Since, in the clinical setting, a localized (skin) infection becomes systemic, the use of this two-stage flank infection model provides better translational relevance than the older one employed where bacterial virulence factors most likely are down-regulated.

The reviewer's comment on *emm* type is very important concerning translational potential, as although *emm1* is dominant globally, it is crucial to test the IgGh₄₇ against other *emm* types. It could be so that the potent opsonic phenotype over IgG1 is only seen against *emm1*, which would diminish its potential promise significantly. We added 6 other *emm* types, from different clusters: *emm4*, *emm12*, *emm28*, *emm79*, *emm81*, and *emm179*. These *emm* types differ greatly in sequence similarity concerning Ab25's epitope span (except for *emm4* and *emm79*), as seen

by our sequence alignment (**Figure 7B**). The bacterial strains used were clinical isolates, further increasing the translational relevance of our experiments. Our results, interestingly, showed that IgGh₄₇ is more potent than IgG1 for 5 out of 6 strains tested (emm4-emm81) and slightly better than IgG1 against *emm179*. Thus, we show that the phenotype of IgGh₄₇ is generalizable to other *emm* types expressed by clinical isolates. We thank the reviewer for this comment, which led to experiments that increase the relevance of our study.

See **lines 467-473, 491-512** in the Results section.

See **lines 601-659** in the Discussion section.

See new **Figure 7**.

Reviewer: *The last sentence of the Discussion hints that this antibody (IgGh) could be useful in treating severe streptococcal infections. Before that could be undertaken, they would need to demonstrate that the antibody does not cross-react with human heart or brain proteins, as the M-protein is thought to be central to the pathogenesis of rheumatic fever. This should at least be acknowledged.*

Author response: This is a vital point the reviewer raises, which we fully agree with, especially since therapeutics and vaccines against M protein have failed due to this issue. Human cross-reactivity was investigated in the original paper, where Ab25 was shown not to cross-react with human tissues, such as cardiac and brain tissue. See below from Bahnan et al. This important fact, which was not previously mentioned, is now highlighted in the manuscript.

See **lines 118-120** in the Results section.

See **lines 645-646** in the Discussion section.

Below is the figure from Bahnan et al:

Figure legend from Bahnan et al:

- A. Tissue microarray stained with 10 µg/ml of Xolair, Ab25, 32, and 49, or 2.5% plasma, 500 µg/ml IVIG or 1:100 anti-Troponin antibody. Each spot is a sampling from different tissue. Tissue sections are 6 µm thick, 1.5 mm wide, and were mounted on positively charged glass slides. None of the tissues showed reactivity with the monoclonal samples, except the positive control anti-Troponin, which was reactive with cardiac tissue, as shown. Out of the 30 tissue types present on the slides, the cardiac samples would otherwise be the tissue type that had the largest risk of reactivity due to the M-protein mimicry with heart tissue.
- B. Zoomed in images of tissue spots representing cardiac tissue.
- C. Sequence alignment between M1 protein and tropomyosin. The highlighted area identifies the cross-linked peptide (closest to the interaction site) between Ab25 and Ab49.

Reviewer #5 (Remarks to the Author):

Author response: *We appreciate the help from reviewer 5 in assessing our manuscript.*

References:

1. Bahnan W. et al. (2022). A human monoclonal antibody bivalently binding two different epitopes in streptococcal M protein protects against infection. *EMBO Mol. Med.* **14**, 1–21.
2. Toledo, A.G., Bratanis, E., Velásquez, E. et al. Pathogen-driven degradation of endogenous and therapeutic antibodies during streptococcal infections. *Nat Commun* **14**, 6693 (2023).
3. Roux et al. Flexibility of human IgG subclasses. *J Immunol.* 1997.
4. Hauri, S., Khakzad, H., Happonen, L. et al. Rapid determination of quaternary protein structures in complex biological samples. *Nat Commun* **10**, 192 (2019). <https://doi.org/10.1038/s41467-018-07986-1>
5. Khakzad H. et al. Structural determination of Streptococcus pyogenes M1 protein interactions with human immunoglobulin G using integrative structural biology. *PLoS Computational Biology.* 2021;**17**(1):1–19.
6. Chowdhury S. et al. Streptococcus pyogenes Forms Serotype- and Local Environment-Dependent Interspecies Protein Complexes. *mSystems.* 2021.
7. Overdijk MB, Verploegen S, Buijsse AO, Vink T, Leusen J et al. Crosstalk between human IgG isotypes and murine effector cells. *J Immunol* 2012; **189**:3430-8.
8. Steplewski Z, Sun LK, Shearman CW, Ghayeb J, Daddona P et al. Biological activity of human-mouse IgG1, IgG2, IgG3, and IgG4 chimeric monoclonal antibodies with antitumor specificity. *Proc Natl Acad Sci U S A* 1988; **85**:4852-6
9. B. Pierre, Properties of mouse and human IgG receptors and their contribution to disease models. *Blood* **119**, 5640–5649 (2012)
10. Saito S, Namisaki H, Hiraishi K, Takahashi N, Iida S. A stable engineered human IgG3 antibody with decreased aggregation during antibody expression and low pH stress. *Protein Sci.* 2019 **28**(5):900-909.

REVIEWER COMMENTS

Reviewer #1 (Remarks to the Author):

The revised manuscript by Izadi et al has addressed many of the concerns raised by the reviewers; however, there are several instances where the presented data do not sufficiently support the study conclusions and the reported effects remain minor with very limited biological relevance.

1. Although the authors conducted phagocytosis studies using monocytes and neutrophils, it does not appear that there is any significant difference between IgG1 and IgG3 (Fig 1L and M) in phagocytosis and/or internalization. No effects were observed when hinge variants were compared (Fig 2D). Overall, the effects remain minor and more prominent when phagocytosis is expressed as normalized phagocytosis score.
2. Along these lines, it is unclear how the authors define phagocytosis. From the gating strategy, it seems that internalization (using pH sensitive dye) is a more representative readout for phagocytosis. % association artificially inflates the effects, since it just measures the physical association/surface binding of opsonized bacteria. For assays like this, the standard approach would be to use Trypan blue to quench any fluorescence signal from bound bacteria and discriminate between binding vs. internalization.
3. Related to #1, in vivo studies showed no superior activity for IgGh over IgG3 or IgG1.
4. For SPR studies, the authors reported differences in FcRn and FcR binding among the 3 mAbs tested, despite the fact that these mAbs were expressed as the same Fc hinge variant. This is truly unexpected, suggestive of potential technical flaws. Also, not all classes of mouse and human FcRs were evaluated.
5. Despite the low (almost non-existent) affinity of IgG3 for mouse FcRn, half-life/plasma stability studies 24 h post-mAb administration showed that both IgG1 and IgG3 exhibited comparable levels.
6. Similarly, mouse FcR binding profile is totally different between IgG1 and IgG3; yet either subclass showed comparable activity in vivo (although both showed no significant effects over PBS).

Reviewer #2 (Remarks to the Author):

The authors have addressed my and other reviewers' concerns and produced a much improved manuscript. I support accepting the manuscript for publication.

Reviewer #4 (Remarks to the Author):

I have re-reviewed the paper based on the authors' responses to just the questions that I raised. The

authors have satisfactorily addressed all my concerns and the paper is significantly better than the first version. I was particularly pleased to see the COVID data showing generalizability and the functional activity against other Strep A strains.

Reviewer #5 (Remarks to the Author):

Dear reviewers,

We thank the reviewers for agreeing to re-review our manuscript. We are grateful for the constructive feedback in the prior revision, which clearly improved the manuscript, and for the positive response to our efforts to address the previous suggestions. We have now addressed the remaining comments.

REVIEWER COMMENTS

Reviewer #1 (Remarks to the Author):

The revised manuscript by Izadi et al has addressed many of the concerns raised by the reviewers; however, there are several instances where the presented data do not sufficiently support the study conclusions and the reported effects remain minor with very limited biological relevance.

Author response:

We thank the reviewer for identifying further areas that could be improved in our manuscript.

The reviewer has identified a lack of clarity regarding the importance we give to the various metrics used to assess phagocytosis. We have explained below and added in the manuscript why the phagocytosis score is one of the most common metrics used in the field and that the effect size differences observed with the IgGh₄₇ constructs are substantial.

We are also grateful to the reviewer for carefully examining our FcR data and how it can be interpreted, leading us to clarify the heterogeneous binding profile of IgGh₄₇ 77 for certain FcR. We have addressed the reviewer's valid critique below, modified the manuscript and added important references to further add clarity to the findings.

Furthermore, given the clear differences seen in phagocytosis score with group A strep for IgGh₄₇ across 6 clinical isolates, the protective effect in vivo of the IgGh₄₇ construct, and the even more potent effect seen with the biological distinct SARS-COV-2, we respectfully disagree on the suggested limited biological importance of our findings. In addition, the biological relevance of our study has been more generalized with our SARS-COV-2 mAb data, given the prominent importance of Fc-mediated function in that field and the potent enhancement of Fc-mediated function of the anti-spike mAbs.

Arman Izadi and Pontus Nordenfelt

Reviewer 1, point 1: Although the authors conducted phagocytosis studies using monocytes and neutrophils, it does not appear that there is any significant difference between IgG1 and IgG3 (Fig 1L and M) in phagocytosis and/or internalization. No effects were observed when hinge variants were compared (Fig 2D). Overall, the effects remain minor and more prominent when phagocytosis is expressed as normalized phagocytosis score.

Author response:

The reviewer raises an important point that in Figures 1L and 1M, the different metrics used to assess phagocytosis show different effects. We used several metrics including association, internalization, bacterial signal, and phagocytosis score, which we normalized to each individual experiment in an attempt to comprehensively understand the phagocyte-prey interactions. Phagocytosis assays using flow cytometry are commonly used, and phagocytosis score is one of the most commonly used metrics used to assess opsonic function in the past few years. In studies where ADCP (antibody-dependent cellular phagocytosis) and ADNP (antibody-dependent neutrophil phagocytosis) are evaluated, phagocytosis score is arguably the most dominant metric used (internalization is rarely used), which takes into account both the percentage of engaged cells with how much prey is engulfed (1-11). It is also essential that the comparison is done at a suitable prey-phagocyte ratio, which we refer to as multiplicity of prey (MOP). This is why we always make dose-response curves and calculate MOP_{50} values so that we know that we perform experiments in scenarios where the phagocytes are not saturated nor have too low a chance of interacting with the prey (12). Taken together, we likely do our phagocytosis assessment more rigorously and carefully than many other labs.

While we agree that internalization is better than association for quantifying the interaction of the phagocyte with the prey, it is not a comprehensive metric and omits the amount of bacteria/virus/prey being engulfed. For instance, it is feasible to have the same degree of internalizing cells for two treatments, but that sole analysis fails to capture that more bacteria were involved in the latter interaction. For this reason, we report multiple metrics relevant to phagocytic interactions, where the phagocytosis score is both a common and efficient way of reporting and analyzing the data and where the other metrics, including internalization, can provide additional details.

We would like to highlight key references showing this to be the case, where examples range from evaluating ADNP and ADCP of polyclonal antibodies raised from vaccination by SARS-COV-2 mRNA vaccines and influenza vaccines(1-11). Thus, the phagocytosis score is highly relevant and is more widely used than internalization and association alone. Moreover, in our manuscript, we observe very large differences, which we can see in Figure 1 (1K, 1H), Figure 4 (4F-G), Figure 5 (5F-I), across clinical isolates (Figure. 7F) and across two biological-distant systems (Figure 8C, 8E). For instance, the effect size is 5-10 fold for the

IgGh₄₇ construct over the other hinge-variants in phagocytosis score, on average 2-fold more across widely different clinical isolates, and for Ab36 IgGh₄₇, it is 23-fold at specific MOPs. From our point of view, these effects are not minor and are comparable to and exceed most effect sizes observed in other referenced studies (1-11). Also, given the increasing evidence showing that Fc-mediated functions are important in immunity against SARS-COV-2, our findings with SARS-COV-2 mAbs extend the biological relevance of our findings (13-17).

*We attribute biological variability to the lower effect size seen for IgG1 vs. IgG3 in Figure 1L and M. To address this, we have added data from 6 more experiments (N=12) where we see that IgG3 exceeds IgG1 1.3-fold and 1.5-fold in the neutrophil and monocyte systems respectively. Please find the **new data in Supp. Fig. 2D and 3D.** and the new text clarifications on phagocytosis score (with adequate references) in the manuscript file. **See lines 148-158 and 194-200 in the results section and lines 796-797 in the methods section.***

Reviewer 1, point 2: Along these lines, it is unclear how the authors define phagocytosis. From the gating strategy, it seems that internalization (using pH sensitive dye) is a more representative readout for phagocytosis. % association artificially inflates the effects, since it just measures the physical association/surface binding of opsonized bacteria. For assays like this, the standard approach would be to use Trypan blue to quench any fluorescence signal from bound bacteria and discriminate between binding vs. internalization.

Author response:

This relates to the point discussed above. We have now clarified why phagocytosis score was the primary metric used to assess opsonic function and that internalization, association, and bacterial signal were used to give the reader further insight. As phagocytosis is a dynamic process, there will be a distribution of cells in these three states (bound prey, internalizing prey, and internalized prey), which all inform about the opsonic ability of a treatment. While Trypan blue is an effective way to remove extracellular signal, we believe it to be less informative than using a pH-sensitive dye or a dual inside-out stain, as Trypan blue removes the ability to assess the amount of prey bound to the phagocyte surface or those not completely internalized.

Reviewer 1, point 3: Related to #1, in vivo studies showed no superior activity for IgGh over IgG3 or IgG1.

Author response:

We disagree with the reviewer on this point. IgG1 and IgG3 show no clear protective effect compared to PBS, while IgGh₄₇ does (Figure 6B-C). The differences are vast: there is a 300-5000-fold reduction in CFU across the three different organs for IgGh₄₇ compared to PBS, while for IgG1 and IgG3, the reduction is 2-5 fold at best (the data is in log₁₀, which might make the vast differences not look as prominent as it is). Furthermore, the CFU/g in the organs are between 10² to 10³ lower after a 24-hour exposure when comparing IgGh₄₇ with IgG1 and IgG3, respectively.

More importantly, 60-80% of all mice in the IgGh₄₇ group have no detectable bacteria in the organs, while PBS reaches 10% at the highest point, and for IgG1-3, it's 10-40%. This means that the IgGh₄₇ treatment completely eliminated bacterial spread in systemic circulation for a majority of the mice, while IgG1, IgG3, and the baseline mouse immune system (PBS) could not. Thus, we believe that we have demonstrated evidence for substantial activity of IgGh47 compared to the other groups. **See discussion at lines 662-664.**

Reviewer 1, point 4: For SPR studies, the authors reported differences in FcRn and FcR binding among the 3 mAbs tested, despite the fact that these mAbs were expressed as the same Fc hinge variant. This is truly unexpected, suggestive of potential technical flaws. Also, not all classes of mouse and human FcRs were evaluated.

Author response:

This is a very important point raised by the reviewer. Across the different FcR, Ab11 IgGh and Ab36 IgGh show almost the exact binding profile to various FcR's, suggesting that for these two mAbs we have a very robust methodology. This is further reflected by the good fit of the Langmuir 1:1 analysis to the actual kinetic interaction, as seen in the Supp. Fig. 11. However, the reviewer refers to the discrepancy with mAb 77 IgGh₄₇ compared to Ab11 and Ab36 IgGh₄₇, which exhibits clear clone-specific differences. At face value, this is perhaps surprising, we agree, but in the antibody field, there have been several instances where clone-specific differences have been observed to various FcR. As an example, here is an excerpt from an article (19) from Margaret Ackerman's lab (one of the leading experts in these matters):

		RIIa-R	RIIa-H	RIIb	RIIIa-F	RIIIa-V
C11	Covalent	1.9 (0.6)		4.9 (0.9)	3.4 (1.1)	2.1 (0.7)
	Captured	1.4 (0.4)		5.1 (1.3)	3.7 (0.8)	1.5 (0.3)
	Antigen	0.2 (0.03)		1.3 (0.09)	0.9 (0.1)	0.1 (0.03)
	Protein A	7.3 (0.3)		5.0 (1.2)	4.2 (1.1)	1.5 (0.3)
N5i5	Covalent	6.5 (0.5)	6.7 (1.4)		8.6 (1.1)	4.0 (2.7)
	Captured	3.9 (1.8)	13.9 (2.3)		5.1 (1.3)	2.6 (0.3)
	Antigen	2.9 (0.5)	2.5 (0.2)		3.0 (0.6)	0.6 (0.1)
	Protein A	4.3 (1.1)	6.1 (1.4)		1.5 (0.4)	3.6 (1.1)
N49P9	Covalent	3.9 (0.9)	5.3 (1.5)	7.7 (0.6)	6.1 (0.9)	1.5 (0.8)
	Captured	3.0 (0.8)	3.2 (0.5)	4.4 (1.1)	4.0 (0.9)	0.8 (0.4)
	Antigen	1.8 (0.3)	1.4 (0.3)	1.8 (0.3)	2.5 (1.0)	0.5 (0.3)
	Protein A	2.5 (0.3)	2.3 (0.8)	3.2 (1.3)	6.1 (2.6)	0.9 (0.6)
N49P7.2	Covalent	4.8 (1.0)	5.8 (0.3)	10.0 (2.1)		2.8 (0.4)
	Captured	3.0 (0.6)	5.5 (3.0)	6.3 (2.3)		2.3 (0.6)

	Antigen	1.4 (0.2)	3.1 (0.7)	5.0 (1.0)		1.2 (0.2)
	Protein A	3.0 (0.1)	3.5 (0.8)	6.3 (1.3)		1.8 (0.3)
VRC01 (IgG3)	Covalent	0.9 (0.2)	6.8 (3.6)	6.0 (1.0)		3.3 (0.4)
	Captured	0.3 (0.06)	1.8 (0.3)	2.3 (1.0)		1.0 (0.4)
	Antigen	0.1 (0.03)	1.1 (0.2)	1.6 (0.3)		0.4 (0.08)
	Protein A	0.3 (0.09)	2.4 (0.2)	2.8 (0.8)		1.7 (0.2)

“Supplemental Table 1: Steady state equilibrium dissociation constants (KD values) measured by surface plasmon resonance.” from <https://doi.org/10.1080/19420862.2023.2231128> (18)

Highlighted in yellow are the Kd (mikroM) values of IgG1 monoclonal antibodies from unique clones. As seen in the table, it is not uncommon to see clone-specific differences in the SPR analysis of FcRs. This clone-specific difference was seen across 4 different ways of analyzing the analyte-ligate interaction: covalent-bound mAb to the chip, captured by protein A, antigen-bound etc. For instance, the IgG1 mAb C11 exhibits an affinity for FcR2A with 190 nM while the N5i5 IgG1 mAb has an affinity of 650 nM, N49P9 has 390 nM and so on. This is not due to technical failure but rather the inherent biological variability of these systems, which are not purely determined by subclass background.

Having established this, we were aware of the possibility of clone-specific differences being an issue, it was for that reason we included three clones of IgGh₄₇ to take into account the possibility of clone-specific variability. In that sense, our data shows that Ab11 and Ab36 IgGh₄₇ have very similar KD to the various FcR (See supp figure 11), while Ab77 IgGh₄₇ is an outlier. The reason for the Ab77 IgGh₄₇'s binding profile is that it exhibits a heterogeneous binding profile, where it initially dissociates quite rapidly, followed by a longer stable phase, which makes the Languir 1:1 analysis model not ideal for explaining this interaction. Thus, the reason for the difference is both biological (variable-domain specific) and technical issues (bad fit). However, since we have very robust and consistent data on Ab11 and 36 IgGh₄₇ in agreement, we believe that the data shows the hybrid construct not to have a detrimental binding profile to the various FcR's. We are appreciative of this critique and have, therefore, added a sentence in the manuscript highlighting the bad fit of Ab77 IgGh₄₇ and additional references to clarify that clone-specific differences are not uncommon. **See changes in lines 485-497 in the result section.**

We analyzed all relevant mouse FcRs to help interpret the *in vivo* data but did not analyze the affinities to human CD16 and CD32 because we felt it would not add so much value to the assays we already performed, such as the human monocyte and neutrophil experiments suggested by the reviewer. Although FcR affinity is an interesting metric, its nature is rather descriptive, especially in the presence of existing *ex vivo*, *in vitro*, and *in vivo* outcome data. FcR-binding assays are usually employed as a surrogate assay in large-system serology assays (20) where ADCP and ADNP assays have not been feasible to perform and are, in our opinion, qualitatively inferior to actual ADNP and ADCP assays themselves.

Reviewer 1, point 5: Despite the low (almost non-existent) affinity of IgG3 for mouse FcRn, half-life/plasma stability studies 24 h post-mAb administration showed that both IgG1 and IgG3 exhibited comparable levels.

Author response:

*The dose of 0.4 mg/mouse was given in these experiments, which corresponds roughly to 20 mg/kg dose. Such a high dose (and commonly given in other mAb in vivo experiments) will certainly affect the pharmacokinetics of all IgG. We decided previously to analyze the human IgG concentration to address the issue of bioavailability with LC-MS/MS rather than speculating based on published data on FcRn affinity. Possibly due to the short timeframe of the experiment, mAb elimination seems not to have been an issue between the three different treatments. Furthermore, there are other complicating factors in vivo that are not captured by cell-free in vitro FcRn affinity experiments. An article in Nature Communications (21) shows that the Fabs of IgGs can interact with the binding of Fc-FcRn. Of note, this was only captured in cell-based assays and not cell-free assays such as SPR. This reinforces our point that the relevance of these assays can be limited, given the complex nature of these issues. We have added a clarification that mAb levels were similar, possibly due to the short time frame found in **line 482-483 in the result section and 635-636 in the discussion.***

Reviewer 1, point 6: Similarly, mouse FcR binding profile is totally different between IgG1 and IgG3; yet either subclass showed comparable activity in vivo (although both showed no significant effects over PBS).

Author response:

*The explanation for why the human subclasses performed much worse than IgGh₄₇ is more likely explained by the weaker in vitro and ex vivo performance of both mAbs. Although it is nice to have FcR affinity data on these subclasses, we do not think it is entirely predictive of in vivo function. Epitope, Fc-orientation, hinge-flexibility, antigen-antibody binding, Fab steric-interruptions, and other various factors might influence the Fc-FcR interactions in vivo, which cannot be predicted by in vitro cell-free assays such as SPR. For instance, two articles show that antigen binding is an important factor influencing FcR affinity (19)-(22). If a small-recombinant protein has these large influences, one can imagine that bacteria coated with lots of M protein interacting with the mAbs (avidity-interactions), which in turn interacts with multiple FcR on immune cells (even more avidity interactions) further add to the complexity of these issues. We therefore think it is very difficult to predict and understand the in vitro, ex vivo, and especially in vivo outcomes based solely on FcR data. Given the reviewers' concern regarding this, we have now added several lines in the discussion section to address the complex nature of these issues with adequate references. **See lines 635-640 in the discussion section.***

We thank the reviewer for these constructive comments.

Reviewer #2 (Remarks to the Author):

The authors have addressed my and other reviewers' concerns and produced a much improved manuscript. I support accepting the manuscript for publication.

Author response:

We thank the reviewer for the appreciated constructive suggestions, which we believe added much depth to our conclusions.

Reviewer #4 (Remarks to the Author):

I have re-reviewed the paper based on the authors' responses to just the questions that I raised. The authors have satisfactorily addressed all my concerns and the paper is significantly better than the first version. I was particularly pleased to see the COVID data showing generalizability and the functional activity against other Strep A strains.

Author response:

We are grateful for the constructive feedback from the reviewer, which helped us show greater generalizability.

Reviewer #5 (Remarks to the Author):

Author response:

We thank the reviewer for co-reviewing our manuscript.

References:

1. Ackerman, M. E., Moldt, B., Wyatt, R. T., Dugast, A. S., McAndrew, E., Tsoukas, S., Jost, S., Berger, C. T., Sciaranghella, G., Liu, Q., Irvine, D. J., Burton, D. R., & Alter, G. (2011). A robust, high-throughput assay to determine the phagocytic activity of clinical antibody samples. *Journal of immunological methods*, 366(1-2), 8–19. <https://doi.org/10.1016/j.jim.2010.12.016>
2. Karsten, C. B., Mehta, N., Shin, S. A., Diefenbach, T. J., Slein, M. D., Karpinski, W., Irvine, E. B., Broge, T., Suscovich, T. J., & Alter, G. (2019). A versatile high-throughput assay to characterize antibody-mediated neutrophil phagocytosis. *Journal of immunological methods*, 471, 46–56. <https://doi.org/10.1016/j.jim.2019.05.006>
3. Butler, A. L., Fallon, J. K., & Alter, G. (2019). A Sample-Sparing Multiplexed ADCP Assay. *Frontiers in immunology*, 10, 1851. <https://doi.org/10.3389/fimmu.2019.01851>
4. Loos, C., Coccia, M., Didierlaurent, A. M., Essaghir, A., Fallon, J. K., Lauffenburger, D., Luedemann, C., Michell, A., van der Most, R., Zhu, A. L., Alter, G., & Burny, W. (2023). Systems serology-based comparison of antibody effector functions induced by adjuvanted vaccines to guide vaccine design. *NPJ vaccines*, 8(1), 34. <https://doi.org/10.1038/s41541-023-00613-1>
5. Xin Tong, Yixiang Deng, Deniz Cizmeci, Laura Fontana, Michael A. Carlock, Hannah B. Hanley, Ryan P. McNamara, Daniel Lingwood, Ted M. Ross, Galit Alter; *Distinct Functional*

Humoral Immune Responses Are Induced after Live Attenuated and Inactivated Seasonal Influenza Vaccination. J Immunol 1 January 2024; 212 (1): 24–34.

6. Bartsch, Y. C., Cizmeci, D., Yuan, D., Mehta, N., Tolboom, J., De Paepe, E., van Heesbeen, R., Sadoff, J., Comeaux, C. A., Heijnen, E., Callendret, B., Alter, G., & Bastian, A. R. (2023). Vaccine-induced antibody Fc-effector functions in humans immunized with a combination Ad26.RSV.preF/RSV preF protein vaccine. *Journal of virology*, 97(11), e0077123. <https://doi.org/10.1128/jvi.00771-23>

7. Routhu, N. K., Stampfer, S. D., Lai, L., Akhtar, A., Tong, X., Yuan, D., Chicz, T. M., McNamara, R. P., Jakkala, K., Davis-Gardner, M. E., St Pierre, E. L., Smith, B., Green, K. M., Golden, N., Picou, B., Jean, S. M., Wood, J., Cohen, J., Moore, I. N., Patel, N., ... Amara, R. R. (2023). Efficacy of mRNA-1273 and Novavax ancestral or BA.1 spike booster vaccines against SARS-CoV-2 BA.5 infection in nonhuman primates. *Science immunology*, 8(88), eadg7015. <https://doi.org/10.1126/sciimmunol.adg7015>

8. Tong, X., McNamara, R. P., Avendaño, M. J., Serrano, E. F., García-Salum, T., Pardo-Roa, C., Bertera, H. L., Chicz, T. M., Levican, J., Poblete, E., Salinas, E., Muñoz, A., Riquelme, A., Alter, G., & Medina, R. A. (2023). Waning and boosting of antibody Fc-effector functions upon SARS-CoV-2 vaccination. *Nature communications*, 14(1), 4174. <https://doi.org/10.1038/s41467-023-39189-8>

9. Boudreau, C. M., Burke, J. S., 4th, Yousif, A. S., Sangesland, M., Jastrzebski, S., Verschoor, C., Kuchel, G., Lingwood, D., Kleanthous, H., De Bruijn, I., Landolfi, V., Sridhar, S., & Alter, G. (2023). Antibody-mediated NK cell activation as a correlate of immunity against influenza infection. *Nature communications*, 14(1), 5170. <https://doi.org/10.1038/s41467-023-40699-8>

10. Kaplonek, P., Cizmeci, D., Kwatra, G., Izu, A., Lee, J. S., Bertera, H. L., Fischinger, S., Mann, C., Amanat, F., Wang, W., Koen, A. L., Fairlie, L., Cutland, C. L., Ahmed, K., Dheda, K., Barnabas, S. L., Bhorat, Q. E., Briner, C., Krammer, F., Saphire, E. O., ... Alter, G. (2023). ChAdOx1 nCoV-19 (AZD1222) vaccine-induced Fc receptor binding tracks with differential susceptibility to COVID-19. *Nature immunology*, 24(7), 1161–1172. <https://doi.org/10.1038/s41590-023-01513-1>

11. Mackin, S. R., Desai, P., Whitener, B. M., Karl, C. E., Liu, M., Baric, R. S., Edwards, D. K., Chicz, T. M., McNamara, R. P., Alter, G., & Diamond, M. S. (2023). Fc-γR-dependent antibody effector functions are required for vaccine-mediated protection against antigen-shifted variants of SARS-CoV-2. *Nature microbiology*, 8(4), 569–580. <https://doi.org/10.1038/s41564-023-01359-1>

12. de Neergaard T, Sundwall M, Wrighton S, Nordenfelt P. High-Sensitivity Assessment of Phagocytosis by Persistent Association-Based Normalization. *J Immunol* 2021; **206**(1):214–24.

13. Zhang, A., Stacey, H. D., D'Agostino, M. R., Tugg, Y., Marzok, A., & Miller, M. S. (2023). Beyond neutralization: Fc-dependent antibody effector functions in SARS-CoV-2 infection. *Nature reviews. Immunology*, 23(6), 381–396. <https://doi.org/10.1038/s41577-022-00813-1>

14. Chan, C. E. Z., Seah, S. G. K., Chye, H., Massey, S., Torres, M., Lim, A. P. C., Wong, S. K. K., Neo, J. J. Y., Wong, P. S., Lim, J. H., Loh, G. S. L., Wang, D., Boyd-Kirkup, J. D., Guan, S., Thakkar, D., Teo, G. H., Purushotorman, K., Hutchinson, P. E., Young, B. E., Low, J. G., ... Hanson, B. J. (2021). The Fc-mediated effector functions of a potent SARS-CoV-2 neutralizing antibody, SC31, isolated from an early convalescent COVID-19 patient, are essential for the

optimal therapeutic efficacy of the antibody. *PloS one*, 16(6), e0253487. <https://doi.org/10.1371/journal.pone.0253487>

15. Hederman, A. P., Natarajan, H., Heyndrickx, L., Ariën, K. K., Wiener, J. A., Wright, P. F., Bloch, E. M., Tobian, A. A. R., Redd, A. D., Blankson, J. N., Rottenstreich, A., Zarbiv, G., Wolf, D., Goetghebuer, T., Marchant, A., & Ackerman, M. E. (2023). SARS-CoV-2 vaccination elicits broad and potent antibody effector functions to variants of concern in vulnerable populations. *Nature communications*, 14(1), 5171. <https://doi.org/10.1038/s41467-023-40960-0>

16. Ullah, I., Prévost, J., Ladinsky, M. S., Stone, H., Lu, M., Anand, S. P., Beaudoin-Bussièrès, G., Symmes, K., Benlarbi, M., Ding, S., Gasser, R., Fink, C., Chen, Y., Tazuin, A., Goyette, G., Bourassa, C., Medjahed, H., Mack, M., Chung, K., Wilen, C. B., ... Uchil, P. D. (2021). Live imaging of SARS-CoV-2 infection in mice reveals that neutralizing antibodies require Fc function for optimal efficacy. *Immunity*, 54(9), 2143–2158.e15. <https://doi.org/10.1016/j.immuni.2021.08.015>

17. Winkler, E. S., Gilchuk, P., Yu, J., Bailey, A. L., Chen, R. E., Chong, Z., Zost, S. J., Jang, H., Huang, Y., Allen, J. D., Case, J. B., Sutton, R. E., Carnahan, R. H., Darling, T. L., Boon, A. C. M., Mack, M., Head, R. D., Ross, T. M., Crowe, J. E., Jr, & Diamond, M. S. (2021). Human neutralizing antibodies against SARS-CoV-2 require intact Fc effector functions for optimal therapeutic protection. *Cell*, 184(7), 1804–1820.e16. <https://doi.org/10.1016/j.cell.2021.02.026>

18. Bahnan, W., Wrighton, S., Sundwall, M., Bläckberg, A., Larsson, O., Höglund, U., Khakzad, H., Godzwon, M., Walle, M., Elder, E., Strand, A. S., Happonen, L., André, O., Ahnlide, J. K., Hellmark, T., Wendel-Hansen, V., Wallin, R. P., Malmstöm, J., Malmström, L., Ohlin, M., ... Nordenfelt, P. (2022). Spike-Dependent Opsonization Indicates Both Dose-Dependent Inhibition of Phagocytosis and That Non-Neutralizing Antibodies Can Confer Protection to SARS-CoV-2. *Frontiers in immunology*, 12, 808932. <https://doi.org/10.3389/fimmu.2021.808932>

19. Crowley, A. R., Mehlenbacher, M. R., Sajadi, M. M., DeVico, A. L., Lewis, G. K., & Ackerman, M. E. (2023). Evidence of variable human Fcγ receptor-Fc affinities across differentially-complexed IgG. *mAbs*, 15(1), 2231128. <https://doi.org/10.1080/19420862.2023.2231128>

20. Brown, E. P., Dowell, K. G., Boesch, A. W., Normandin, E., Mahan, A. E., Chu, T., Barouch, D. H., Bailey-Kellogg, C., Alter, G., & Ackerman, M. E. (2017). Multiplexed Fc array for evaluation of antigen-specific antibody effector profiles. *Journal of immunological methods*, 443, 33–44. <https://doi.org/10.1016/j.jim.2017.01.010>

21. Brinkhaus, M., Pannecoucke, E., van der Kooi, E. J., Bentlage, A. E. H., Derksen, N. I. L., Andries, J., Balbino, B., Sips, M., Ulrichs, P., Verheesen, P., de Haard, H., Rispiens, T., Savvides, S. N., & Vidarsson, G. (2022). The Fab region of IgG impairs the internalization pathway of FcRn upon Fc engagement. *Nature communications*, 13(1), 6073. <https://doi.org/10.1038/s41467-022-33764-1>

22. Zhao, J., Nussinov, R., & Ma, B. (2019). Antigen binding allosterically promotes Fc receptor recognition. *mAbs*, 11(1), 58–74. <https://doi.org/10.1080/19420862.2018.1522178>

REVIEWER COMMENTS

Reviewer #1 (Remarks to the Author):

In the rebuttal letter, the authors provide additional arguments in support of the findings of the study. However, the scientific rigor and translational relevance of the study is very limited, and the presented data do not fully support the study conclusions. The effects, even when multiple metrics are assessed, are in most cases not statistically significant and not consistent across the experimental systems used (in vivo, ex vivo, in vitro).

A few of the many examples that show the lack of scientific rigor:

1. (Related to points 1 and 2 from the previous review) It's true that the authors have used multiple metrics for phagocytosis (association, phagocytic index, internalization etc), claiming that this is the most robust approach to assess phagocytosis. Still the definition of phagocytosis (despite a few previous studies) has always been the uptake and internalization of opsonized particles. Association is just the first step in the process (binding to cell membrane) and it's not reflecting the ability of phagocytes to internalize particles. Even if we consider that the authors arguments on the importance of phagocytosis metrics are valid, there are still no significant effects whatsoever to justify the authors conclusions. The claim that IgG3 exhibits more potent phagocytosis is not supported by any of the data in Figure 1. Except for panel G and experiments using THP-1 cells (even there the effects are statistically minor), there is no significantly better activity of IgG3 over IgG1 for monocyte and neutrophil-based phagocytosis (using all the metrics that the authors provide – normalized association, internalization, normalized bacterial signal, normalized phagocytosis score). This is also the case for Figure 5 – no statistically significant effects to support the authors conclusions. IgGh is no better than IgG1 for THP-1 and monocyte -based assays; there's a minor effect for some (but not all) the metrics in the neutrophil assay.
2. (Related to point 3 from the previous review) The point of the experiment is to demonstrate improved activity of the IgGh47 over the parental IgG1 mAb. So, the comparison should have been between IgGh47 vs IgG1, not vs PBS. Even in the IgGh vs PBS comparison (which is not ideal), the only significant effect is in the liver CFU counts. No significant effects are seen in the spleen ($p=0.05$) or the kidney ($p=0.07$). The authors argue in the rebuttal letter: "More importantly, 60-80% of all mice in the IgGh47 group have no detectable bacteria in the organs, while PBS reaches 10% at the highest point, and for IgG1-3, it's 10-40%." This is just the authors' argument. In reality, there is no statistical comparison to support that there's a meaningful, significant effect.
3. (Related to point 6 from the previous review) "The explanation for why the human subclasses performed much worse than IgGh47 is more likely explained by the weaker in vitro and ex vivo performance of both mAbs." This argument is fundamentally wrong – differences in the in vivo activity among human IgG subclasses cannot be explained by in vitro or ex vivo assays using human cells or cell lines. There are substantial differences in the affinity of human subclasses between mouse and human FcRs; so, any findings between the two systems cannot be interpreted interchangeably. While it is true that "epitope, Fc-orientation, hinge-flexibility, antigen-antibody binding, Fab steric-interruptions, and other various factors might influence the Fc-FcR interactions in vivo", it is well-established that the

predominant determinant of the in vivo effector activity of antibodies is the affinity for FcRs. Without characterization of the affinity of the generated variants for mouse FcRs, the interpretation of the in vivo data is problematic.

General reply to reviewer 1:

We are grateful for the in-depth and constructive comments and suggestions by the reviewer. We have now performed a new analysis, Kaplan-Meier curves, for the in vivo data concerning the efficacy of the hybrid construct in preventing bacterial dissemination, as requested by reviewer 1 with adequate statistical evidence to now support our conclusions. We have carefully analyzed the manuscript regarding statistics, highlighting many significant effects that were not explicitly shown before. Furthermore, we have toned down our conclusions and description of the results concerning in vitro and ex vivo phagocytosis when the effect size has been small or not statistically significant throughout the entire manuscript (Fig. 1, 4-8). Concerning Figure 1, we have now expanded the data shown from the curves for each individual MOP and performed the statistical analysis for those. This MOP-based analysis shows that for the THP1-cells (ADCP), IgG3 exceeds IgG1 several-fold for the various metrics (including internalization). Finally, we have added a section in the discussion where we discuss the limitations of interpreting the in vivo data without mouse FcR data on Ab25 constructs. We are grateful for these constructive comments, which have clarified the significance of our work. Please find our point-by-point response below.

REVIEWER COMMENTS

Reviewer #1 (Remarks to the Author):

"In the rebuttal letter, the authors provide additional arguments in support of the findings of the study. However, the scientific rigor and translational relevance of the study is very limited, and the presented data do not fully support the study conclusions. The effects, even when multiple metrics are assessed, are in most cases not statistically significant and not consistent across the experimental systems used (in vivo, ex vivo, in vitro).

A few of the many examples that show the lack of scientific rigor:

1. (Related to points 1 and 2 from the previous review) It's true that the authors have used multiple metrics for phagocytosis (association, phagocytic index, internalization etc), claiming that this is the most robust approach to assess phagocytosis. Still the definition of phagocytosis (despite a few previous studies) has always been the uptake and internalization of opsonized particles. Association is just the first step in the process (binding to cell membrane) and it's not reflecting the ability of phagocytes to internalize particles."

Author response:

We do agree with the fact that the end goal of opsonization is internalizing particles, which the reviewer points out. The process of opsonization does not, however, occur instantly, and as the reviewer points out, association is the first step of this process. Given that the experiments are standardized to allow 30 minutes of interaction, it is important that a method to assess phagocytosis analyzes all types of interactions during this process, including association as a first crucial step. Cells that are associating can, over time, be internalized, and those that have internalized can internalize more bacteria bound on the membrane (typically a matter of time before that occurs). Therefore, to present the interaction in its entirety (association, bacterial signal, internalization, phagocytosis score) as we do is a more robust and sensitive approach rather than only looking at one metric (such as % internalization or phagocytosis score).

"Even if we consider that the authors arguments on the importance of phagocytosis metrics are valid, there are still no significant effects whatsoever to justify the authors conclusions. The claim that IgG3

exhibits more potent phagocytosis is not supported by any of the data in Figure 1. Except for panel G and experiments using THP-1 cells (even there the effects are statistically minor), there is no significantly better activity of IgG3 over IgG1 for monocyte and neutrophil-based phagocytosis (using all the metrics that the authors provide – normalized association, internalization, normalized bacterial signal, normalized phagocytosis score). This is also the case for Figure 5 – no statistically significant effects to support the authors conclusions. IgG3 is no better than IgG1 for THP-1 and monocyte -based assays; there's a minor effect for some (but not all) the metrics in the neutrophil assay."

Author response:

We thank the reviewer for providing specific feedback regarding statistical significance, which we address below, divided into an IgG3 and an IgG1 section:

IgG3 data: "The claim that IgG3 exhibits more potent phagocytosis is not supported by any of the data in Figure 1. Except for panel G and experiments using THP-1 cells (even there the effects are statistically minor),

Author Response: *We are grateful that the reviewer highlighted parts of the manuscript that can be improved, and many points raised are valid and appreciated, which we address below. However, some claims by the reviewer regarding our data need clarification:*

- In Figure 1C-D, we report a strong (5.7-fold) and statistically significant (CI are not overlapping) difference in THP-1 cell association with bacteria opsonized with IgG3 compared to IgG1. We have now added significance stars to make statistically significant differences accessible for the reader and in the text.

*-In the previous Figure 1I, we see a 1.5-fold (43 % vs. 27%) difference in % internalization for IgG3 vs IgG1, which is statistically significant (**).*

- In the previous Figure 1G, 1J, we see a strong, significant increase in the amount of bacteria captured by the cells. The differences in 1J are, for instance, a statistically significant, 4-fold increase.

- In the previous Figure 1K, the combined phagocytic score metric shows a strong, statistically significant increase with a 5-fold difference (29000 vs 6000).

- Concerning Figures 1F, G, and H, we previously combined all the MOP values as curves and performed statistical comparisons across the combined MOPs, which we have realized now was incorrect. The statistical comparisons should have been done for each MOP condition (as in I and K) since combining the data otherwise adds dose-dependent variation that will affect the statistical power. We have now addressed this and presented the data with table graphs for each corresponding experimental condition (MOPs) and performed statistical analysis on those (so MOP 50 IgG3 vs MOP 50 IgG1). As seen for internalization (new Figure. 1F), IgG3 exceeds IgG1 for all MOP with, on average, a 3-fold (1.5-4-fold span) difference, which are all statistically significant (-***). Similarly, we show bacterial signals where IgG3 outperforms IgG1 on average 5-fold (3-8-fold span) (which are all statistically significant differences *-***) in the new Figure 1G. Finally, we show that across the various MOPs used, IgG3 has, on average, more than 10-fold (5-20 fold span) higher efficacy than IgG1 (again, statistically significant across all MOPs *-**) in the new Figure 1H. We believe the new presentation of data and clarifications show that IgG3 exceeds IgG1 for ADCP (THP-1 cells). We thank the reviewer for allowing us to present our findings better and make accurate statistical comparisons.*

"...there is no significantly better activity of IgG3 over IgG1 for monocyte and neutrophil-based phagocytosis (using all the metrics that the authors provide – normalized association, internalization, normalized bacterial signal, normalized phagocytosis score)."

Author Response:

As we had already written, we saw no initial significant differences with primary monocytes (previous Fig. 1L). With neutrophils, we primarily observe a modest difference in the ability to capture bacteria (non-significant). We acknowledge the large spread when using primary cells, a well-known fact and the reason we use a cell line for more detailed analysis. In the previous revision, we performed additional experiments (making the total N=12) with primary cells to gain statistical power (the current and previous Supp. Fig. 2D and Supp Fig. 3D). With that, we see that the phagocytic score is significantly higher, with a 1.3-fold and 1-5-fold increase with IgG3 over IgG1 with monocytes and neutrophils, respectively. These are more modest effects but statistically significant and to be expected when using primary cells.

We have tempered down and rephrased this section of the manuscript on the instances when the effect size is not large nor statistically significant for IgG1 vs IgG3 comparisons. We have also clarified the main point of the IgG3 analysis in the manuscript: despite lower binding (12.9-fold), it shows a more potent Fc-mediated function with THP-1 cells. See details on manuscript modifications below.

IgGh data: "This is also the case for Figure 5 – no statistically significant effects to support the authors conclusions. IgGh is no better than IgG1 for THP-1 and monocyte-based assays; there's a minor effect for some (but not all) the metrics in the neutrophil assay."

Author Response: *We assess the effect of the IgGh constructs in Figures 4, 5, 7, and 8. In Figure 4, we had already established that IgG1 is inferior to IgGh47 several-fold, which is for almost all metrics assessed (with statistical significance), but not statistically significant nor large effect differences for the proportion of cells that internalize bacteria. We have clarified that for that section of the manuscript and highlighted the lack of effect concerning the internalization metric.*

To comment specifically on the results presented in Figure 5 that the reviewer mentions: In these sets of experiments, the effect was not as strong with IgG3 and IgGh, but they were still significantly better (non-overlapping 95% confidence intervals) in the ability to capture bacteria than IgG1 (2.5-3-fold) (MOP₅₀) and in their capacity to trigger neutrophil internalization (modest effect of 1.2-fold better than both parent IgG's). So we agree that initially if one only looks at that figure, the benefit of IgGh compared to IgG1 is only clear in the ability to capture more bacteria. Given the reviewer's emphasis on internalization, we have now added the analysis from MOP 100 showing that IgGh exceeds IgG1 and IgG3 (2-fold) for the THP-1 cells to replace the previous one at MOP 200, which did not show statistically significant effects. We altered that section's text to reflect this and modified Fig. 5G. The point of Figure 5 is to show that IgGh47 is at least comparable to IgG3 in opsonic ability. However, In addition to Figure 5, there is a generalizability to other GAS strains (Fig. 7), and, most importantly, hybrid class engineering of SARS-CoV-2 monoclonals (Fig. 8). Throughout those figures, the data shows strong, statistically significant effects on opsonic function of the IgGh47 subclass compared to IgG1. There were some instances in Figure 7 that needed clarification on statistical significance and for Figure 8 as well, which we have now altered as done for Figures 1, 4 and 5. We thank the reviewer for the constructive criticism, which helped us highlight and statistically present the data more properly.

Manuscript changes:

- We have now added significance stars to Fig. 1C-D to make the significance clearer for the MOP-curve non-linear regression analysis

- We have now restructured Figure 1 to give more focus on % internalization, bacterial signal and phagocytosis score for the various MOPs seen in the new figure 1F, G, and H (the data in J, K and H are already included in the aforementioned new graphs)
- We have rephrased the text concerning the IgG3 data and IgGh47 data (throughout the manuscript) when not significant and especially tempered down the interpretation of the primary cells data. See Lines 161-200; 393-395; 411-454; 529-535; 560-574
- We have rephrased some words in the abstract lines 32-33 and Introduction summary 106-111
- In Fig. 5 we have added MOP 100 instead of 200 to better highlight the statistically significant results
- Corresponding additions and changes to figure legends in Figures 1 and 5.
- Added statistical significance symbols to Figures 1B, 5B, and 8B, altered 8C to go from lower to higher MOP (3.75 > 30), removed the inserts, and highlighted the differences in phagocytosis score better in Fig. 8E.

"2. (Related to point 3 from the previous review) The point of the experiment is to demonstrate improved activity of the IgGh47 over the parental IgG1 mAb. So, the comparison should have been between IgGh47 vs IgG1, not vs PBS. Even in the IgGh vs PBS comparison (which is not ideal), the only significant effect is in the liver CFU counts. No significant effects are seen in the spleen ($p=0.05$) or the kidney ($p=0.07$). The authors argue in the rebuttal letter: "More importantly, 60-80% of all mice in the IgGh47 group have no detectable bacteria in the organs, while PBS reaches 10% at the highest point, and for IgG1-3, it's 10-40%." This is just the authors' argument. In reality, there is no statistical comparison to support that there's a meaningful, significant effect."

Author reply:

*We are grateful for the reviewer's criticism of our previous argument. Upon further inspection, the reviewer is quite correct in that this is not a solid argument for the in vivo efficacy. Inspired by this argument, we therefore instead analyzed the probability of bacterial dissemination (adverse outcome) across the groups as Kaplan-Meier curves (discussed in the previous rebuttal), which now shows that IgG1 vs. IgGh is statistically different for the liver (**). At the same time, for IgGh vs IgG3 it's statistically significant for the liver (*) and the spleen (*). Only IgGh was statistically significant compared to PBS for all three organs (**-*), while IgG1 and IgG3 were not for each organ. While for the CFU count, we do not see statistically significant differences for IgG1 vs. IgGh, we do for IgG3 vs. IgGh in the spleen; this section of the result section has now been tempered down, and the lack of difference in CFU count explicitly stated. Please find the new graphs in Figure 6E with subsequent text changes in the result section. This new analysis shows that the real effect is not to lower CFU count in the organs (infected mice in hybrid cohorts have similar CFU counts as other treatments) but rather to prevent that bacteria from spreading to said organs (seen as a lower risk of bacterial dissemination in Kaplan-Meier curves). This new insight has been better highlighted in the text. We thank the reviewer for helping us improve our work, which we feel makes our claims more valid and the conclusions better supported.*

Modifications:

- Text changes in the result section in lines 459-484
- Kaplan-Meier curves are added in Figure 6E to show the prevention of bacterial dissemination and the statistical significance of these results
- Corresponding figure legends change to figure 6E
- Small text addition to the methods section in lines 884-887

"3. (Related to point 6 from the previous review) "The explanation for why the human subclasses performed much worse than IgGh47 is more likely explained by the weaker in vitro and ex vivo performance of both mAbs.". This argument is fundamentally wrong – differences in the in vivo activity among human IgG subclasses cannot be explained by in vitro or ex vivo assays using human cells or cell lines. There are substantial differences in the affinity of human subclasses between mouse and human FcRs; so, any findings between the two systems cannot be interpreted interchangeably. While it is true that "epitope, Fc-orientation, hinge-flexibility, antigen-antibody binding, Fab steric-interruptions, and other various factors might influence the Fc-FcR interactions in vivo", it is well-established that the predominant determinant of the in vivo effector activity of antibodies is the affinity for FcRs. Without characterization of the affinity of the generated variants for mouse FcRs, the interpretation of the in vivo data is problematic."

Author reply:

This is quite an interesting discussion, and the reviewer is correct that affinities to FcR can be one of the most predictive and important metrics to include when discussing Fc-mediated effector functions. Concerning the in vivo data, the reviewer is correct that we cannot extrapolate across biological systems and that the previous sentence was erroneously written. We thank the reviewer for pointing this out. We tried to perform SPR experiments with Ab25, which failed for this particular clone due to chip interactions at the experimental conditions required for the assay protocol, while the SARS-COV-2 mAbs worked. Due to clone-specific differences existing for SPR analysis, we are not confident in extrapolating those results to Ab25. Without these affinity data, we agree with the reviewer that we must discuss the in vivo more cautiously. We have now highlighted the limitations of not having data on mouse FcR for Ab25 in the discussion section. We thank the reviewer for this interesting exchange, which has resulted in a more nuanced text and interpretation.

Modifications:

-Please find the text changes in the discussion section lines 656-672